mathematical modelling

stem cell ageing, modelling, feedback regulation

**Author for correspondence:**
Wing-Cheong Lo
e-mail: wingclo@cityu.edu.hk

†Co-first author.

# Modelling stem cell ageing: a multi-compartment continuum approach

Yanli Wang[1,†], Wing-Cheong Lo[2,†] and Ching-Shin Chou[1]

[1]Department of Mathematics, The Ohio State University, Columbus, OH, USA
[2]Department of Mathematics, City University of Hong Kong, Hong Kong, People's Republic of China

 W-CL, 0000-0001-7771-5213

Stem cells are important to generate all specialized tissues at an early life stage, and in some systems, they also have repair functions to replenish the adult tissues. Repeated cell divisions lead to the accumulation of molecular damage in stem cells, which are commonly recognized as drivers of ageing. In this paper, a novel model is proposed to integrate stem cell proliferation and differentiation with damage accumulation in the stem cell ageing process. A system of two structured PDEs is used to model the population densities of stem cells (including all multiple progenitors) and terminally differentiated (TD) cells. In this system, cell cycle progression and damage accumulation are modelled by continuous dynamics, and damage segregation between daughter cells is considered at each division. Analysis and numerical simulations are conducted to study the steady-state populations and stem cell damage distributions under different damage segregation strategies. Our simulations suggest that equal distribution of the damaging substance between stem cells in a symmetric renewal and less damage retention in stem cells in the asymmetric division are favourable strategies, which reduce the death rate of the stem cells and increase the TD cell populations. Moreover, asymmetric damage segregation in stem cells leads to less concentrated damage distribution in the stem cell population, which may be more robust to the stochastic changes in the damage. The feedback regulation from stem cells can reduce oscillations and population overshoot in the process, and improve the fitness of stem cells by increasing the percentage of cells with less damage in the stem cell population.

## 1. Introduction

Stem cells are characterized by their ability to give rise to a variety of cell types through self-renewal and differentiation [1]. Although the process is very dynamic, the stem cell population is stable and remains almost steady. When a stem cell divides, each progeny

**Figure 1.** Cell divisions and damage segregation in stem cells. (*a*) Three types of division in stem cell population: SR stands for symmetric renewal, ASR & D stands for asymmetric renewal and differentiation and SD stands for symmetric differentiation. (*b*) Commonly recognized ageing factors, such as protein aggregates, dysfunction organelles and DNA damage are segregated asymmetrically between two cells during division.

has the potential to remain as a stem cell (self-renewal) or becomes a cell with a more specialized function (differentiation) (figure 1*a*). If the division produces the same type of cells, such as two stem cells and two differentiated cells, it is called a symmetric division. On the other hand, if the division produces two different types of cells, i.e. a stem cell and a differentiated cell, then it is called an asymmetric division. Available data suggest that most stem cells are able to switch between symmetric and asymmetric divisions, and the balance between these modes is controlled by various internal and external signals to produce appropriate numbers of stem and differentiated cells [1–3].

Stem cells can be found in most mammalian tissues, and they participate in tissue repair in response to damage and maintaining tissue homeostasis [4,5]. Research suggests that the decline in adult tissue maintenance and the increase in cancer formation might be a consequence of stem cell ageing [6]. Although age-related manifestation in stem cell population and function differ across tissues and organisms, decline in regenerative capacity due to depletion or dysfunction of stem cells is a hallmark [7]. Protein aggregates, dysfunction organelles and DNA damage are commonly identified as factors of ageing [4,5]. To slow down the accumulation of ageing factors, a hypothesis suggests that mitotic cells (actively dividing cells) might asymmetrically segregate damage away from the cell whose fate is to become a new stem cell [8]. The asymmetric inheritance of cellular components in dividing cells was first observed in yeast, and has been extensively studied over the past decades. In yeast, carbonylated proteins, extrachromosomal ribosomal DNA circles and dysfunction mitochondria are retained by a mother cell during asymmetric division, while its daughter is rejuvenated with little damage [9–11]. Recent evidence suggests that stem cells may employ a similar mechanism to protect one progeny from ageing [12–18] (figure 1*b*). An asymmetric partition of damaged proteins in stem cell division was observed in adult flies' intestine and germline [12], mammalian stem cells [16], and murine neural stem cells [18]. Despite these emerging findings, it still remains elusive how stem cells cope with damage accumulation and how this is related to stem cell proliferation and differentiation.

Driven by the lack of knowledge of mechanisms regulating stem cell maintenance and differentiation in the ageing process, mathematical models have been employed to address key questions and provide quantitative insights into stem cell renewal and differentiation, as well as the decline of cellular functions in the ageing process. Several mathematical models were proposed to study stem cell population and ageing, which fall into two categories: individual-based modelling and continuous population modelling [19,20].

Individual-based modelling simulates individuals or agents that have a unique set of state variables and usually interact with each other in the local environment [20,21]. The advantages of this approach include that it can take stochastic effects into consideration to describe phenomena on the level of individual cells, and that detailed molecular dynamics and cell–cell interaction can be incorporated. In [22], a stochastic model of stem cell organization was introduced to explain the observed heterogeneity of haematopoietic stem cells by the stochastic switching between the growth environments and the self-organizing process based on within-tissue plasticity properties. Assuming that cell proliferation is negatively affected by telomere shortening, an agent-based stochastic model [23] was proposed to study telomere-dependent stem cell replicative ageing. This model provides a good approximation of the qualitative growth of cultured human mesenchymal stem cells. In [24], mutation accumulation in large populations of stem cells was modelled by a discrete-time branching process where each division produces 0, 1 or 2 stem cell daughters, each of which randomly accumulates a mutation. This model

demonstrated that symmetric division could reduce the risk of accumulating phenotypically silent heritable damage in individual stem cells. However, a major drawback of individual-based modelling is the computational inefficiency, especially when the population is large. When a population is large and homogeneous, continuous population model using ODEs [25–31] or PDEs [32–38] are more appropriate to describe the population dynamics. In [39], a maturity-structured two-compartment model was proposed to study the regulation of mammalian red blood cell production. The model consists of two transport equations describing population densities of mitotic cells and post-mitotic cells, and an ODE describing hormone dynamics. The model showed that a perturbation of blood-donation type leads to damped oscillatory return to normal status, and that an elevated random peripheral destruction of red blood cells leads to sustained oscillations. In [40], a three-compartment ODE model was applied to study the dynamics of stem cells, transit-amplifying cells and terminally differentiated (TD) cells in the olfactory epithelium of mice. The authors identified conditions on parameters for the stability of the system when negative feedback loops are present either as Hill functions or in more general forms. Their analysis suggested that two factors, autoregulation of the proliferation of transit-amplifying progenitor cells and low death rate of TD cells, enhance the stability of the system. In [41], the authors applied the mean-field approach to approximate an agent-based model for studying heterogeneity within the haematopoietic stem cell population. Their proposed PDE model can capture the key structure of the model including the 'age'-structure of stem cells and improve the efficiency of the numerical algorithms. In [42], a system of PDEs was used to model mutation accumulation hierarchy and differentiation hierarchy of cells with stem cells on the top level and to examine cancer development and growth. In their model, maturity is treated as a continuous variable, while the number of mutation accumulation and telomere shortening are treated as discrete cell classes. The boundary conditions describe transition among different cell classes at division: cells lose telomeres and acquire mutations. The study showed that the more mutation classes and higher proliferation rate are sufficient to explain the faster growth of the cancer cell population.

To understand the effects of different damage segregation strategies on stem cell ageing, we propose a novel model to integrate stem cell proliferation and differentiation with damage accumulation in the stem cell ageing process, and feedback regulation from stem and TD cells. A system of hyperbolic PDEs is constructed to model two compartments in cell lineage: mitotic cells (stem cells) and post-mitotic cells (TD cells). It is assumed that the cell cycle progression of stem cells is a continuous process while stem cell division is discrete. The boundary conditions of the PDEs model the stem cell renewal and differentiation at division when damage segregation takes place. Cell death is modelled as an outcome of damage accumulation. Stem cell proliferation and differentiation are regulated by feedbacks from the population of TD cells and stem cells. Ageing effect is modelled through the inhibition from the damage accumulation on stem cell proliferation and self-renewal. Our analysis and numerical simulations are carried out to compare the effects of different regulation and damage segregation strategies on population dynamics and stem cell fitness, which have not been discussed in the previous studies of stem cell regulation.

Our simulations suggest that equal distribution of the damaging substance between stem cells in symmetric renewal and less damage retention in stem cell in asymmetric division are favourable strategies, which reduce the death rate of stem cells and increase TD cell populations. Also, asymmetric damage segregation in stem cells leads to less concentrated damage distribution in stem cells population, which may be more robust to the stochastic change in damage. Compared to the feedbacks solely from TD cells, the feedback regulation from stem cells (autoregulation) can reduce oscillations and population overshoot in the process, and improve the fitness of stem cells by increasing the percentage of stem cells with less damage in the stem cell population.

This paper is structured as follows. The general description of our model is given in §2. In §3, a simple model without feedback regulations is presented to analyse the relation between population dynamics and various parameters. In §4, two more complex models with feedbacks from TD cells and stem cells are proposed to study different regulation mechanisms and the effect of segregation strategies.

# 2. Model description

In our mathematical model, a simplified conceptual model, we consider two types of cells: mitotic cells, which include stem cells and multiple progenitor cells, and post-mitotic cells, which include all TD cells. For simplicity, we call mitotic cells stem cells, and denote its population density by $S$; and we call post-mitotic cells TD cells, and denote its population density by $T$.

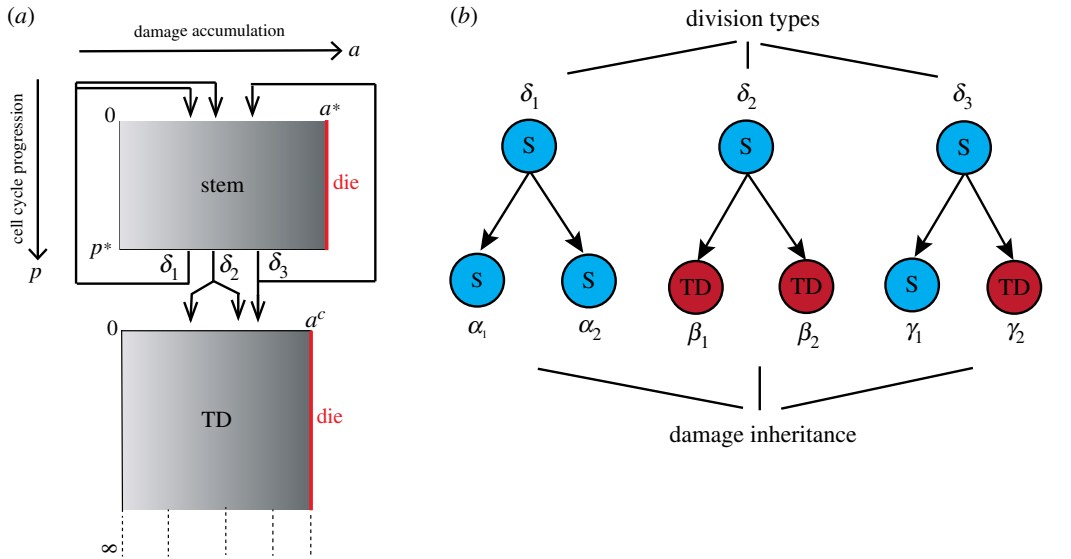

**Figure 2.** Model descriptions of two-compartment stem cell system. Stem cells (mitotic cells) and TD cells (post-mitotic cells) are modelled as two compartments. (a) Stem cells renew themselves and replenish TD cells. The cell cycle progression ($p$) and damage accumulation ($a$) are modelled as continuous processes. Stem cell division and damage segregation take place at the end of its cell cycle ($p = p^*$). Cells die when damage reaches a lethal threshold ($a^*$ or $a^c$). (b) In the simple model in §3, the proportions of three types of division are constants $\delta_i$, and the damage segregation rules are fixed, i.e. $\alpha_i$, $\beta_i$, $\gamma_i$ are constants.

To consider the evolution of cell populations, we define both $S(t, p, a)$ and $T(t, p, a)$ as functions of time $t$ and two other continuous biological state variables: cell cycle progression $p$ and damage level $a$. To describe cell maturation and ageing processes, we make the following assumptions:

— When the amount of damage $a$ accumulates and reaches a certain threshold $a^*$, stem cells die by an apoptosis-like process as a result of ageing via damage accumulation; similar to stem cells, when the amount of damaged substance reaches a certain threshold $a^c$, TD cells are removed by an apoptosis-like process.
— Cell cycle progression $p$ is an indicator variable, for which a stem cell divides when $p$ increases to a threshold $p^*$, unless damage has already reached a threshold $a^*$, in which case the stem cell is removed before division. Although TD cells no longer divide, for simplicity, we keep cell cycle progression variable $p$ in $T$, but assume the upper boundary for $p$ is infinity.

By conservation law, a system of transport equations are derived to describe the evolution of $S$ and $T$:

$$\frac{\partial S}{\partial t} + \frac{\partial}{\partial p}(V_p S) + \frac{\partial}{\partial a}(V_a S) = 0 \tag{2.1}$$

and

$$\frac{\partial T}{\partial t} + \frac{\partial}{\partial p}(U_p T) + \frac{\partial}{\partial a}(U_a T) = 0, \tag{2.2}$$

where $V_p$ and $U_p$ are rates of cell cycle progression for stem and TD cells, respectively; $V_a$ and $U_a$ are rates of damage accumulation for stem and TD cells, respectively. Note that various of feedbacks may regulate cell cycle progression and cell damage accumulations, and these functions will be specified in the following sections.

The boundary conditions at $p = 0$ describe the reproduction/division process accompanied by the segregation of damage substances. First, we assume that there are three types of cell divisions:

 (i) The daughter cells after division are two stem cells; this occurs with probability $\delta_1$.
 (ii) The daughter cells after division are two TD cells; this occurs with probability $\delta_2$.
 (iii) The daughter cells after division are one stem cell and one TD cell; this occurs with probability $\delta_3$.

One of these three types of cell division occurs at the end of cell cycle $p = p^*$ with probability $\delta_1$, $\delta_2$ or $\delta_3$, where $\delta_1 + \delta_2 + \delta_3 = 1$ (figure 2a). These three probabilities may be regulated by various feedbacks and will be specified in later sections.

Upon the completion of a cell cycle, the stem cell cycle progression $p$ will be reset to zero and the damaged proteins are inherited from mother to daughters. The damage inheritance can be described by transition kernels $r_{i,S}(a, a')$ and $r_{i,T}(a, a')$. Based on the above assumptions, the boundary conditions for $S$ and $T$ cells at $p = 0$ are as follows:

$$V_p S(t, 0, a) = \int_0^{a^*} \delta_1 V_p r_{1,S}(a, a') S(t, p^*, a') \, da' + \int_0^{a^*} \delta_3 V_p r_{3,S}(a, a') S(t, p^*, a') \, da' \tag{2.3}$$

and

$$U_p T(t, 0, a) = \int_0^{a^*} \delta_2 V_p r_{2,T}(a, a') S(t, p^*, a') \, da' + \int_0^{a^*} \delta_3 V_p r_{3,T}(a, a') S(t, p^*, a') \, da'. \tag{2.4}$$

The first term of the right-hand side of (2.3) represents the stem cell production process through type (i) cell division; the second term of the right-hand side of (2.3) represents the stem cell process through type (iii) cell division; the first term of the right-hand side of (2.4) represents the TD cell production process through type (ii) cell division; the second term of the right-hand side of (2.4) represents the TD cell production process through type (iii) cell division. The transition kernel $r_{i,S}(a, a')$ describes how daughter stem cells with damage $a$ come from the mother cells with damage $a'$ after the $i$-th type of division; the transition kernel $r_{i,T}(a, a')$, describes how daughter TD cells with damage $a$ come from the mother cells with damage $a'$ in the $i$-th type of division. The transition kernels satisfy the following conservation conditions:

$$\int_0^{a^*} \int_0^{a^*} r_{1,S}(a, a') \, da \, da' = 2, \quad \int_0^{a^*} \int_0^{a^c} r_{2,T}(a, a') \, da \, da' = 2,$$

$$\int_0^{a^*} \int_0^{a^*} r_{3,S}(a, a') \, da \, da' = 1, \quad \int_0^{a^*} \int_0^{a^c} r_{3,T}(a, a') \, da \, da' = 1,$$

$$\int_0^{a^*} a \, r_{1,S}(a, a') \, da = a', \quad \int_0^{a^c} a \, r_{2,T}(a, a') \, da = a'$$

$$\int_0^{a^*} a \, r_{3,S}(a, a') \, da + \int_0^{a^c} a \, r_{3,T}(a, a') \, da = a'.$$

Since type (i) and type (ii) divisions induce two daughter stem cells and two daughter TD cells, respectively, the integrals of $r_{1,S}(a, a')$ and $r_{2,T}(a, a')$ equal 2; since type (iii) division induces one daughter stem cell and one daughter TD cell, the integrals of $r_{3,S}(a, a')$ and $r_{3,T}(a, a')$ both equal one. The last three integrals are based on the assumption that the amount of damage is conserved during cell division. No-flux conditions on the boundary $a = 0$ are imposed to ensure conservation of population in the direction of $a$

$$S(t, p, 0) = 0, \quad \text{for } t > 0, \ p \in [0, p^*] \tag{2.5}$$

and

$$T(t, p, 0) = 0, \quad \text{for } t > 0, \ p \in [0, \infty). \tag{2.6}$$

For TD cells, we assume that the population density of TD is zero when $p$ is very large

$$\lim_{p \to \infty} T(t, p, a) = 0. \tag{2.7}$$

The populations of stem and TD cells at time $t$ are given by the integrals

$$P_S(t) = \int_0^{a^*} \int_0^{p^*} S(t, p, a) \, dp \, da \quad \text{and} \quad P_T(t) = \int_0^{a^c} \int_0^{\infty} T(t, p, a) \, dp \, da.$$

Based on this model, we will study the population dynamics under two scenarios:

(i) For the simplest case where no feedbacks are involved in regulating the stem cell division and differentiation, the dynamics of stem cell population mainly depend on stem cell damage segregation rules. Different segregation rules may result in exploded, stabilized or extinct stem cell pool.

(ii) Feedbacks from TD cells and stem cells are introduced and population evolutions are simulated to study the effect of different damage segregation rules.

# 3. Model without feedback regulations

In this section, we start with the simple model where no feedbacks are included, and we additionally make the following assumptions to simplify the model (figure 2b):

— Cell cycle progression velocities $V_p = v_p$ and $U_p = u_p$ are constants, so are the damage accumulation rates $V_a = v_a$ and $U_a = u_a$.
— The portions of three types of divisions $\delta_i$ are constants.
— When stem cells divide, damage is partitioned into portions $\alpha_1$ and $\alpha_2$ between stem cells, $\beta_1$ and $\beta_2$ between TD cells, or $\gamma_1$ and $\gamma_2$ between stem and TD cells, where $\alpha_1 + \alpha_2 = 1$, $\beta_1 + \beta_2 = 1$ and $\gamma_1 + \gamma_2 = 1$. Without loss of generality, we assume $\alpha_1 \leq \alpha_2$ and $\beta_1 \leq \beta_2$.

Under these assumptions, the transition kernels in the boundary conditions are Dirac delta functions

$$r_{1,S}(a, a') = \delta\left(\frac{a}{a'} - \alpha_1\right) + \delta\left(\frac{a}{a'} - \alpha_2\right), \quad r_{2,T}(a, a') = \delta\left(\frac{a}{a'} - \beta_1\right) + \delta\left(\frac{a}{a'} - \beta_2\right),$$

$$r_{3,S}(a, a') = \delta\left(\frac{a}{a'} - \gamma_1\right), \quad r_{3,T}(a, a') = \delta\left(\frac{a}{a'} - \gamma_2\right),$$

and the boundary conditions in (2.3) and (2.4) become

$$S(t, 0, a) = \frac{\delta_1}{\alpha_1} S\left(t, p^*, \frac{a}{\alpha_1}\right) + \frac{\delta_1}{\alpha_2} S\left(t, p^*, \frac{a}{\alpha_2}\right) + \frac{\delta_3}{\gamma_1} S\left(t, p^*, \frac{a}{\gamma_1}\right) \tag{3.1}$$

and

$$T(t, 0, a) = \frac{v_p}{u_p} \frac{\delta_2}{\beta_1} S\left(t, p^*, \frac{a}{\beta_1}\right) + \frac{v_p}{u_p} \frac{\delta_2}{\beta_2} S\left(t, p^*, \frac{a}{\beta_2}\right) + \frac{v_p}{u_p} \frac{\delta_3}{\gamma_2} S\left(t, p^*, \frac{a}{\gamma_2}\right). \tag{3.2}$$

Now, we study the role of the damage segregation rules in the long-term behaviour of the stem cell population. Since the segregation rule is fixed, after sufficiently long time, the cellular damage at the end of cell cycle, temporarily assuming no death, converges to a *limit damage band*

$$\left[\frac{p^*}{v_p(1 - \omega_1)}, \frac{p^*}{v_p(1 - \omega_2)}\right],$$

where $\omega_1$ and $\omega_2$ are the minimum and the maximum of $\alpha_1, \alpha_2, \gamma_1$, respectively. The derivation of damage limit band can be found in appendix A 1.

The population dynamics turns out to depend on the proportions of three division types of stem cells and the position of the lethal threshold $a^*/v_a$ with respect to the limit damage band $[p^*/v_p(1 - \omega_1), p^*/v_p(1 - \omega_2)]$. Before proceeding, we define a key lumped parameter for studying the long-term population behaviour

$$f_r = \frac{2\delta_1 + \delta_3}{2} = \frac{1 + \delta_1 - \delta_2}{2}.$$

Here, the parameter $f_r$ is called the *self-renewal fraction*. Using the limit damage band and the self-renewal fraction, we can find some conditions for different long-term behaviours.

**Proposition 3.1.** *Stem cells become extinct, if the renewal fraction $f_r < 1/2$ or the limit damage band is completely above the lethal threshold, i.e. $a^*/v_a < p^*/v_p(1 - \omega_1)$.*

**Proposition 3.2.** *Assume that the limit damage band lies completely below the lethal threshold, i.e. $a^*/v_a > p^*/v_p(1 - \omega_2)$ and*

$$\int_0^{p^*} \int_0^{a^* - v_a(p^* - p)/v_p} S\left(0, p, a\right) \mathrm{d}a \, \mathrm{d}p > 0.$$

*Then stem cell population blows up if the renewal fraction $f_r > 1/2$, or is eventually conserved if $f_r = 1/2$.*

The proofs of propositions 3.1 and 3.2 can be found in appendix A 2. In the situations shown in propositions 3.1 and 3.2, either damage accumulation does not affect cells or no cell can survive after sufficiently long time. Next, we consider the more intriguing intermediate situations as follows, based

on the following assumptions on the renewal fraction and the lethal threshold $a^*/v_a$ lies within the limit damage band

$$f_r > \frac{1}{2} \quad \text{and} \quad \frac{p^*}{v_p(1-\omega_1)} \le \frac{a^*}{v_a} < \frac{p^*}{v_p(1-\omega_2)}. \tag{3.3}$$

Under the above assumptions in (3.3), there are many situations of damage segregation rules, and more importantly not all of the situations are biologically meaningful. Here, we will focus on one situation: the damage retention in stem cells through asymmetric division is smaller than the damage segregation portions in symmetric stem cell renewal, i.e.,

$$\gamma_1 \le \alpha_1 < \alpha_2. \tag{3.4}$$

This setting is based on the biological observation that asymmetric division is a favourable mechanism for stem cell lineage to remove the damage. For avoiding the case that the lethal threshold $a^*/v_a$ lies below $p^*/v_a(1-\alpha_1)$, i.e. all asymmetrically renewing stem cells are destined to die, we further assume that

$$\frac{p^*}{v_p(1-\alpha_1)} < \frac{a^*}{v_a}. \tag{3.5}$$

By the assumptions (3.3)–(3.5), we obtain that

$$f_r > \frac{1}{2} \quad \text{and} \quad \frac{p^*}{v_p(1-\gamma_1)} \le \frac{p^*}{v_p(1-\alpha_1)} < \frac{a^*}{v_a} < \frac{p^*}{v_p(1-\alpha_2)}. \tag{3.6}$$

Under the assumption (3.6), the stem cell population may approach zero or blow up in the long-term behaviour. The following proposition provides a condition to guarantee extinction of stem cells, as well as a condition for population blow-up, with the proof given in appendix A 2.

**Proposition 3.3.** *Assume that* (3.6) *holds.*

(a) *Define*

$$n = \min\left\{ m \in \mathbb{N} : \frac{a^*}{v_a} < \frac{p^*}{v_p(1-\alpha_2)} - \frac{p^*}{v_p}\left(\frac{1}{1-\alpha_2} - \frac{1}{1-\gamma_1}\right)\alpha_2^m \right\}. \tag{3.7}$$

*If there exists a combination of* $\delta_i$ *such that* $(2f_r)^n - \delta_1^n < 1$, *then the stem cell population is eventually zero.*

(b) *If* $\delta_2 = 0$ *and*

$$\int_0^{p^*} \int_0^{a^* - v_a(p^*-p)/v_p} S(0, p, a) \, \mathrm{d}a \, \mathrm{d}p > 0,$$

*then the stem cell population goes to infinity, or is bounded below to avoid stem cell extinction.*

From proposition 3.3, we can obtain several necessary conditions to maintain the stem cell population. Since the term $(2f_r)^n - \delta_1^n$ is increasing with $n$ and goes to infinity when $n$ tends to infinity, $n$ has to be small enough to satisfy the condition $(2f_r)^n - \delta_1^n < 1$ in proposition 3.3. From the definition of $n$, we observe that if

— the difference between $a^*/v_a$ and $p^*/v_p(1-\alpha_2)$ becomes small, or
— the difference between $1/(1-\alpha_2)$ and $1/(1-\gamma_1)$ becomes large, or
— the right-hand side $p^*/v_p(1-\alpha_2) - (p^*/v_p)(1/(1-\alpha_2) - (1/(1-\gamma_1)))\alpha_2^m$ becomes large for each $m$,

the value of $n$ will increase and then $(2f_r)^n - \delta_1^n < 1$ cannot be satisfied in most of the combinations of $\delta_1$, $\delta_2$ and $\delta_3$. This result provides some conditions of the parameters to prevent stem cell extinction.

Let us discuss the above three possibilities. To reduce the difference between $a^*/v_a$ and $p^*/v_p(1-\alpha_2)$, the ratio of cell cycle progression speed $v_p$ to damage accumulation speed $v_a$ should be large enough and close to $p^*/a^*(1-\alpha_2)$; to increase the difference between $1/(1-\alpha_2)$ and $1/(1-\gamma_1)$, damage retention in asymmetric division should decrease, i.e. $\gamma_1$ should be small; to increase the right-hand side $p^*/v_p(1-\alpha_2) - (p^*/v_p)(1/(1-\alpha_2) - (1/(1-\gamma_1)))\alpha_2^m$ for each $m$, damage distribution in self-renewal should become more symmetric, i.e. $\alpha_1$ should increase to close to 0.5. In conclusion, when $v_p/v_a$ increases, $\gamma_1$ decreases or $\alpha_1$ increases, it may provide a better condition to maintain the stem cell population. These results are supported by the numerical simulations shown in figure 3.

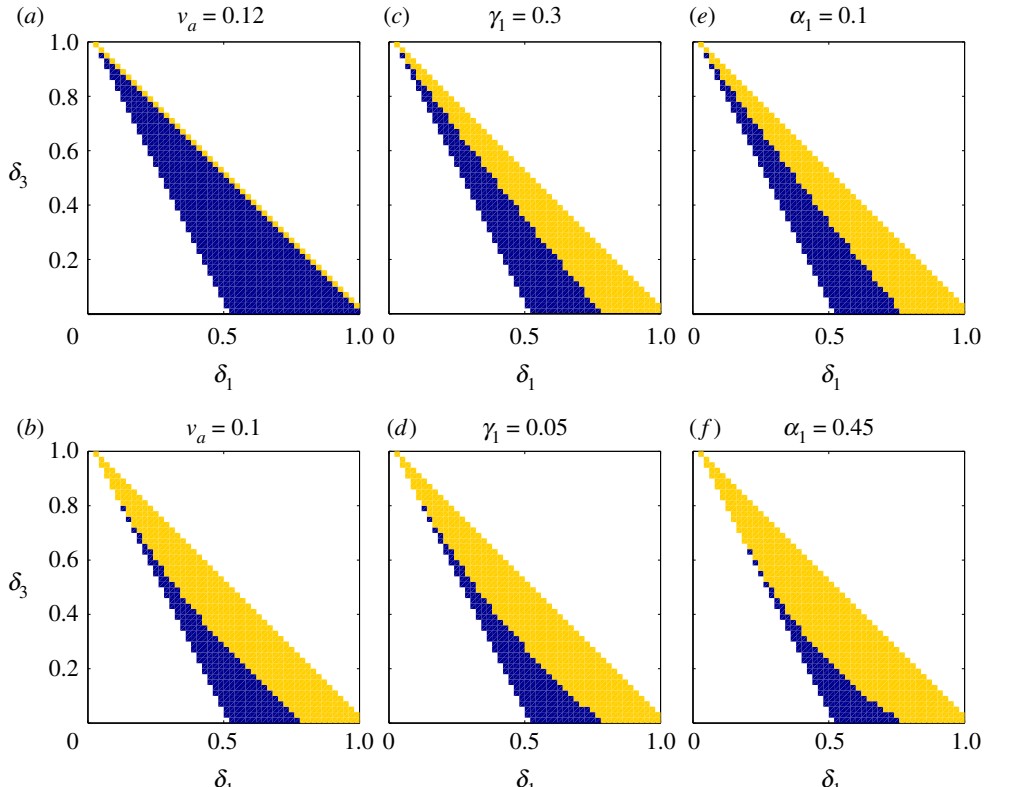

**Figure 3.** The long-term stem cell population dynamics with different combinations of the division probabilities $\delta_1$ and $\delta_3$. The yellow regions represent the combinations of $(\delta_1, \delta_3)$ whose corresponding population blow up or is eventually conserved, while the blue regions represent the situation where population goes to zero. The parameter values are set to be $\alpha_1 = 0.3$, $\gamma_1 = 0.1$, $p^* = 1$, $a^* = 1$, $v_p = 0.2$ and $v_a = 0.1$, if not mentioned in the subfigures.

In figure 3, we study the long-term population dynamics with different combinations of the division probabilities $\delta_1$ and $\delta_3$. We uniformly generate 600 pairs of $\delta_1$ and $\delta_3$ with $0 < \delta_1 + \delta_3 \leq 1$ and $2\delta_1 + \delta_3 \geq 1$. We skip the region $2\delta_1 + \delta_3 < 1$ as it implies stem cell extinction whatever other parameters are (by proposition 3.1). For each pair of $\delta_1$ and $\delta_3$, the system is solved by the numerical method described in appendix A5. The blue regions in figure 3 represent the cases of stem cell extinction and the yellow regions represents the non-extinction cases (the stem cell population may tend to a constant non-zero value, approach infinity, or keep oscillating). When $v_p/v_a$ increases (in figure 3a, $v_p/v_a = 1.67$; in figure 3b, $v_p/v_a = 2$), $\gamma_1$ decreases (in figure 3c, $\gamma_1 = 0.3$; in figure 3d, $\gamma_1 = 0.05$) or $\alpha_1$ increases (in figure 3e, $\alpha_1 = 0.1$; in figure 3f, $\alpha_1 = 0.45$), the blue regions become smaller (the yellow regions become larger), which implies that there are more combinations of parameters $(\delta_1, \delta_3)$ allowing stem cell survival.

For most of the combinations of parameters, the stem cell populations in the models without feedback regulation blow up to infinity or diminish to zero. Although the no-regulation assumption is not realistic, the simple model not only provides us a guidance on the selection of parameters used in the model with feedbacks but also reveals that population dynamics are results of all factors: the cell cycle progression of stem cell, damage accumulation, fractions of divisions and damage segregation rules. In the next section, we will consider the combinations of parameters that guarantee exponential growth in the stem cell population and study feedback regulations that could lead to non-zero steady-state population.

# 4. Model with feedback regulations

Biological evidence shows that some mammalian stem cells can switch between symmetric and asymmetric divisions in response to external and internal regulations [1,3]. For example, both epidermal [43] and neural [44] progenitors change from the primarily symmetric division that expands the stem-cell pool during embryonic development to primarily asymmetric in mid to late gestation. It is also observed that nervous [45] and haematopoietic [46] stem cells in adults can divide symmetrically to replace lost cells through injury, although they divide asymmetrically under

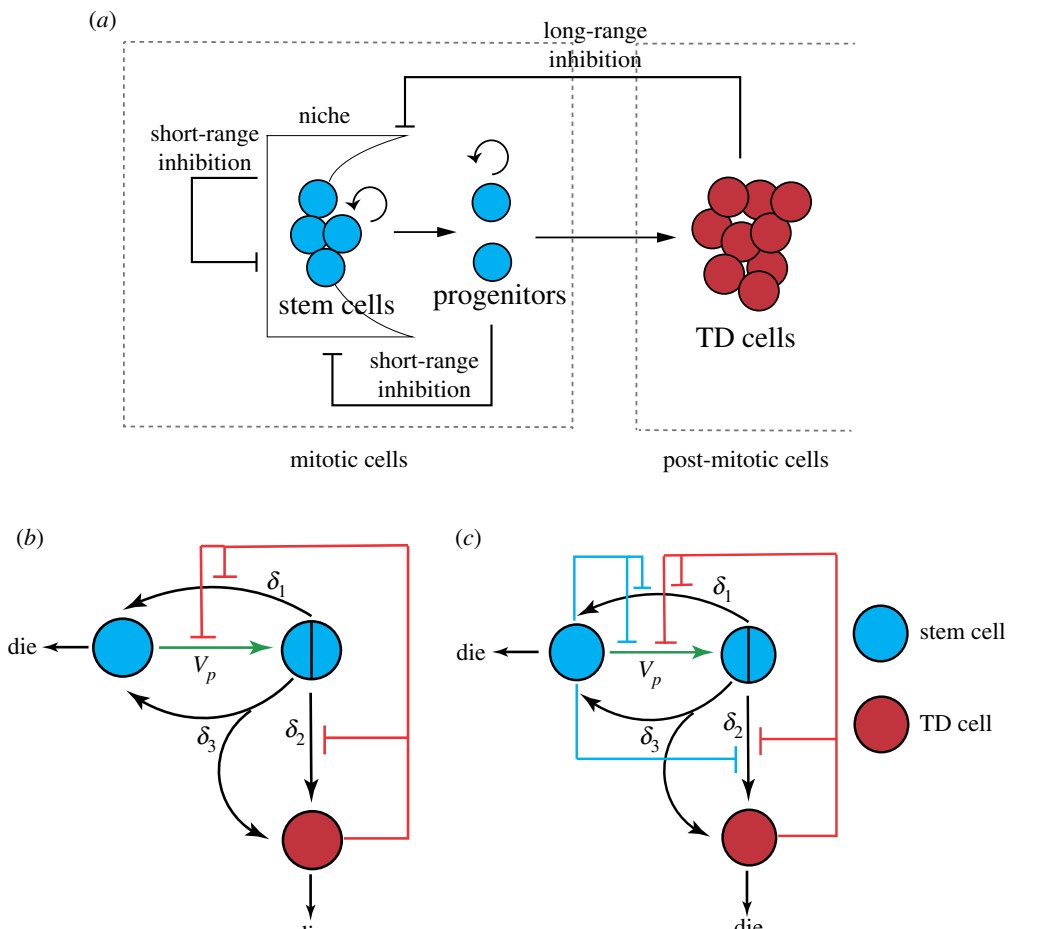

**Figure 4.** Feedbacks in stem cell lineage. (*a*) Two types of feedback occur: the long-range feedback responding to the population of TD cells, and short-range feedback acting in an autocrine fashion in stem cells. (*b*) Two-compartment model with feedbacks from TD cells. (*c*) Two-compartment model with feedbacks from stem cells and TD cells.

steady-state conditions. In particular, two types of feedbacks have been proposed [34,47,48] (also see figure 4*a*): long-range feedbacks responding to the population of TD cells, and short-range feedbacks acting in an autocrine fashion from stem cells. These two types of feedbacks regulate stem cell population through controlling two types of parameters:

  (i)  the speed of stem cell division, $V_p$ (or stem cell cycle progression),
  (ii) the probabilities of three types of cell division, $\delta_i$ (differentiation versus renewal).

However, the underlying mechanisms are not fully understood. In addition, the effect of damage segregation rules on cell population is completely unknown. In the following, we use mathematical models with different feedbacks and damage segregation rules to study which types of feedback and damage segregation mechanisms are more plausible given the experimental observations.

For simplicity, we assume that damage segregation rules are fixed, i.e. $\alpha_i, \beta_i, \gamma_i$ are constants. Based on the results from the simple model in §3, to ensure that the stem cell population will blow up to infinity without any feedback regulation, additional assumptions are made:

— Self-renewal fraction $f_r > \frac{1}{2}$.
— Damage accumulation speed $V_a = v_a$ is slower than cell cycle progression of stem cells.

## 4.1. Feedback only from TD cells

The long-range feedback, which acts through signals sent by differentiated cells and inhibits stem cell division and self-renewal, has been biologically observed in numerous tissues, including muscle [49], bone [50], skin

[51], nervous system [52] and haematopoietic systems [53]. Despite this wealth of data, there is less understanding of the exact mechanisms of feedback regulation. A significant number of mathematical models have been developed to explore the possible mechanisms behind feedback regulations [26,27,32,34–40,54,55]. The dynamics of signalling molecule $s(t)$ can be described by a simple ODE as follows:

$$\frac{d}{dt}s(t) = \alpha - (\mu + \beta P^m)s(t), \tag{4.1}$$

where $\alpha$ is a constant synthesis rate, and the degradation is proportional to the level of $s$ and affected by the cell population $P$ of stem or TD cells. Since the dynamics of the signalling molecules take place on a faster time scale than the processes of cell proliferation and differentiation, we assume that the feedback signal can be approximated by a quasi-steady-state solution [25,27,32,34–40,54,55]. By properly rescaling $s$, the quasi-steady state of $s$ is given by a Hill function

$$s = \frac{1}{1 + (kP)^m}, \tag{4.2}$$

where $k$ is a regulation constant to account for the sensitivity to the cell population and $m$ is the Hill exponent. The function in (4.2) reflects the assumption that the signal intensity achieves its maximum in the absence of cells and decreases asymptotically to zero if the number of cells increases. Note that Hill functions are widely used to describe ligand–receptor interactions, which also makes them natural choices to model the actions of secreted feedback factors [37].

According to biological evidence, we may model the feedback from TD cells in the following ways (figure 4$b$): when stem cell or TD cell population is small (in the early stage of development or with drastic loss due to injury), symmetric division predominates over asymmetric division; during the stable stage (mid and late gestation or tissue homeostasis), stem cells switch to asymmetric division. Such feedbacks may be added to the model by modifying division fractions $\delta_i$ as follows:

$$\delta_i(P_T) = \frac{\delta_i^0}{1 + (k_i^T P_T)^{m_T}}, \tag{4.3}$$

where $i \in \{1, 2\}$, $\delta_i^0$ are basal division fractions, $k_i^T$ are regulation constants, $m_T$ is the Hill exponent and $P_T(t)$ is the TD cell population. The remaining division fraction $\delta_3$ is defined as $1 - \delta_1 - \delta_2$.

Recent studies on cell cycles of neural stem cells have shown evidence that TD cells may be a source of signalling molecules that inhibits cell cycle progression of stem cells [56]. Therefore, besides the above regulation on cell fate decisions, negative feedback from TD cells can also be involved in stem cell proliferation $V_p$ (cell cycle progression speed), i.e. excessive number of TD cell may slow down the proliferation of stem cell, and hence reduce the population of both stem cells and TD cells. Thus, $V_p$ can be modified as

$$V_p(P_T) = \frac{v_p}{1 + (k_v^T P_T)^{m_T}}, \tag{4.4}$$

where $v_p$ is the initial cell cycle progression speed and $k_v^T$ is a regulation constant. For other velocities, we set $V_a = v_a$, $U_a = u_a$ and $U_p = u_p$.

Among all parameters, we are most interested in the effect of segregation rules $\alpha_i$, $\beta_i$, $\gamma_i$ on the population dynamics. A set of reasonable estimations of the parameters other than $\alpha_i$, $\beta_i$, $\gamma_i$ are chosen based on appropriate biological and mathematical assumptions, and are shown in table 1. The detailed reasoning of the selection of parameters is provided in appendix A.3.1. In brief, we assume that initially, the stem cells are expanding, the cell cycle progression is fast, and the symmetric renewal predominates in three types of division. In the rest of this subsection, we will focus on the discussion of the effect of different combinations of segregation rules $\alpha_i$, $\beta_i$, $\gamma_i$.

In the simulations, we consider the following situations: $\alpha_1$ varies from 0.1 to 0.5, indicating that the symmetry in damage segregation increases between the two stem daughter cells; $\beta_1$ varies from 0.1 to 0.5, indicating that the symmetry in damage segregation increases between TD cells; $\gamma_1$ varies from 0.1 to 0.9, indicating that the damage retention in stem cells increases in asymmetric division. Under different segregation rules, the following aspects are studied:

— the population size of TD cells and the population ratio of TD cells to stem cells at steady state;
— the death rate of stem cells and the fraction of three types of division at steady state; and
— the dynamics of population evolution and the damage distribution in stem cell population at steady state.

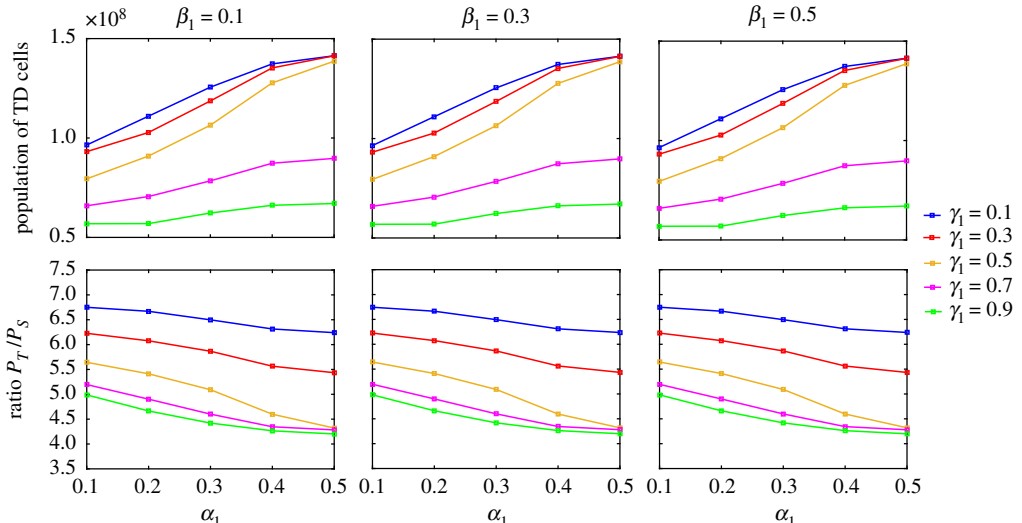

**Figure 5.** Model with feedback only from TD cells: TD cell population and population ratio at steady state for different combinations of $\alpha_1$, $\beta_1$ and $\gamma_1$. The parameters used in these simulations are shown in table 1.

**Table 1.** The choices of parameters, estimated based on biological evidence and previous modelling works, with details in appendix A.3.1.

| parameters | meaning | value |
| --- | --- | --- |
| $p^*$ | cell cycle threshold | 1 |
| $a^*$ | damage level threshold for stem cell | 1 |
| $a^c$ | damage level threshold for TD cell | 1 |
| $v_p$ | maximum cell cycle progression speed of stem cells | 0.2 |
| $v_a$ | constant damage accumulation speed of stem cells | 0.06 |
| $u_p$ | constant cell cycle progression speed of TD cells | 0.02 |
| $u_a$ | constant damage accumulation speed of TD cells | 0.02 |
| $\delta_1^0$ | maximum fraction of symmetric renewal of stem cells | 0.6 |
| $\delta_2^0$ | maximum fraction of symmetric differentiation of stem cells | 0.3 |
| $\delta_3^0$ | maximum fraction of asymmetric division of stem cells | 0.1 |
| $k_1^T$ | regulation constant | $10^{-8}$ |
| $k_2^T$ | regulation constant | $0.5 \times 10^{-8}$ |
| $k_v^T$ | regulation constant | $0.5 \times 10^{-8}$ |
| $m_T$ | Hill exponent | 2 |

### 4.1.1. Population size of TD cells and population ratio of TD cells to stem cells

Figure 5 shows that the damage distribution between TD cells in symmetric differentiation ($\beta_1$ and $\beta_2$) does not affect population size at steady state. When the parameter $\beta_1$, which determines how damage is distributed between two TD cells in the symmetric differentiation of stem cells, varies from 0.1 to 0.5, neither stem cell population nor TD cell population has a big change. This result is biologically meaningful, since the replenishment of TD cells is efficient compared to the loss of TD cells due to damage accumulation. In the following part, we will focus on the case with $\beta_1 = 0.5$.

Figure 5 also shows that as damage segregation $\alpha_i$ between stem cells becomes more symmetric, TD cell populations increase, while the ratios $P_T/P_S$ are almost constant with a slight decrease. For fixed $\alpha_1$, as damage retention $\gamma_1$ in asymmetric division increases, both TD cell populations and the population ratios decrease. Interestingly, when $\alpha_1 = 0.5$, although the populations of TD cells are very close for the cases $\gamma_1 = 0.1$, 0.3 and 0.5, the population ratios differ dramatically. Compared to $\gamma_1 = 0.1$, one needs a

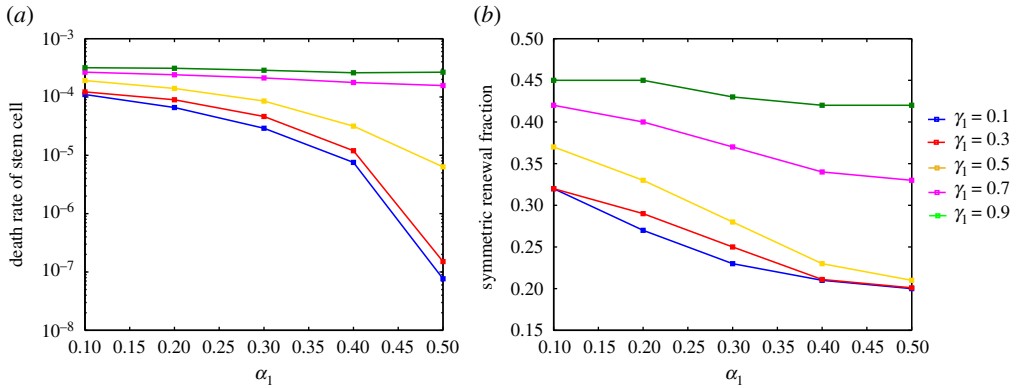

**Figure 6.** Results of the death rates of stem cell and the fractions of symmetric division in steady state for the model with feedbacks from TD cells under different damage segregation rules. (*a*) Death rate of stem cell at steady state. The definition of death rate can be found in equation (4.5). (*b*) Symmetric renewal fraction $\delta_1$ at steady state. The parameters used in these simulations are shown in table 1 and $\beta_1 = 0.5$.

larger stem cell pool to generate a similar number of TD cells when damage retention is relative higher, i.e. $\gamma_1 = 0.5$. Moreover, it would be very difficult to find explicit formulae for the populations, since the population sizes at steady state are affected by all parameters as observed in the simulations. However, when steady state exists, we can give an upper-bound estimate of the populations of stem cells and TD cells, with details given in appendix A4.

### 4.1.2. Death rate of stem cells and fraction of three types of division

Other than studying the population size, we also compare the death rate of stem cells and the fraction of three types of division at steady state. In our model, the death rates of stem cell ($r_S$) and TD cell ($r_T$) are measured in the following way:

$$r_S = \frac{\int_0^{p^*} v_a S(t, p, a^*)\, \mathrm{d}p}{P_S} \quad \text{and} \quad r_T = \frac{\int_0^{\infty} u_a T(t, p, a^c)\, \mathrm{d}p}{P_T}, \tag{4.5}$$

where the numerators are population out-fluxes due to death and the denominators are the total populations. Figure 6 shows that more equal distribution of damage in symmetric renewal and less damage retention in asymmetric division result in a lower death rate of stem cells and less symmetric division at steady state. According to our simulations, the segregation rules do not have much impact on the death rate of TD cells, which ranges from $2.130 \times 10^{-4}$ to $3.345 \times 10^{-4}$. However, the death rate of stem cells changes dramatically as we vary the segregation rules. From figure 6*a*, we can see that the death rate of stem cells increases as $\alpha_1$ decreases or $\gamma_1$ increases. The lowest death rate of stem cells is attained when damage is equally distributed between stem cells in symmetric renewal and damage retention is minimal in asymmetric division. To maintain the steady stem cell population and to replenish the short-lived TD cells, a higher death rate should be associated with a fast turnover of stem cells. Indeed, when we examine the fractions of three types of division, we find that the higher death rate of stem cells is always associated with the higher fraction of symmetric renewal (figure 6*b*). These results suggest that to maintain stabilized populations, tissues may have different mechanisms that involve different damage segregation rules and division rules.

### 4.1.3. Dynamics of population evolution and damage distribution of stem cell population

The comparison of population dynamics among different combinations of $\alpha_1$ and $\gamma_1$ (figure 7*a*) shows that higher damage retention results in oscillations in population evolution. Oscillations start to appear as we increase the damage retention $\gamma_1$ in stem cells during asymmetric division ($\gamma_1 = 0.7$ in figure 7*a*). Also, the oscillations become more severe when the damage distribution among stem cells in symmetric renewal becomes more symmetric ($\alpha_1 = 0.5$ and $\gamma_1 = 0.7$ in figure 7*a*). Moreover, population overshoots before steady state are observed in all cases with $\alpha_1 \neq 0.5$.

The results above suggest that more equal distribution of damage in symmetric renewal and less damage retention in asymmetric division are favourable segregation rules in four aspects: populations converge to steady states faster without oscillations or severe population overshoot; the death rate of stem cells is

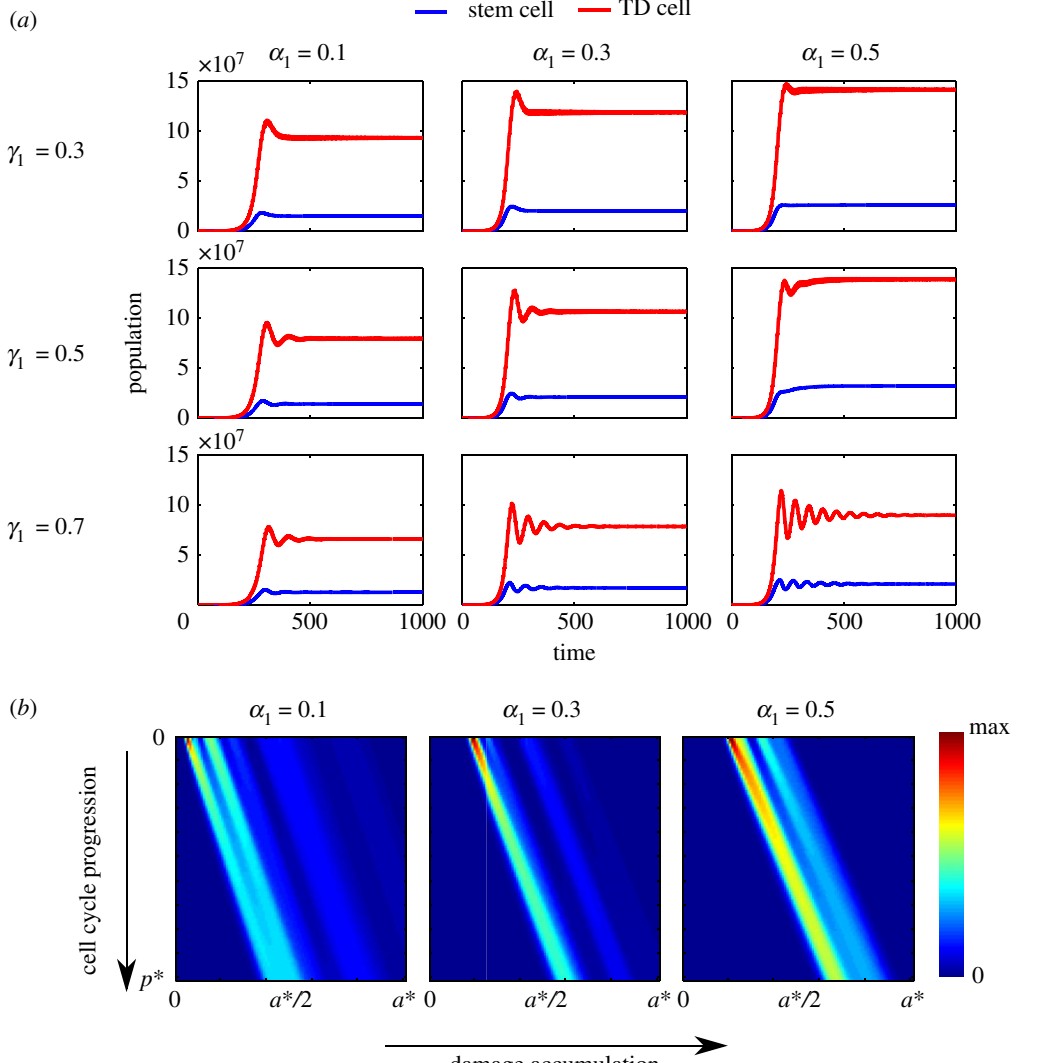

**Figure 7.** Dynamics of population evolution and damage distribution of stem cell population at steady state for the model with feedbacks from TD cells under different damage segregation rules. (a) Sample population dynamics under different combinations of $\alpha_1$ and $\gamma_1$. The red curves stand for TD cell population and the blue curves are for stem cell population. (b) Population density and damage distribution for stem cells. In these three cases, we set $\gamma_1 = 0.3$. The other parameters used in these simulations are listed in table 1 and $\beta_1 = 0.5$.

much smaller than that of TD cells; asymmetric division predominates in three types of division at steady state; not only the size of TD cell population is larger, the population ratio $P_T/P_S$ is also larger.

However, the asymmetric damage segregation between stem cells in self-renewal may have benefits that cannot be shown from the analysis of total populations as integrated results. Our PDE modelling approach allows us to investigate more details of population density and damage distribution of stem cells. Figure 7b gives the stem cell population densities corresponding to different segregation rules. As we studied in §3, damage segregation rules determine the limit damage band after a sufficiently long time. In particular, the less symmetric the segregation rules are, the wider the limit damage bands will be. Although asymmetric segregation rules may result in a greater percentage of stem cells inheriting more damage and accelerated death, it also leads to a higher percentage of healthier stem cells with less damage. This can be observed in the samples shown in figure 7b. When the symmetry of damage segregation in stem cell population increases, the damage distributions at the end of cell cycle for stem cells become more concentrated, as such symmetry increases. The concentrated damage distribution could be a disadvantage, since it is less resistant to external perturbations. In this sense, we think that some stem cells may sacrifice the lower death rate for more diversified damage distribution, in order to protect the stem cell pool from a possible unfavourable situation that may lead to a sudden increase in damage.

## 4.2. Feedback from both TD and stem cells

The maintenance of the stem cell pool is not only affected by signalling from mature cells, but also by the stem cell pool itself. Stem cells reside in the so-called stem cell niche, where both cellular and non-cellular components interact in order to control stem cell proliferation and differentiation [57–60]. The regulations from the stem cell population include two aspects: a negative feedback control of stem cell proliferation as a result of contact inhibition; and a self-renewal inhibition factor secreted by the stem cell niche, in which case the more stem cells there are, the less likely stem cells will divide symmetrically. Experiments also show that cells inheriting the majority of damage protein aggregates during asymmetric division have an increased cell cycle length and tendency to differentiate [13,14,18].

In addition to feedback regulations from differentiated cells, we are also interested in how the dynamics of populations would change if regulations from stem cells are included (figure 4c). Many modelling works (e.g. [34,40]) only consider the feedbacks from stem cell populations, due to the simplicity of their models. However, we include damage as a state variable in our model and could simulate feedbacks from both the stem cell population and the stem cell cellular damage level.

Thus, in our model, we assume that the excessive stem cell population and elevated stem cell cellular damage slow down stem cell proliferation and modify $V_p$ in (4.4) as

$$V_p(P_T, P_S, a) = \frac{v_p}{1 + (k_v^T P_T)^{m_S} + (k_v^S P_S)^{m_S}} f(a), \tag{4.6}$$

where

$$f(a) = a_1 + \frac{b_1}{1 + e^{-k_1^a(a - a_1^0)}}$$

is a decreasing function of $a$ with sigmoid shape, in which $k_1^a$, $a_1^0$, $a_1$ and $b_1$ are constants.

We also assume that the excessive stem cell population inhibits stem cell symmetric division, and that the elevated stem cell cellular damage promotes stem cell differentiation via decreasing stem cell self-renewal. As a result, we modify $\delta_1$ and $\delta_2$ in (4.3) in the same way as above

$$\delta_1(P_T, P_S, a) = \frac{\delta_1^0}{1 + (k_i^T P_T)^{m_S} + (k_i^S P_S)^{m_S}} g(a) \tag{4.7}$$

and

$$\delta_2(P_T, P_S, a) = \frac{\delta_2^0}{1 + (k_i^T P_T)^{m_S} + (k_i^S P_S)^{m_S}}, \tag{4.8}$$

where

$$g(a) = a_2 + \frac{b_2}{1 + e^{-k_2^d(a - a_2^0)}}$$

is another function of a sigmoid shape with constants $k_2^a$, $a_2^0$, $a_2$ and $b_2$.

As a continuation of §4.1, our analysis still focuses on the discussion of segregation rules. In addition to the parameters in table 1, the values of more parameters are given in table 2, for which the detailed reasoning can be found in appendix A.3.2.

To examine the change in population dynamics after introducing feedbacks from stem cells, the same combinations of segregation rules as in §4.1 are considered: $\alpha_1$ varies from 0.1 to 0.5, $\beta_1$ varies from 0.1 to 0.5, and $\gamma_1$ varies from 0.1 to 0.9. Here, we study the effect of the feedbacks from stem cells on the following two aspects:

— the population size of TD cells and the population ratio of TD cells to stem cells at steady state; and
— the dynamics of population evolution and the damage distribution of stem cell population at steady state.

### 4.2.1. Population size of TD cells and the population ratio of TD cells to stem cells

Similar to the model in §4.1, damage distribution $\beta_i$ between TD cells does not affect the results much. Since the regulation is stronger when we consider feedbacks from both stem cells and TD cells, the stabilized population of TD cells is smaller than that in §4.1 under the same parameters. It may not be meaningful to directly compare the size of populations in different models, since we do not have

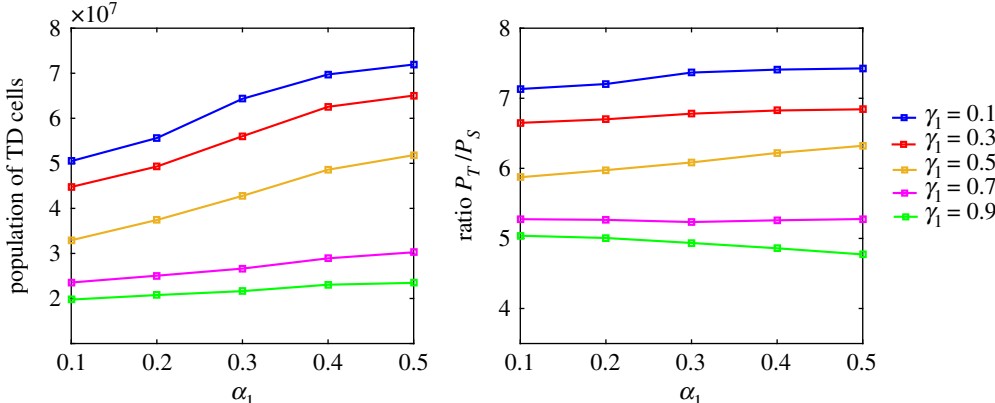

**Figure 8.** TD cell population and population ratio in steady state for different combinations of $\alpha_1$, $\beta_1$ and $\gamma_1$ for the model with feedbacks from stem cells and TD cells. The parameters used in these simulations are shown in tables 1 and 2, and $\beta_1 = 0.5$.

**Table 2.** The choices of additional parameters, estimated based on biological evidence and previous modelling works, with details in appendix A.3.2.

| parameters | meaning | value |
| --- | --- | --- |
| $k_1^S$ | regulation constant | $10^{-7}$ |
| $k_2^S$ | regulation constant | $0.25 \times 10^{-7}$ |
| $k_v^S$ | regulation constant | $0.5 \times 10^{-7}$ |
| $m_S$ | Hill exponent | 2 |
| $a_1, a_2$ | constant for sigmoid regulation | 1.1 |
| $b_1, b_2$ | constant for sigmoid regulation | $-0.7$ |
| $k_1^a, k_2^a$ | stiffness parameter | 20 |
| $a_1^0, a_2^0$ | damage threshold for sigmoid regulation | 0.75 |

real experimental data. However, similar trends are observed in the model with the feedbacks from stem cells (figure 8): for fixed $\alpha_1$, as damage retention $\gamma_1$ in stem cell increases, both the TD cell population and the population ratio $P_S/P_T$ decrease; for fixed $\gamma_1$, as damage segregation between stem cells becomes more symmetric, the TD cell population increases, but the population ratio is almost unchanged. It is worth mentioning that the population ratio increases a little bit for each damage segregation configuration, after adding feedbacks from stem cells. This means that the regulation is more efficient, since one needs a smaller stem cell pool to generate the same number of TD cells.

### 4.2.2. Dynamics of population evolution and damage distribution of stem cell population

A notable consequence of adding feedbacks from stem cells is the reduction of the oscillations in population dynamics. Comparing figure 7a with figure 9a, after considering the feedbacks from stem cells, the oscillations in population dynamics disappear even when $\gamma_1$ is large, and that the population overshoot problem resolves when $\alpha_1$ is small.

Since the feedbacks from stem cells described by (4.7) and (4.6) also depend on stem cell cellular damage level, we want to examine the effect of such regulations on damage distribution at the end of cell cycle by comparing the models with/without the feedbacks from stem cells (figures 7b and 9b). Similar to §4.1, the damage distributions at the end of cell cycle become more concentrated, as the degree of symmetry in damage segregation increases. However, compared to the results in §4.1 (figure 7b), the model with the feedback from stem cells (figure 9b) provides better fitness, since under the same segregation rules, there are more cells with less damage at the end of cell cycle at steady state. This effect becomes most significant when the segregation rule is symmetric. This result suggests that slowing down the cell cycle progression of stem cells with a high level of damage and promoting such stem cells to differentiate can indeed improve the overall health of the stem cell population.

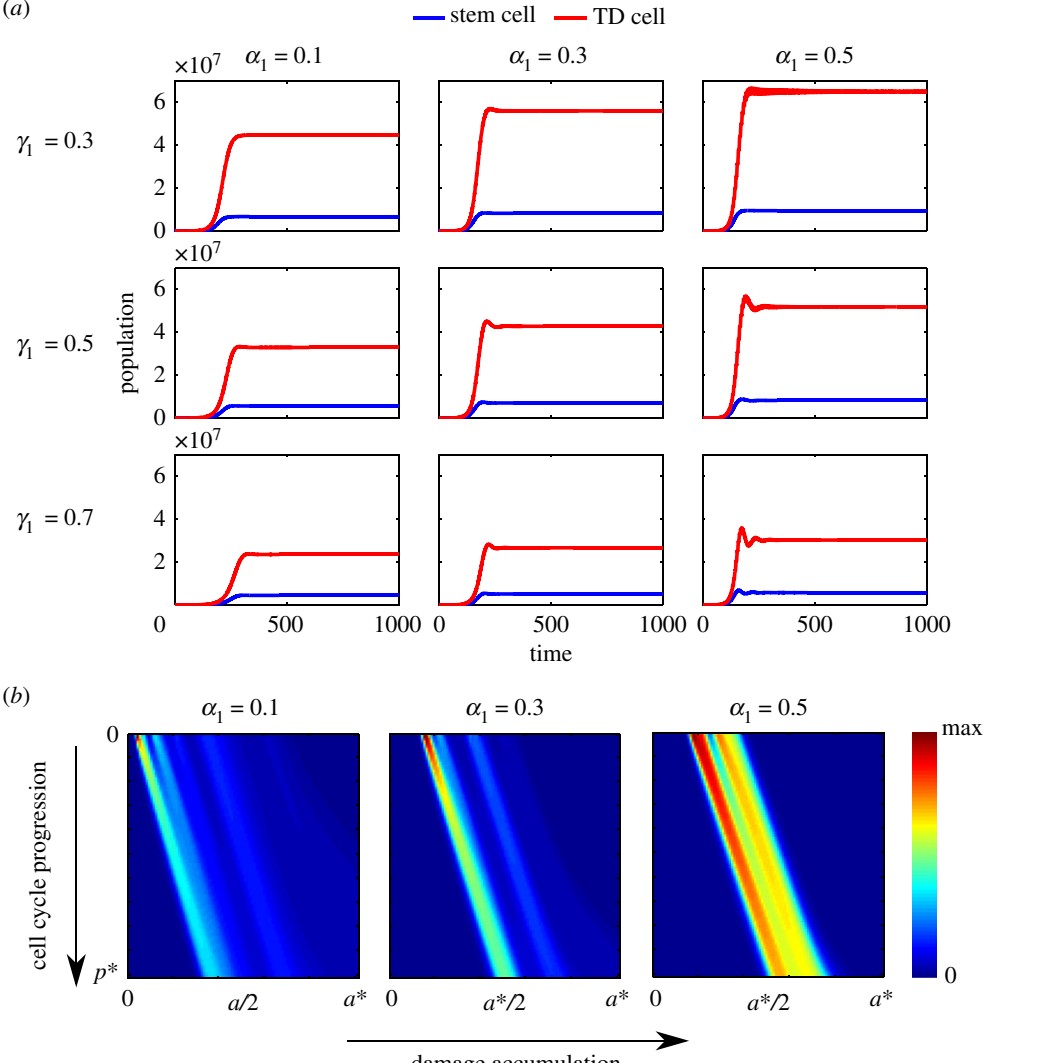

**Figure 9.** Dynamics of population evolution and damage distribution of stem cell population at steady state for the model with feedbacks from stem cells and TD cells under different segregation rules. (*a*) Sample population dynamics under different combinations of $\alpha_1$ and $\gamma_1$. The red curves stand for TD cell population and the blue curves are for stem cell population. (*b*) Population density and damage distribution for stem cells. In these three cases, we set $\gamma_1 = 0.3$. The other parameters used in these simulations are listed in tables 1 and 2, and $\beta_1 = 0.5$.

# 5. Conclusion

In this research, a novel model was developed to integrate stem cell proliferation and differentiation with damage accumulation in stem cell ageing process. A system of two structured PDEs are used to model stem cells (including all multiple progenitors) and TD cells. In our model, cell cycle progression and damage accumulation are continuous while division is discrete, and the damage segregation takes place at each division. Ageing effect is included through the inhibition from damage accumulation on stem cell proliferation and self-renewal. Regulations from TD cell and stem cell populations are incorporated through negative feedbacks on stem cell proliferation and symmetric division.

Our results showed that more equal distribution of damage between stem cells in symmetric renewal and less damage retention in stem cell in asymmetric division are still favourable segregation rules resulting in higher population size and greater population ratio; asymmetric damage segregation in stem cells leads to less concentrated damage distribution in stem cells population, which may be more robust to sudden increase in damage. These two results provide an answer for why some types of stem cells can switch between symmetric and asymmetric divisions in response to external and internal regulations [1,3].

Other than the feedbacks solely from TD cells, adding feedback regulations from stem cells can reduce oscillations and population overshoot in some situations with unfavourable damage

segregation rules, say, in the situation where $\alpha_1$ is small and $\gamma_1$ is large. Moreover, the stem cell feedback regulation can slow down the proliferation of stem cells with high level of damage to increase their tendency to differentiate. Overall, the feedback regulation system can improve the fitness of stem cells by increasing the percentage of stem cells with low level of damage. This result provides a new aspect to understand the role of the self-regulations in stem cell lineage [32,47,52,61].

In this study, our model provides a framework to study how the damage segregation rules at division affect the population dynamics. Moreover, our model is an excellent tool to understand the data which may result from observing the dynamics of regeneration with damage distribution in stem cell population after a knock-out process. In the future work, we can extend our study to the system with more than two stages, including progenitor cells [40,61] to get a better understanding for the system of stem cell lineage.

Data accessibility. This article has no additional data. The code for the numerical method can be found in the electronic supplementary material.

Authors' contributions. All authors were involved in the design of the model and the analysis of the numerical data; Y.W. and W.-C.L. carried out the numerical tests, and drafted the manuscript; C.-S.C. critically revised the manuscript. All authors gave final approval for publication and agree to be held accountable for the work performed therein.

Competing interests. We declare we have no competing interests.

Funding. C.-S.C. is supported by NSF DMS 1813071 and DMS 1253481. W.-C.L. is supported by the General Research Fund of the Research Grants Council of the Hong Kong Special Administrative Region (project no. CityU 11303117) and CityU Strategic Research Grant (project no. CityU 11300518).

Acknowledgements. W.-C.L. thanks for the support of Mathematical Biosciences Institute during his visit at Ohio State University.

# Appendix A

## A.1. Derivation of the limit damage band

Suppose a stem cell currently has damage level $a_0$ in the beginning of a cell cycle. We want to record the damage in the following cycles. At the end of the first cycle, the damage accumulates to $a_0 + p^*/v_p$. After division, a stem daughter cell inherits damage $(a_0 + p^*/v_p)\ell_1$ with $\ell_1 \in \{\alpha_1, \alpha_2, \gamma_1\}$, whose damage gets to $(a_0 + p^*/v_p)\ell_1 + p^*/v_p = a_0\ell_1 + (p^*/v_p)(\ell_1 + 1)$ at the end of the second cycle. After division, a stem daughter cell inherits damage $(a_0\ell_1 + (p^*/v_p)(\ell_1 + 1))\ell_2$ with $\ell_2 \in \{\alpha_1, \alpha_2, \gamma_1\}$, and the damage accumulates to $(a_0\ell_1 + (p^*/v_p)(\ell_1 + 1))\ell_2 + tp^*/v_p = a_0\ell_1\ell_2 + (p^*/v_p)(\ell_1\ell_2 + \ell_2 + 1)$ at the end of the third cycle. By induction, we can easily show that the damage in a stem daughter cell (if not dead yet) at the end of the $m$-th cycle is

$$a_0 \prod_{i=0}^{m-1} \ell_i + \left(\frac{p^*}{v_p}\right) \sum_{i=0}^{m-1} \prod_{k=i}^{m-1} \ell_k \quad \text{with } \ell_0 = 1 \text{ and } \ell_i \in \{\alpha_1, \alpha_2, \gamma_1\} \text{ for } i \geq 1.$$

Let $\omega_1$ and $\omega_2$ be the minimum and the maximum of $\alpha_1, \alpha_2, \gamma_1$. In the above, considering a fixed segregation portion $\ell_i = \omega_j$ for all $i \geq 1$, we get the damage level

$$a_0 \omega_j^{m-1} + \left(\frac{p^*}{v_p}\right) \frac{1 - \omega_j^m}{1 - \omega_j}, \quad j = 1, 2.$$

The limits as $m \to \infty$ yield the limit damage band

$$\left[\frac{p^*}{v_p(1 - \omega_1)}, \frac{p^*}{v_p(1 - \omega_2)}\right].$$

## A.2. Proofs of the propositions

### A.2.1. Analytic solution

Before we prove the propositions, we first obtain the analytic solution which can provide us a tool used in the proofs. To simplify the analysis, we make the following change of variables which will be

used in the proofs:

$$\widetilde{p} = \frac{p}{v_p} \quad \text{and} \quad \widetilde{a} = \frac{a}{v_a},$$

and the domains of new variables are $0 \le \widetilde{p} \le t_p = p^*/v_p$ and $0 \le \widetilde{a} \le t_a = a^*/v_a$. Given constant cell cycle progression and damage accumulation for stem cells, the biological meaning of $t_p$ and $t_a$ can be interpreted as

— $t_p$ is the duration of one cell cycle.
— $t_a$ is the duration needed to accumulate damage to the lethal threshold from zero damage.

With the new variables, (2.1) and (2.2) become

$$\frac{\partial \widetilde{S}}{\partial t} + \frac{\partial \widetilde{S}}{\partial \widetilde{p}} + \frac{\partial \widetilde{S}}{\partial \widetilde{a}} = 0 \tag{A 1}$$

and

$$\frac{\partial \widetilde{T}}{\partial t} + \frac{u_p}{v_p}\frac{\partial \widetilde{T}}{\partial \widetilde{p}} + \frac{u_a}{v_a}\frac{\partial \widetilde{T}}{\partial \widetilde{a}} = 0, \tag{A 2}$$

where $\widetilde{S}(t, \widetilde{p}, \widetilde{a}) = S(t, p, a)$ and $\widetilde{T}(t, \widetilde{p}, \widetilde{a}) = T(t, p, a)$. For simplicity, we will drop all the tildes on the notations. The dynamics of stem cells is independent of TD cells, while the dynamics of TD cells is determined by stem cell through boundary condition (3.2). Once the behaviour of stem cell is determined, the behaviour of TD cell can be easily derived. So we mainly focus on the dynamics of stem cells.

After a change of variable, (A 1) together with the initial condition and boundary condition (3.1) can be solved by the method of characteristics. Due to the complexity of boundary conditions, no closed form of $S(t, p, a)$ can be obtained, and the solution is expressed in a recursive form.

For the first cell cycle, i.e. $0 < t < t_p$, we have three cases

(i) If $t \le p$ and $t \le a$, the solution is determined by the initial condition

$$S(t, p, a) = S(0, p - t, a - t); \tag{A 3}$$

(ii) If $t > a$ and $a < p$, the solution is determined by the boundary condition on $a = 0$

$$S(t, p, a) = S(t - a, p - a, 0); \tag{A 4}$$

(iii) If $t > p$ and $p < a$, the solution is determined by boundary condition on $p = 0$

$$S(t, p, a) = S(t - p, 0, a - p). \tag{A 5}$$

According to these three cases, $S(t, p, a)$ can be solved as

$$S(t, p, a) = \begin{cases} S(0, p - t, a - t) & \text{if } t \le p \text{ and } t \le a, \\ 0 & \text{if } t > a \text{ and } a < p, \\ H(t, p, a, \alpha_1, t_a)\dfrac{\delta_1}{\alpha_1}S\left(0, t_p - (t - p), \dfrac{a - p}{\alpha_1} - (t - p)\right) \\ +H(t, p, a, \alpha_2, t_a)\dfrac{\delta_1}{\alpha_2}S\left(0, t_p - (t - p), \dfrac{a - p}{\alpha_2} - (t - p)\right) \\ +H(t, p, a, \gamma_1, t_a)\dfrac{\delta_3}{\gamma_1}S\left(0, t_p - (t - p), \dfrac{a - p}{\gamma_1} - (t - p)\right) & \text{if } t > p \text{ and } p < a, \end{cases} \tag{A 6}$$

where the function $H$ is given by

$$H(t, p, a, \omega, t_a) = \begin{cases} 1 & \text{if } \frac{a-p}{\omega} < t_a \text{ and } \frac{a-p}{\omega} - (t - p) > 0, \\ 0 & \text{otherwise.} \end{cases} \tag{A 7}$$

For the time beyond the first cell cycle, i.e. $t > t_p$, by tracing back the process of damage accumulation by

one cycle, $S(t, p, a)$ is solved as

$$S(t, p, a) = H(t_p, p, a, \alpha_1, t_a)\frac{\delta_1}{\alpha_1}S\left(t - t_p, p, \frac{a-p}{\alpha_1} - (t_p - p)\right)$$

$$+ H(t_p, p, a, \alpha_2, t_a)\frac{\delta_1}{\alpha_2}S\left(t - t_p, p, \frac{a-p}{\alpha_2} - (t_p - p)\right) \tag{A 8}$$

$$+ H(t_p, p, a, \gamma_1, t_a)\frac{\delta_3}{\gamma_1}S\left(t - t_p, p, \frac{a-p}{\gamma_1} - (t_p - p)\right).$$

Applying (A 8) iteratively and eventually (A 6), we can obtain the density of the stem cell population at any time.

## A.2.2. Proof of proposition (3.1)

First, we suppose $f_r < 1/2$. Consider any fixed time $t \gg t_p$. By (A 8), we have

$$0 \le S(t + t_p, p, a) \le \frac{\delta_1}{\alpha_1}S\left(t, p, \frac{a-p}{\alpha_1} - (t_p - p)\right)$$

$$+ \frac{\delta_1}{\alpha_2}S\left(t, p, \frac{a-p}{\alpha_2} - (t_p - p)\right)$$

$$+ \frac{\delta_3}{\gamma_1}S\left(t, p, \frac{a-p}{\gamma_1} - (t_p - p)\right).$$

Then by integration and substitutions

$$0 \le P_S(t + t_p) = \int_0^{t_p}\int_0^{t_a} S(t + t_p, p, a)\,\mathrm{d}a\,\mathrm{d}p$$

$$\le (\delta_1 + \delta_1 + \delta_3)\int_0^{t_p}\int_0^{t_a} S(t, p, a)\,\mathrm{d}a\,\mathrm{d}p$$

$$= (2f_r)P_S(t).$$

By the continuity of $P_S(t)$, it attains a maximum $P_{\max}$ in a time interval $[t_0, t_0 + t_p]$ for a fixed $t_0$. Repeated application of the above inequality, in view of the arbitrariness of $t$, implies that for any $n \ge 1$, and for any $t$ in $[t_0 + n\,t_p, t_0 + (n+1)t_p]$, we have

$$P_S(t) \le (2f_r)^n P_{\max},$$

which clearly establishes that $\lim_{t\to\infty} P_S(t) = 0$.

For the second part, when the limit damage band exceeds the lethal threshold, i.e. $t_a < t_p/(1 - \omega_1)$, we let $\varepsilon = t_p/(1 - \omega_1) - t_a > 0$. If we assume that $(a - p)/\omega < t_a$, where $\omega \in \{\alpha_1, \alpha_2, \gamma_1\}$, we have

$$a < \omega t_a + p,$$

$$a < \omega\left(\frac{t_p}{1 - w} - \varepsilon\right) + p$$

$$\text{and} \quad a > \frac{a-p}{\omega} - (t_p - p) + C,$$

where $C = (1 - \omega_2)\varepsilon > 0$ which is independent of $a$ and $p$. Suppose that $t \ll t_p$, by (A 8), we have

$$S(t, p, a) = H(t_p, p, a, \alpha_1, t_a)\frac{\delta_1}{\alpha_1}S\left(t - t_p, p, g(a, \alpha_1)\right)$$

$$+ H(t_p, p, a, \alpha_2, t_a)\frac{\delta_1}{\alpha_2}S\left(t - t_p, p, g(a, \alpha_2)\right)$$

$$+ H(t_p, p, a, \gamma_1, t_a)\frac{\delta_3}{\gamma_1}S\left(t - t_p, p, g(a, \gamma_1)\right),$$

where $g(a, \omega) = (a - p)/\omega - (t_p - p)$. For each term in the right-hand side, if $(a - p)/\omega \ge t_a$, $H$ is zero; if $(a - p)/\omega < t_a$, we have $g(a, \omega) < a - C$. When $g(a, \omega)$ is less than or equal to zero, the term $S(t - t_p, p, g(a, \omega)) = 0$; when $g(a, \omega)$ is larger than zero, we can repeat the process and express the solution at $t$ in terms of the solutions at $t - nt_p$ where $n$ is a positive integer. Because $g(a, \omega) < a - C$, the damage level will be reduced by a constant value in each process. Hence, with finite value $n$, all the terms $S(t - nt_p, p, g)$ will become zero as $g$ is less than zero.

Intuitively, the stem cell population always diminishes to zero, since after sufficiently long time the offspring of stem cells with initial zero damage, even in the slowest way of accumulating damage, i.e. $\ell_i = \omega_1$ for every $i$, would gain damage to reach $t_a$ and die, and other offspring in the evolution gain more damage and would die sooner. This result is true, no matter what the proportions of three division types are.

## A.2.3. Proof of proposition (3.2)

By the assumption that $t_a > t_p/(1 - \omega_2)$, no cell will die due to damage accumulation and $f_r$ completely determines the evolution of stem cell population. If $f_r > 1/2$, the portion of generated stem cells through divisions is greater than 1, which means that the stem cell population is increasing to infinity. Indeed, for any $t$, let $\omega \in \{\alpha_1, \alpha_2, \gamma_1\}$, we define

$$I(\omega) := \frac{1}{\omega} \int_0^{t_p} \int_0^{\omega_2 t_a + p} H(t_p, p, a, \omega, t_a) S\left(t - t_p, p, \frac{a - p}{\omega} - (t_p - p)\right) \mathrm{d}a\, \mathrm{d}p.$$

By the definition of $H$ and change of variables with $\omega \le \omega_1$ and $t_a > t_p/(1 - \omega_2)$,

$$I(\omega) \ge \frac{1}{\omega} \int_0^{t_p} \int_{\omega t_p - \omega p + p}^{\omega t_a + p} S\left(t - t_p, p, \frac{a - p}{\omega} - (t_p - p)\right) \mathrm{d}a\, \mathrm{d}p$$

$$= \int_0^{t_p} \int_0^{t_a - t_p + p} S\left(t - t_p, p, a\right) \mathrm{d}a\, \mathrm{d}p \qquad \text{(A 9)}$$

$$\ge \int_0^{t_p} \int_0^{\omega_2 t_a + p} S\left(t - t_p, p, a\right) \mathrm{d}a\, \mathrm{d}p.$$

By (A 8) and (A 9),

$$P_S(t) \ge \int_0^{t_p} \int_0^{\omega_2 t_a + p} S\left(t, p, a\right) \mathrm{d}a\, \mathrm{d}p$$

$$= \delta_1 I(\alpha_1) + \delta_1 I(\alpha_2) + \delta_3 I(\gamma_1)$$

$$\ge 2 f_r \int_0^{t_p} \int_0^{\omega_2 t_a + p} S\left(t - t_p, p, a\right) \mathrm{d}a\, \mathrm{d}p.$$

Repeating this procedure, if $t = n t_p + \tilde{t}$ and $\tilde{t} \in [0, t_p)$, then we have

$$P_S(t) \ge (2 f_r)^n \int_0^{t_p} \int_0^{t_a - t_p + p} S\left(\tilde{t}, p, a\right) \mathrm{d}a\, \mathrm{d}p.$$

By (A 6), we can have

$$\int_0^{t_p} \int_0^{t_a - t_p + p} S\left(\tilde{t}, p, a\right) \mathrm{d}a\, \mathrm{d}p \ge \int_0^{t_p} \int_0^{t_a - t_p + p} S\left(0, p, a\right) \mathrm{d}a\, \mathrm{d}p.$$

With the assumptions $f_r > 1/2$ and

$$\int_0^{t_p} \int_0^{t_a - t_p + p} S\left(0, p, a\right) \mathrm{d}a\, \mathrm{d}p > 0.$$

Now we proved that the population will blow up when $n \to \infty$.

Intuitively, after sufficiently long time, there is no death by damage accumulation since $t_a \ge t_p/(1 - \omega_2)$. Hence in the situation that $f_r = 1/2$, we can use a similar method to show that one of the offspring after stem cell division remains a stem cell and the stem cell population is conserved.

## A.2.4. Proof of proposition (3.3)

We assume that (3.6) holds

$$f_r > \frac{1}{2} \quad \text{and} \quad \frac{t_p}{1 - \gamma_1} \le \frac{t_p}{1 - \alpha_1} < t_a < \frac{t_p}{1 - \alpha_2}.$$

We first prove (a). For integers $m \geq 1$, define

$$n(k) = \min\left\{ m \in \mathbb{N} : t_a < \frac{1 - \gamma_1^k}{1 - \gamma_1} t_p \alpha_2^m + \frac{1 - \alpha_2^m}{1 - \alpha_2} t_p \right\}.$$

Let $P_S(kt_p)$ denote the stem cell population at the end of the $k$-th cycle, when the damage levels in all cells are larger than $a_k = (1 - \gamma_1^k)/(1 - \gamma_1)t_p$, from the discussion in A1. The damage level in stem daughter cells (if not dead yet) after $n(k)$ more cycles would be greater than

$$a_k \prod_{i=0}^{n-1} \ell_i + t_p \sum_{i=0}^{n-1} \prod_{k=i}^{n-1} \ell_k \quad \text{with } \ell_0 = 1 \text{ and } \ell_i \in \{\alpha_1, \alpha_2, \gamma_1\} \text{ for } i \geq 1, \tag{A 10}$$

where $n = n(k)$. If we set $\ell_i = \alpha_2$ for all $0 \leq i \leq n(k)$, the definition of $n(k)$ can imply that the damage level in (A 10) is larger than $t_a$ at the end of the $(k + n(k))$-th cycle, and the daughter cells in this situation will die. Thus, for the possible situations for the cell to survive at the end of the $(k + n(k))$-th cycle, there should be at least one $\ell_i = \gamma_1$ or $\alpha_1$ for $1 \leq i \leq n(k)$. Clearly, the proportion of such stem daughter cells in the population is

$$(2f_r)^{n(k)} - \delta_1^{n(k)}.$$

Suppose there exists a combination of $m$ and $\delta_i$ such that $(2f_r)^{n(k)} - \delta_1^{n(k)} < 1$. By the similar method used in A.2.2, we have

$$P_S((k + n(k))t_p) \leq P_S(kt_p)((2f_r)^{n(k)} - \delta_1^{n(k)}).$$

It is easy to show that the sequence $\{(2f_r)^n - (\delta_1)^n\}$ is increasing in $n$, under the assumption $f_r > \frac{1}{2}$; the sequence $\{n(k)\}$ is non-increasing in $k$, i.e. $n(k+1) \leq n(k)$. Then applying the above inequality with $k$ replaced by $k + n(k)$, one gets

$$P_S((k + n(k) + n(k + n(k)))t_p)$$
$$\leq P_S((k + n(k))t_p)((2f_r)^{n(k+n(k))} - \delta_1^{n(k+n(k))})$$
$$\leq P_S((k + n(k))t_p)((2f_r)^{n(k)} - \delta_1^{n(k)})$$
$$\leq P_S(kt_p)((2f_r)^{n(k)} - \delta_1^{n(k)})^2.$$

Repeating this argument, we have a sequence of population approach to zero as $(2f_r)^{n(k)} - \delta_1^{n(k)} < 1$. Since $\{(2f_r)^{n(k)} - (\delta_1)^{n(k)}\}$ is non-increasing in $k$, we can just consider the condition $\{(2f_r)^{n(k)} - (\delta_1)^{n(k)}\} < 1$ when $k$ is sufficiently large. Then, we can simplify the condition $\{(2f_r)^n - (\delta_1)^n\} < 1$ with

$$n = \min\left\{ m \in \mathbb{N} : t_a < \frac{1}{1 - \gamma_1} t_p \alpha_2^m + \frac{1 - \alpha_2^m}{1 - \alpha_2} t_p \right\}.$$

Therefore, part (a) is proven. Next we work with (b). Similar to the method used in the proof of A.2.3 with the assumption

$$\frac{p^*}{v_p(1 - \gamma_1)} \leq \frac{p^*}{v_p(1 - \alpha_1)} < t_a,$$

we obtain

$$P_S(t) \geq \int_0^{t_p} \int_0^{\omega_2 t_a + p} S\left(t, p, a\right) \mathrm{d}a \, \mathrm{d}p$$
$$\geq \delta_1 I(\alpha_1) + \delta_3 I(\gamma_1)$$
$$\geq (\delta_1 + \delta_3) \int_0^{t_p} \int_0^{\omega_2 t_a + p} S\left(t - t_p, p, a\right) \mathrm{d}a \, \mathrm{d}p.$$

When $\delta_2 = 0$, $\delta_1 + \delta_3 = 1$ and

$$P_S(t) \geq \int_0^{t_p} \int_0^{\omega_2 t_a + p} S\left(t - t_p, p, a\right) \mathrm{d}a \, \mathrm{d}p.$$

By repeating this procedure, we have

$$P_S(t) \geq \int_0^{t_p} \int_0^{t_a - t_p + p} S\left(0, p, a\right) \, da \, dp > 0$$

which implies that the stem cell population blows up to infinity, or is bounded below to avoid stem cell extinction.

## A.3. Parameter estimation

### A.3.1. Parameters in the model with feedback from TD cells

Although there is no concrete biological data in stem cell division types or specific regulation mechanisms to our knowledge, we discuss the selection of parameters in table 1 referring to both the biological observations [43–46] and the previous mathematical models [25–27,32,34–40,54,55]. For the simulations we considered here, we assume that damage segregates symmetrically between stem cells and between TD cells, i.e. $\alpha_1 = 0.5$, $\beta_1 = 0.5$, while the damage retention in asymmetric division is low, i.e. $\gamma_1 = 0.25$.

*Cell cycle progression speed and damage accumulation speeds $v_p$, $v_a$ and $u_a$.* It is generally accepted that the population of mitotic cells is much smaller than that of TD cells, although it is very difficult to identify and count stem cells and progenitors [57–60]. Due to lack of experimental data, we refer to some existing modelling works for a reasonable range for the ratio of mitotic cells to TD cells. In the simulations of [37,55], populations of post-mitotic cells are about 4–10-fold that of mitotic cells.

On the other hand, mitotic cells are assumed to have a smaller death rate than post-mitotic cells [7]. However, there is no precise description of death rates due to two reasons: the death rate of stem cells is not observable, and the death rate of TD cells is tissue specific. In various modelling works, the assumed death rate of TD cells ranges from $10^{-1}$ to $10^{-4}$ of the total population [25,39]. Analysis and simulations in [25,39,40] suggest that a large death rate of TD cells results in instability of population. In our model, the death rates are measured in the following way:

$$r_S = \frac{\int_0^{p^*} v_a S(t, p, a^*) \, dp}{P_S} \tag{A 11}$$

and

$$r_T = \frac{\int_0^\infty u_a T(t, p, a^c) \, dp}{P_T}, \tag{A 12}$$

where the numerators are population out-fluxes due to death and the denominators are the total populations. By properly choosing $v_p$, $v_a$, $u_a$, we wish to achieve that $P_T/P_S$ is no less than 5, $r_T$ is about $10^{-4}$, and $r_S$ is reasonably small, compared to $r_T$. Here, we take $v_p = 0.2$, $v_a = 0.06$ and $u_a = 0.02$.

*Maximum fractions of three types of cell division $\delta_1^0$, $\delta_2^0$ and $\delta_3^0$.* It was observed that stem cells mainly undergo asymmetric renewal and differentiation when stem cell pool is expanding or when TD cells suffer from great loss, and enter into an inactively dividing stage and then mainly undergo asymmetric division at steady state [43–46]. Since the feedbacks we consider are negative regulations, $\delta_1^0$ provides the maximum fraction of symmetric renewal. By the property of the Hill function, $\delta_1(P_T)$ can switch between large and small values in response to small and large populations of TD cells. But we observed that as we increase the initial $\delta_1^0$, population overshoot and oscillations appear before convergence (figure 10a). Moreover, the population ratio $P_T/P_S$ decreases (from 5.664 to 3.622) and death rates of stem cell $r_S$ increases (from $1.191 \times 10^{-7}$ to $1.365 \times 10^{-4}$) as $\delta_1^0$ increases from 0.6 to 0.8. Hence, we take $\delta_1^0 = 0.6$, $\delta_2^0 = 0.3$ and $\delta_3^0 = 0.1$.

*Regulation constants $k_1^T$, $k_2^T$ and $k_v^T$.* Many works suggest that regulation parameters $k_i^T$ and $k_v^T$ are closely related to the TD cell population at steady state [26,27,35,39]. These modelling works mainly focus on the blood system, which is one of the most well-studied systems and serves as a paradigm for understanding stem cells. The regulation parameter $k$ in those models ranges from $10^{-7}$ to $10^{-10}$, which corresponds to red blood cell population of size $10^7 \sim 10^{10}$. However, some other works that do not specify the type of stem cells may choose some other ranges. For example, [37] chose $10^{-3}$ for regulation constant and the resulting population magnitude is about $10^3$. In our simulations, we choose $10^{-8}$ as the scale for the magnitudes of $k_i^T$ and $k_v^T$.

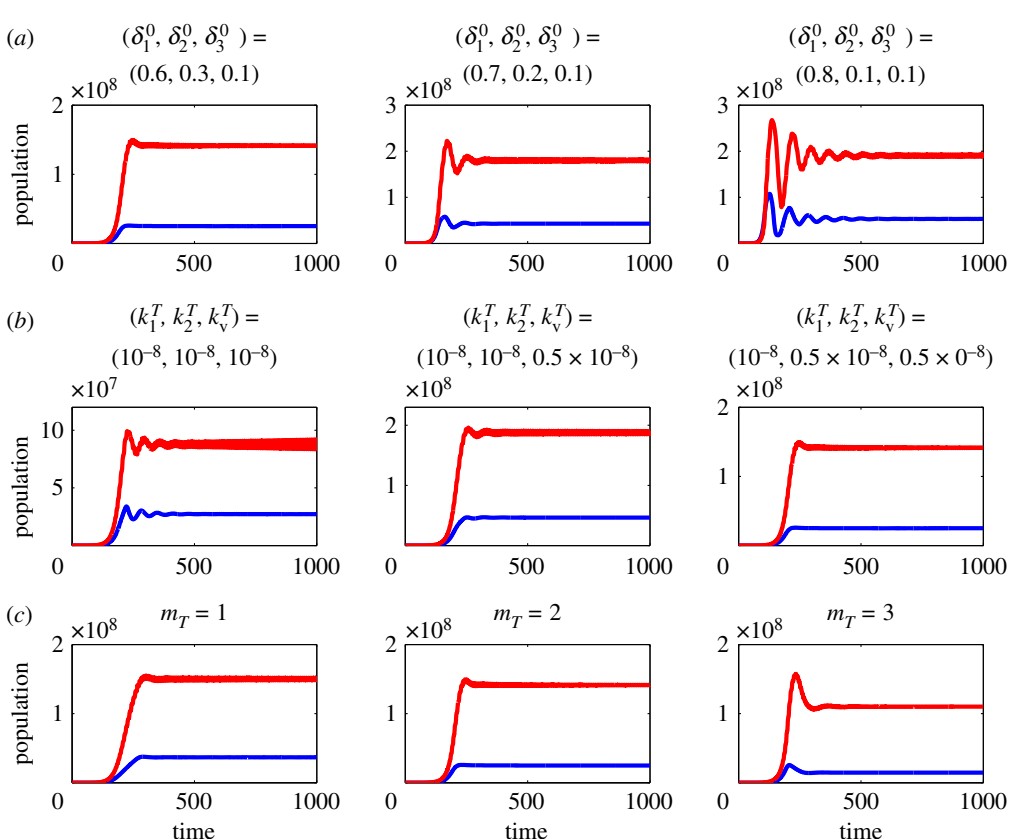

**Figure 10.** Simulations for the model with feedback from TD cells for different settings of parameters. (a) Comparison of different initial division fractions. (b) Comparison of different regulation parameters. (c) Comparison of Hill exponents. The other parameters which are not listed in the figures are $k_1^T = 10^{-8}$, $k_2^T = 0.5 \times 10^{-8}$, $k_v^T = 0.5 \times 10^{-8}$, $\alpha_1 = 0.5$, $\beta_1 = 0.5$, $\gamma_1 = 0.25$, $v_p = 0.2$, $v_a = 0.06$, $u_a = 0.02$, $m_T = 2$, $\delta_1^0 = 0.6$, $\delta_2^0 = 0.3$, $\delta_3^0 = 0.1$.

On the other hand, by choosing appropriate regulation parameters $k_1^T$, $k_2^T$ and $k_v^T$ in equations (4.3) and (4.4), asymmetrical division will predominate in three types of division of stem cells at steady state. In our simulations shown in figure 10b, we found that equal strengths of feedback on stem cell symmetric renewal $\delta_1$ and cell cycle progression $V_p$ would result in oscillations. Stronger regulation on $V_p$ relative to $\delta_1$ is needed to obtain stabilized population evolution (figure 10b). We also found that the relative strengths of regulation on symmetric renewal $\delta_1$ and symmetric differentiation $\delta_2$ play a role in determining the death rate of stem cells ($r_S = 2.408 \times 10^{-4}$ when $(k_1^T, k_2^T, k_v^T) = (10^{-8}, 10^{-8}, 0.5 \times 10^{-8})$; $r_S = 1.191 \times 10^{-7}$ when $(k_1^T, k_2^T, k_v^T) = (10^{-8}, 0.5 \times 10^{-8}, 0.5 \times 10^{-8})$). To obtain a reasonably small death rate of stem cells, stronger regulation on $\delta_2$ is also needed. Hence, we take $k_1^T = 10^{-8}$, $k_2^T = 0.5 \times 10^{-8}$ and $k_v^T = 0.5 \times 10^{-8}$.

*Hill exponent for feedbacks $m_T$.* The modelling work [40] suggests that a low Hill exponent in the feedback is needed to avoid oscillations, where the authors observed unstable steady states for Hill exponent $m \geq 3$ through direct simulations in their ODE models. We also found similar phenomena in our model, where oscillations appear in the simulations with $m_T > 4$. Thus we need $m_T \in \{1, 2, 3\}$. To make the best estimation of $m_T$, we also need to consider all biological observations. Our simulations show that when $m_T = 1$ the death rate of stem cell is only about one third of that of TD cells ($r_S = 9.586 \times 10^{-5}$ when $m_T = 1$; $r_S = 1.191 \times 10^{-7}$ when $m_T = 2$), while when $m_T = 3$ there is population overshoot before convergence to steady state although the death rate of stem cell decreases to $10^{-15}$ (figure 10c). Hence we choose $m_T = 2$ in our model.

After a large number of trials, we arrive at the best choice of the parameters in table 1 with consideration of all biological and mathematical reasons. Under these parameters, the resulting death rates are $r_S = 1.191 \times 10^{-7}$ and $r_T = 2.357 \times 10^{-4}$. The populations at steady state are $P_S = 2.496 \times 10^7$, $P_T = 1.414 \times 10^8$ with the ratio $P_T/P_S = 5.664$. But we need to point out that all our assumptions are pieced together through different biological and mathematical research, more precise estimations of parameters need real experimental data.

## A.3.2. Parameters in the model with feedbacks from TD cells and stem cells

Inspired by the analysis in appendix A.3.1, we selected the most reasonable parameters in table 2, based on both biological and mathematical reasons as follows.

*Regulation constants $k_1^S$, $k_2^S$ and $k_v^S$, and Hill exponent for feedbacks $m_S$.* Similar to appendix A.3.1, due to the consideration of reasonable population ratio and death rate of stem cells, we still assume that the feedbacks from stem cell population on cell cycle progression $V_p$ and symmetric differentiation $\delta_2$ are stronger than that on symmetric renewal $\delta_1$. That is, we intentionally choose smaller $k_v^S$, $k_2^S$, compared to $k_1^S$. And the Hill exponent $m_S$ is chosen to be 2.

To determine the magnitude of $k_1^S$, we simulate the population evolution with $k_1^S$ ranging from $10^{-6}$ to $10^{-8}$. We found that the relative magnitude of $k_1^T$ and $k_1^S$ has little influence on the population ratio. To effectively model the regulation from the stem cell population, $10^{-7}$ turns out to be the best choice, according to our simulations under the choice of parameters in table 1. Hence, we take $k_1^S = 10^{-7}$, $k_2^S = 0.25 \times 10^{-7}$ and $k_v^S = 0.5 \times 10^{-7}$.

*Constants for sigmoid regulation.* According to experiments, mitotic cells inheriting more damage protein aggregates have an increased cell cycle length and tendency to differentiate [13,18]. To model these observations, we assume that the multipliers $f(a)$ and $g(a)$ decrease in $a$ and have sigmoid shape. Parameters $a_i$ and $b_i$ in the multipliers determine their maximum and minimum strengths, $a_i^0$ locates the threshold in damage of the transition and $k_i^a$ controls the stiffness of the transition. Due to lack of experimental data or previous modelling works to refer to, we assume $f(a)$ and $g(a)$ take the same form and range between [0.4, 1.1] with a transition at $a_1^0 = 0.75$. Hence, we take $a_1 = a_2 = 1.1$, $b_1 = b_2 = -0.7$, $k_1^a = k_2^a = 20$ and $a_1^0 = 0.75$.

With a large number of simulations, we present the best choice of the parameters in table 2 and conclude that this set of parameters is our best estimation. Under these parameters, the resulting death rates are $r_S = 2.560 \times 10^{-7}$, $r_T = 2.314 \times 10^{-4}$ for stem cells and TD cells, respectively. The populations at steady state are $P_S = 9.593 \times 10^6$, $P_T = 6.709 \times 10^7$ with the ratio $P_T/P_S = 6.994$.

## A.4. Estimation of populations in the models with feedbacks from TD cells

First, we recall the assumptions: $a^* \leq a^c$, $V_a = v_a$, $U_p = u_p$ and $U_a = u_a$ are constants, and $\delta_i$ and $V_p$ have feedbacks from TD cells described by (4.3) and (4.4). The populations of stem and TD cells at time $t$ are give by

$$P_S(t) = \int_0^{a^*} \int_0^{p^*} S(t, p, a) \, dp \, da \tag{A 13}$$

and

$$P_T(t) = \int_0^{a^c} \int_0^{\infty} T(t, p, a) \, dp \, da. \tag{A 14}$$

Under the above assumptions, we integrate (2.1) and (2.2) in $p$ and $a$ to get

$$\frac{dP_S}{dt} = -v_a \int_0^{p^*} S(t, p, a^*) - S(t, p, 0) \, dp - V_p(P_T) \int_0^{a^*} S(t, p^*, a) - S(t, 0, a) \, da \tag{A 15}$$

and

$$\frac{dP_T}{dt} = -u_a \int_0^{\infty} T(t, p, a^c) - T(t, p, 0) \, dp + u_p \int_0^{a^c} T(t, 0, a) \, da. \tag{A 16}$$

By boundary conditions (2.5), (2.6), (3.1) and (3.2), we rewrite the above equations as

$$\frac{dP_S}{dt} = -v_a \int_0^{p^*} S(t, p, a^*) \, dp - V_p(P_T) \left[ \int_0^{a^*} S(t, p^*, a) \, da - \frac{\delta_1(P_T)}{\alpha_1} \int_0^{a^*} S\left(t, p^*, \frac{a}{\alpha_1}\right) da \right.$$

$$\left. - \frac{\delta_1(P_T)}{\alpha_2} \int_0^{a^*} S\left(t, p^*, \frac{a}{\alpha_2}\right) da - \frac{\delta_3(P_T)}{\gamma_1} \int_0^{a^*} S\left(t, p^*, \frac{a}{\gamma_1}\right) da \right]$$

and

$$\frac{dP_T}{dt} = -u_a \int_0^\infty T(t, p, a^c) \, dp + V_p(P_T) \left[ \frac{\delta_2(P_T)}{\beta_1} \int_0^{a^c} S\left(t, p^*, \frac{a}{\beta_1}\right) da \right.$$
$$\left. + \frac{\delta_2(P_T)}{\beta_2} \int_0^{a^c} S\left(t, p^*, \frac{a}{\beta_2}\right) da + \frac{\delta_3(P_T)}{\gamma_2} \int_0^{a^c} S\left(t, p^*, \frac{a}{\gamma_2}\right) da \right].$$

Making a change of variable in $a$ and noticing that $a^* \leq a^c$, we obtain a system of ODEs, called system I:

$$\frac{dP_S}{dt} = -v_a \int_0^{p^*} S(t, p, a^*) \, dp + V_p(P_T)(\delta_1(P_T) - \delta_2(P_T)) \int_0^{a^*} S(t, p^*, a) \, da \qquad (A\ 17)$$

and

$$\frac{dP_T}{dt} = -u_a \int_0^\infty T(t, p, a^c) \, dp + V_p(P_T)(1 + \delta_2(P_T) - \delta_1(P_T)) \int_0^{a^*} S(t, p^*, a) \, da. \qquad (A\ 18)$$

Consider a related system II given by the following equation and (A 18):

$$\frac{dP_S}{dt} = V_p(P_T)(\delta_1(P_T) - \delta_2(P_T)) \int_0^{a^*} S(t, p^*, a) \, da. \qquad (A\ 19)$$

Then under the same initial condition, the solution to system I is not larger than that to system II, by comparison theorems for ODEs.

Now we assume that both systems I and II have steady-state solutions, denoted by $(\bar{P}_S, \bar{P}_T)$ and $(\tilde{P}_S, \tilde{P}_T)$, respectively. Then $\tilde{P}_T$ can be obtained by solving

$$\delta_1(P_T) - \delta_2(P_T) = 0.$$

Assuming appropriate values of parameters to guarantee solvability of the last equation, we have the upper-bound estimate

$$\bar{P}_T \leq \tilde{P}_T = \left( \frac{\delta_1^0 - \delta_2^0}{\delta_2^0 (k_1^T)^{m_T} - \delta_1^0 (k_2^T)^{m_T}} \right)^{1/m_T}. \qquad (A\ 20)$$

Next, we continue to estimate $\bar{P}_S$. First, we observe that in the steady state both $S(t, p, a)$ and $T(t, p, a)$ are constant along characteristic curves and we refer the reader to figure 11 for an illustration of the situation. Since in system II, there is no outflux of stem cells on the boundary $a = a^*$, we have

$$\tilde{P}_S = p^* \int_0^{a^*} S(t, p^*, a) \, da.$$

On the other hand, by the intermediate value theorem for integrals, we have, for some $h \in [h_1, h_2]$ that

$$\tilde{P}_T = h \int_0^\infty T(t, p, a^c) \, dp,$$

where

$$h_1 = a^c - \frac{\eta_2}{1 - \omega_2} \Delta a \quad \text{and} \quad h_2 = a^c - \frac{\eta_1}{1 - \omega_1} \Delta a$$

with $\eta_1 = \min\{\beta_1, \beta_2, \gamma_2\}$, $\eta_2 = \max\{\beta_1, \beta_2, \gamma_2\}$ and $\Delta a = (p^*/V_p(\bar{P}_T))v_a$. Using that $\tilde{P}_T$ is a steady-state population, we have by (A 17)

$$u_a \tilde{P}_T = h V_p(\tilde{P}_T) \int_0^{a^*} S(t, p^*, a) \, da = \frac{h V_p(\tilde{P}_T) \tilde{P}_S}{p^*}$$

and thus the estimate

$$\bar{P}_S \leq \tilde{P}_S = \frac{u_a p^*}{h V_p(\tilde{P}_T)} \tilde{P}_T \leq \frac{u_a p^*}{V_p(\bar{P}_T) h_1} \tilde{P}_T, \qquad (A\ 21)$$

where $\tilde{P}_T$ is given in (A 20).

Since in §4.1, we intentionally choose the parameters such that the death rate of stem cells is small, i.e. the outflux $v_a \int_0^{p^*} S(t, p, a^*) \, dp$ is small, the upper bound in (A 20) is very close to the real TD cell

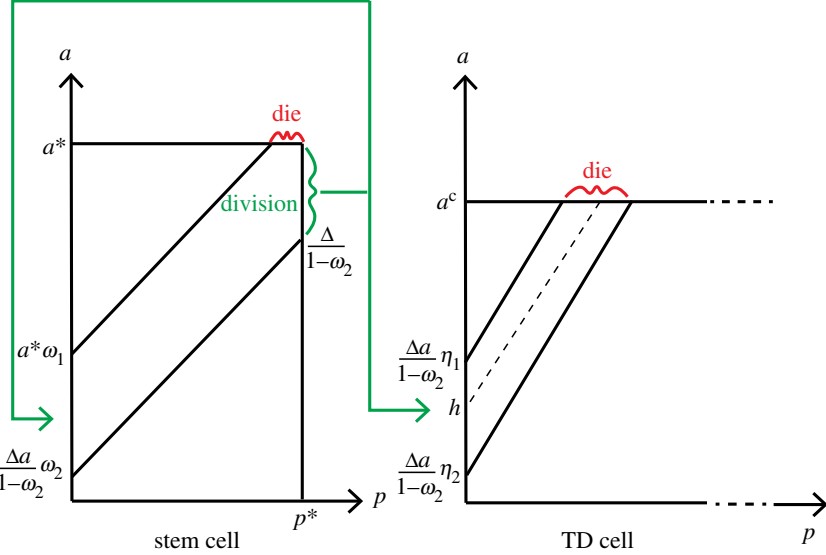

**Figure 11.** Demonstration of upper bound estimation of the population.

population in the steady state. However, the estimation for stem cells may not be sharp, due to the variation of the segregation rules. As an example, with the parameters in table 1, we have that the estimated populations are $(\bar{P}_S, \bar{P}_T) = (\frac{6}{13}\sqrt{2} \times 10^7, \sqrt{2} \times 10^8)$ and the steady-state populations in the simulation are $(\bar{P}_S, \bar{P}_T) = (2.496 \times 10^7, 1.414 \times 10^8)$.

## A.5. Numerical scheme

Consider functions $V_p$, $V_a$, $U_p$, $U_a$ dependent on $p$, $a$ only, equations (2.1) and (2.2) can be rewritten as the following system, with $W = [S, T]^t$:

$$\frac{\partial}{\partial t} W + \frac{\partial}{\partial p} F(W) + \frac{\partial}{\partial a} G(W) = 0, \tag{A 22}$$

where

$$F(W) = [f_1(W), f_2(W)]^t = [V_p S, U_p T]^t$$

and

$$G(W) = [g_1(W), g_2(W)]^t = [V_a S, U_a T]^t.$$

In the following simulation, we use third-order WENO scheme and third-order TVD Runge–Kutta time integrator.

As $U_p$, $U_a$, $V_p$, $V_a$ are all positive, the numerical approximation $W_{i,j}$ to the exact solution $W(p_i, a_j, t)$ satisfies the following ODE system:

$$\frac{\mathrm{d} W_{i,j}(t)}{\mathrm{d}t} = -\frac{\hat{F}_{i+1/2,j} - \hat{F}_{i-1/2,j}}{\Delta p} - \frac{\hat{G}_{i,j+1/2} - \hat{G}_{i,j-1/2}}{\Delta a}, \tag{A 23}$$

where $\hat{F}_{i+1/2,j}$ is called numerical flux, the design of which is the key ingredient to a successful scheme. For the third-order WENO scheme, the numerical flux $\hat{F}_{i+1/2,j}$ is defined as follows:

$$\hat{F}_{i+1/2,j} = \omega_1 \hat{F}^{(1)}_{i+1/2,j} + \omega_1 \hat{F}^{(2)}_{i+1/2,j}, \tag{A 24}$$

where $\hat{F}^{(m)}_{i+1/2,j}$ for $m = 1, 2$, are the two second-order accurate fluxes on two different stencils given by

$$\hat{F}^{(1)}_{i+1/2,j} = -\frac{1}{2}F_{i-1,j} + \frac{3}{2}F_{i,j} \quad \hat{F}^{(2)}_{i+1/2,j} = \frac{1}{2}F_{i,j} + \frac{1}{2}F_{i+1,j}. \tag{A 25}$$

The nonlinear weights $\omega_m$ are given by

$$\omega_m = \frac{\alpha_m}{\sum_{k=1}^{2} \alpha_k}, \quad m = 1, 2, \tag{A 26}$$

where

$$\alpha_k = \frac{\gamma_k}{(\varepsilon + \beta_k)^2} \quad k = 1, 2, \tag{A 27}$$

and

$$\beta_1 = (F_{i,j} - F_{i-1,j})^2 \quad \beta_2 = (F_{i+1,j} - F_{i,j})^2 \tag{A 28}$$

and

$$\gamma_1 = \frac{1}{3} \quad \text{and} \quad \gamma_2 = \frac{2}{3}. \tag{A 29}$$

The parameter $\varepsilon$ ensures that the denominator never gets to 0, and is fixed at $\varepsilon = 10^{-6}$ in the computation in this work. Similar construction can be applied to the direction of $a$.

The time concretization is implemented by a third-order TVD Runge–Kutta method

$$\left.\begin{aligned} W^{(1)} &= W^n + \Delta t L(W^n, t^n), \\ W^{(2)} &= \frac{3}{4} W^n + \frac{1}{4} W^{(1)} + \frac{1}{4} \Delta t L(W^{(1)}, t^n + \Delta t) \\ W^{n+1} &= \frac{1}{3} W^n + \frac{2}{3} W^{(2)} + \frac{2}{3} \Delta t L(W^{(2)}, t^n + \frac{1}{2}\Delta t), \end{aligned}\right\} \tag{A 30}$$

and

where $L$ denotes the r.h.s. of equation (A 23).

A CFL condition is needed for stability

$$\alpha \frac{\Delta t}{\min\{\Delta p, \Delta a\}} < 1, \tag{A 31}$$

where $\alpha = \max\{U_p, U_a, V_p, V_a\}$. In this paper, we take $\Delta p = \Delta a = 0.01$ and $\Delta t = 0.005$.

The code for the numerical method can be found in electronic supplementary material.

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
