## [Reviewer comments · Royal Society Open Science]

Review History

RSOS-191848.R0 (Original submission)

Review form: Reviewer 1

Is the manuscript scientifically sound in its present form?

Yes

Are the interpretations and conclusions justified by the results?

Yes

Is the language acceptable?

Yes

Do you have any ethical concerns with this paper?

No

Have you any concerns about statistical analyses in this paper?

No

Recommendation?

Major revision is needed (please make suggestions in comments)

Comments to the Author(s)

In their manuscript „Modeling stem cell aging: a multi-compartment continuum approach“ Wang and coworkers present a PDE model to describe damage accumulation among (dividing) stem cells and (non-dividing) terminally differentiated cells. The model comprises a third dimension in which a divisional clock is included. Stem cells divide either symmetrically or asymmetrically with respect to fate and additionally segregate damage in a tunable fashion. The authors study which parameter regimens and control loops are suitable to establish stable populations with different optimality criteria, such as minimizing cell death or damage accumulation.

The manuscript is well written and I enjoyed reading it. I am also aware that it is a challenging task to present a sophisticated mathematical model in a way that it appeals to researchers beyond the mathematical community and also attracts researchers from more biological sciences. In fact, my suggestions point precisely in this direction. For my understanding, the paper could also be published as it is, but I feel that it would be much stronger if a little more effort is made towards a biologically oriented readership.

- In the introduction I am missing a clear motivation why the authors develop and present their particular model. What is missing in other models, what is the benefit of yours? Are there questions that cannot be answered by other models but yours?
- The literature review is strangely biased. I am missing references to the works of Roeder and Loeffler that started from conceptual considerations and translated them in different mathematical frameworks. Especially a reference to Roeder, Herberg and Horn (Bull Math Biol. 2009) is in place as even the title is so close to the submitted manuscript.
- A few more words to motivate the boundary conditions on page 5 would be helpful.
- I am missing biological interpretations of the results at several points throughout the manuscript. For example in Fig 3 the authors have marked “stable” regions. What does it mean biologically? Are they plausible?
- Page 13, line 45: strange sentence. Something missing?
- As my main critic I am missing simulations of perturbation scenarios. In fact, most of our knowledge on stem cell organization results from experiments in which the systems are challenged and return (or do not return) back to normal. In the hematopoietic case these are typically (serial) transplantations, irradiation, knock-outs or ochemotox. It would give all the modeling approaches much more meaning and potentially allow to identify crucial regulatory mechanisms, if at least some of these experiments could be mimicked. Describing a “homeostatic system” is nice, but not too exciting.
- Conclusion: The authors are missing the point to draw conclusions about the suitability of their model. Are their biological findings that favor one feedback or one regulation over the other. Is there any finding with biological relevance, something testable beyond the modeling work?

Review form: Reviewer 2

Is the manuscript scientifically sound in its present form?

Yes

Are the interpretations and conclusions justified by the results?

Yes

Is the language acceptable?

Yes

Do you have any ethical concerns with this paper?

No

Have you any concerns about statistical analyses in this paper?

No

Recommendation?

Accept with minor revision (please list in comments)

Comments to the Author(s)

In this work, the authors presented a very first mathematical model on stem cell proliferation and differentiation with an age structure. One unique feature of the model is its inclusion of a variable for molecular damage in stem cells, which then describes the aging process in stem cells. The case without feedbacks was analyzed in details with interesting analytical results, and the cases with various feedbacks that are difficult to analyze were computationally simulated. The integrated study of models, rigor analysis, and simulations represents a major advance in stem cell modeling, as aging modeling becomes an increasingly important topic. The results are clearly presented, and the paper is well written. I recommend it for publication. One minor comment is its connection with specific biology. It would make the work more significant if the model could be applied to, or at least be discussed for, a specific biological system in which the stem cell age structure is important.

Decision letter (RSOS-191848.R0)

14-Jan-2020

Dear Dr Lo,

The editors assigned to your paper ("Modeling stem cell aging: a multi-compartment continuum approach") have now received comments from reviewers. We would like you to revise your paper in accordance with the referee and Associate Editor suggestions which can be found below (not including confidential reports to the Editor). Please note this decision does not guarantee eventual acceptance.

Please submit a copy of your revised paper before 06-Feb-2020. Please note that the revision deadline will expire at 00.00am on this date. If we do not hear from you within this time then it will be assumed that the paper has been withdrawn. In exceptional circumstances, extensions may be possible if agreed with the Editorial Office in advance. We do not allow multiple rounds of revision so we urge you to make every effort to fully address all of the comments at this stage. If deemed necessary by the Editors, your manuscript will be sent back to one or more of the original reviewers for assessment. If the original reviewers are not available, we may invite new reviewers.

- Data accessibility

If you wish to submit your supporting data or code to Dryad (<http://datadryad.org/>), or modify your current submission to dryad, please use the following link:
<http://datadryad.org/submit?journalID=RSOS&manu=RSOS-191848>

- Competing interests

- Authors' contributions

- Acknowledgements

- Funding statement

on behalf of Dr Andrew Angel (Associate Editor) and Mark Chaplain (Subject Editor)
 openscience@royalsociety.org

Associate Editor's comments (Dr Andrew Angel):

Both reviewers have recommended revisions to the manuscript. One recommended major revisions but these mainly relate to readability for a more biological audience and are largely optional. I am recommending major revision to allow sufficient time to make changes in line with the suggestions should you wish to do so.

Associate Editor: 2

Comments to the Author:

The study is, to the best of my knowledge, novel and scientifically sound. Moreover, the model presented could provide a solid foundation for further study. Therefore I am recommending the manuscript be sent for peer review.

Comments to Author:

Reviewers' Comments to Author:

Reviewer: 1

Comments to the Author(s)

In their manuscript „Modeling stem cell aging: a multi-compartment continuum approach“ Wang and coworkers present a PDE model to describe damage accumulation among (dividing) stem cells and (non-dividing) terminally differentiated cells. The model comprises a third dimension in which a divisional clock is included. Stem cells divide either symmetrically or asymmetrically with respect to fate and additionally segregate damage in a tunable fashion. The authors study which parameter regimens and control loops are suitable to establish stable populations with different optimality criteria, such as minimizing cell death or damage accumulation.

The manuscript is well written and I enjoyed reading it. I am also aware that it is a challenging task to present a sophisticated mathematical model in a way that it appeals to researchers beyond the mathematical community and also attracts researchers from more biological sciences. In fact, my suggestions point precisely in this direction. For my understanding, the paper could also be published as it is, but I feel that it would be much stronger if a little more effort is made towards a biologically oriented readership.

- In the introduction I am missing a clear motivation why the authors develop and present their particular model. What is missing in other models, what is the benefit of yours? Are there questions that cannot be answered by other models but yours?
- The literature review is strangely biased. I am missing references to the works of Roeder and Loeffler that started from conceptual considerations and translated them in different mathematical frameworks. Especially a reference to Roeder, Herberg and Horn (Bull Math Biol. 2009) is in place as even the title is so close to the submitted manuscript.
- A few more words to motivate the boundary conditions on page 5 would be helpful.

- I am missing biological interpretations of the results at several points throughout the manuscript. For example in Fig 3 the authors have marked “stable” regions. What does it mean biologically? Are they plausible?
- Page 13, line 45: strange sentence. Something missing?
- As my main critic I am missing simulations of perturbation scenarios. In fact, most of our knowledge on stem cell organization results from experiments in which the systems are challenged and return (or do not return) back to normal. In the hematopoietic case these are typically (serial) transplantations, irradiation, knock-outs or ochemotox. It would give all the modeling approaches much more meaning and potentially allow to identify crucial regulatory mechanisms, if at least some of these experiments could be mimicked. Describing a “homeostatic system” is nice, but not too exciting.
- Conclusion: The authors are missing the point to draw conclusions about the suitability of their model. Are their biological findings that favor one feedback or one regulation over the other. Is there any finding with biological relevance, something testable beyond the modeling work?

Reviewer: 2

Comments to the Author(s)

In this work, the authors presented a very first mathematical model on stem cell proliferation and differentiation with an age structure. One unique feature of the model is its inclusion of a variable for molecular damage in stem cells, which then describes the aging process in stem cells. The case without feedbacks was analyzed in details with interesting analytical results, and the cases with various feedbacks that are difficult to analyze were computationally simulated. The integrated study of models, rigor analysis, and simulations represents a major advance in stem cell modeling, as aging modeling becomes an increasingly important topic. The results are clearly presented, and the paper is well written. I recommend it for publication. One minor comment is its connection with specific biology. It would make the work more significant if the model could be applied to, or at least be discussed for, a specific biological system in which the stem cell age structure is important.

Author's Response to Decision Letter for (RSOS-191848.R0)

See Appendix A.

Decision letter (RSOS-191848.R1)

10-Feb-2020

Dear Dr Lo,

It is a pleasure to accept your manuscript entitled "Modeling stem cell aging: a multi-compartment continuum approach" in its current form for publication in Royal Society Open Science.

on behalf of Dr Andrew Angel (Associate Editor) and Mark Chaplain (Subject Editor)
openscience@royalsociety.org

Associate Editor Comments to Author (Dr Andrew Angel):

As far as i can discern, the corrections have addressed the suggestions of the reviewers. I am recommending the manuscript now be accepted as is.

Appendix A

Ms. Ref. RSOS-191848

Modeling stem cell aging: a multi-compartment continuum approach

Authors: Yanli Wang, Wing-Cheong Lo, and Ching-Shan Chou

Responses to the referees

We thank the referees for their valuable comments. The paper has been carefully revised according to their comments. Here is a list of the revision in response to the referees' comments. The line number and the page/section/figure number in our answers refer to those in the revised version, unless otherwise stated. In this report, we also attach a version in which the changes are highlighted in **red**.

Reviewer #1:

The manuscript is well written and I enjoyed reading it. I am also aware that it is a challenging task to present a sophisticated mathematical model in a way that it appeals to researches beyond the mathematical community and also attracts researches from more biological sciences. In fact, my suggestions point precisely in this direction. For my understanding, the paper could also be published as it is, but I feel that it would be much stronger if a little more effort is made towards a biologically oriented readership.

Response: Thanks for the suggestion. According to the comments before, we have provided more biological explanations for the modeling part, introduction and conclusion in the revision for the biologically oriented readership.

- In the introduction I am missing a clear motivation why the authors develop and present their particular model. What is missing in other models, what is the benefit of yours? Are there questions that cannot be answered by other models but yours?

Response: Our model is the first model which integrates stem cell proliferation and differentiation with damage accumulation, feedback regulation and damage segregation in the stem cell aging process. It can help us to understand the effects of different regulation and damage segregation strategies through comparing population dynamics, stem cell fitness and damage distribution in the population. This explanation has been included in the introduction of the revision (Pages 3-4).

- The literature review is strangely biased. I am missing references to the works of Roeder and Loeffler that started from conceptual considerations and translated them in different mathematical frameworks. Especially a reference to Roeder, Herberg and Horn (Bull Math Biol. 2009) is in place as even the title is so close to the submitted manuscript.

Response: Thanks for the suggestions. We have added a discussion of some new references, including the papers of Roeder and Loeffler, in the introduction of the revision (Page 3).

- A few more words to motivate the boundary conditions on page 5 would be helpful.

Response: Thanks for the suggestion. We have added more explanation for the terms of the boundary conditions on page 5.

- I am missing biological interpretations of the results at several points throughout the manuscript. For example in Fig 3 the authors have marked “stable” regions. What does it mean biologically? Are they plausible?

Response: In the manuscript, the term “stable steady state” means that the population can return to its steady state after a relatively small perturbation. In Fig.3, the exact meaning of the yellow region is “no stem cell extinction”, means that the stem cell population is bounded below. The explanation has been included on page 9 in the revision.

- Page 13, line 45: strange sentence. Something missing?

Response: Thanks for pointing out. The sentence is edited and re-organized with the previous sentence (line 44), “The comparison of population dynamics among different combinations of α_1 and γ_1 (Fig. 7A) shows that higher damage retention results in oscillations in population evolution”, on page 14 of the revision.

- As my main critic I am missing simulations of perturbation scenarios. In fact, most of our knowledge on stem cell organization results from experiments in which the systems are challenged and return (or do not return) back to normal. In the hematopoietic case these are typically (serial) transplantations, irradiation, knock-outs or ochemotox. It would give all the modeling approaches much more meaning and potentially allow to identify crucial regulatory mechanisms, if at least some of these experiments could be mimicked. Describing a “homeostatic system” is nice, but not too exciting.

Response: Thanks for the suggestion. The aim of this study is to provide a framework model which integrates stem cell proliferation and differentiation with damage accumulation, feedback regulation and damage segregation in the stem cell aging process. Since there is no enough biological data on damage distribution in stem cell population at this stage, we focus on studying the stability which can provide a basic theoretical study on the dynamic of system under a perturbation; we also study the damage segregation during cell division which can motivate more biological studies about the damage distribution in stem cell population. We have added this comment in the conclusion for motivating further study.

- Conclusion: The authors are missing the point to draw conclusions about the suitability of their model. Are their biological findings that favor one feedback or one regulation over the other. Is there any finding with biological relevance, something testable beyond the modeling work?

Response: Thanks for the suggestion. We have added some references for the feedback regulation in the conclusion.

Reviewer #2:

In this work, the authors presented a very first mathematical model on stem cell proliferation and differentiation with an age structure. One unique feature of the model is its inclusion of a variable for molecular damage in stem cells, which then describes the aging process in stem cells. The case without feedbacks was analyzed in details with interesting analytical results, and the cases with various feedbacks that are difficult to analyze were computationally simulated. The integrated study of models, rigor analysis, and simulations represents a major advance in stem cell modeling, as aging modeling becomes an increasingly important topic. The results are clearly presented, and the paper is well written. I recommend it for publication. One minor comment is its connection with specific biology.

It would make the work more significant if the model could be applied to, or at least be discussed for, a specific biological system in which the stem cell age structure is important.

Response: Thanks for the comments. We have provided more biological explanations for the modeling part, introduction, and conclusion in the revision for the biologically oriented readership.

Subject Areas:

Mathematical and Computational
Biology

Keywords:

Stem cell aging, Modeling, Feedback
regulation

Author for correspondence:

Wing-Cheong Lo

e-mail: wingclo@cityu.edu.hk

Modeling stem cell aging: a multi-compartment continuum approach

Yanli Wang^{1,*}, Wing-Cheong Lo^{2,*} and
Ching-Shin Chou¹

¹Department of Mathematics, The Ohio State
University, Columbus, OH, USA

² Department of Mathematics, City University of Hong
Kong, Hong Kong, China

*Co-first author

Stem cells are important to generate all specialized tissues at an early life stage, and in some systems, they also have repair functions to replenish the adult tissues. Repeated cell divisions lead to the accumulation of molecular damages in stem cells, which are commonly recognized as drivers of aging. In this paper, a novel model is proposed to integrate stem cell proliferation and differentiation with damage accumulation in the stem cell aging process. A system of two structured PDEs is used to model the population densities of stem cells (including all multiple progenitors) and terminally differentiated (TD) cells. In this system, cell cycle progression and damage accumulation are modeled by continuous dynamics, and damage segregation between daughter cells is considered at each division. Analysis and numerical simulations are conducted to study the steady state populations and stem cell damage distributions under different damage segregation strategies. Our simulations suggest that equal distribution of the damaging substance between stem cells in a symmetric renewal and less damage retention in stem cells in the asymmetric division are favorable strategies, which reduce the death rate of the stem cells and increase the TD cell populations. Moreover, asymmetric damage segregation in stem cells leads to less concentrated damage distribution in the stem cell population, which may be more robust to the stochastic changes in the damage. The feedback regulation from stem cells can reduce oscillations and population overshoot in the process, and improve the fitness of stem cells by increasing the percentage of cells with less damage in the stem cell population.

1. Introduction

Stem cells are characterized by their ability to give rise to a variety of cell types through self-renewal and differentiation [1]. Although the process is very dynamic, the stem cell population is stable and remains almost steady. When a stem cell divides, each progeny has the potential to remain as a stem cell (self-renewal) or becomes a cell with a more specialized function (differentiation) (Fig. 1A). If the division produces the same type of cells, such as two stem cells and two differentiated cells, it is called a symmetric division. On the other hand, if the division produces two different types of cells, i.e., a stem cell and a differentiated cell, then it is called an asymmetric division. Available data suggest that most stem cells are able to switch between symmetric and asymmetric divisions, and the balance between these modes is controlled by various internal and external signals to produce appropriate numbers of stem and differentiated cells [1–3].

Stem cells can be found in most mammalian tissues, and they participate in tissue repair in response to damage and maintaining tissue homeostasis [4,5]. Research suggests that the decline in adult tissue maintenance and the increase in cancer formation might be a consequence of stem cell aging [6]. Although age-related manifestation in stem cell population and function differ across tissues and organisms, decline in regenerative capacity due to depletion or dysfunction of stem cells is a hallmark [7]. Protein aggregates, dysfunction organelles, and DNA damages are commonly identified as factors of aging [4,5]. To slow down the accumulation of aging factors, a hypothesis suggests that mitotic cells (actively dividing cells) might asymmetrically segregate damages away from the cell whose fate is to become a new stem cell [8]. The asymmetric inheritance of cellular components in dividing cells was first observed in yeast, and has been extensively studied over the past decades. In yeast, carbonylated proteins, extrachromosomal ribosomal DNA circles and dysfunction mitochondria are retained by a mother cell during asymmetric division, while its daughter is rejuvenated with little damage [9–11]. Recent evidence suggests that stem cells may employ a similar mechanism to protect one progeny from aging [12–18] (Fig. 1B). An asymmetric partition of damaged proteins in stem cell division was observed in adult flies' intestine and germline [13], mammalian stem cells [15], and murine neural stem cells [16]. Despite these emerging findings, it still remains elusive how stem cells cope with damage accumulation and how this is related to stem cell proliferation and differentiation.

Figure 1. Cell divisions and damage segregation in stem cells. A) Three types of division in stem cell population: SR stands for symmetric renewal, ASR&D stands for asymmetric renewal and differentiation, and SD stands for symmetric differentiation. B) Commonly recognized aging factors, such as protein aggregates, dysfunction organelle, and DNA damages are segregated asymmetrically between two cells during division.

Driven by the lack of knowledge of mechanisms regulating stem cell maintenance and differentiation in the aging process, mathematical models have been employed to address key questions and provide quantitative insights into stem cell renewal and differentiation, as well as the decline of cellular functions in the aging process. Several mathematical models were proposed

to study stem cell population and aging, which fall into two categories: individual-based modeling and continuous population modeling [19,20].

Individual-based modeling simulates individuals or agents that have a unique set of state variables and usually interact with each other in the local environment [20,21]. The advantages of this approach include that it can take stochastic effects into consideration to describe phenomena on the level of individual cells, and that detailed molecular dynamics and cell-cell interaction can be incorporated. In [22], a stochastic model of stem cell organization was introduced to explain the observed heterogeneity of hematopoietic stem cells by the stochastic switching between the growth environments and the self-organizing process based on within-tissue plasticity properties. Assuming that cell proliferation is negatively affected by telomere shortening, an agent-based stochastic model [23] was proposed to study telomere-dependent stem cell replicative aging. This model provides a good approximation of the qualitative growth of cultured human mesenchymal stem cells. In [24], mutation accumulation in large populations of stem cells was modeled by a discrete-time branching process where each division produces 0, 1 or 2 stem cell daughters, each of which randomly accumulates a mutation. This model demonstrated that symmetric division could reduce the risk of accumulating phenotypically silent heritable damage in individual stem cells. However, a major drawback of individual-based modeling is the computational inefficiency, especially when the population is large. When a population is large and homogeneous, continuous population model using ODEs [25–31] or PDEs [32–38] are more appropriate to describe the population dynamics. In [39], a maturity-structured two-compartment model was proposed to study the regulation of mammalian red blood cell production. The model consists of two transport equations describing population densities of mitotic cells and post-mitotic cells, and an ODE describing hormone dynamics. The model showed that a perturbation of blood-donation type leads to damped oscillatory return to normal status, and that an elevated random peripheral destruction of red blood cells leads to sustained oscillations. In [40], a three-compartment ODE model was applied to study the dynamics of stem cells, transit amplifying cells and terminally differentiated cells in the olfactory epithelium of mice. The authors identified conditions on parameters for the stability of the system when negative feedback loops are present either as Hill functions or in more general forms. Their analysis suggested that two factors, autoregulation of the proliferation of transit amplifying progenitor cells and low death rate of terminally differentiated cells, enhance the stability of the system. In [41], the authors applied the mean field approach to approximate an agent-based model for studying heterogeneity within the hematopoietic stem cell population. Their proposed PDE model can capture the key structure of the model including the “age”-structure of stem cells and improve the efficiency of the numerical algorithms. In [42], a system of PDEs was used to model mutation accumulation hierarchy and differentiation hierarchy of cells with stem cells on the top level and to examine cancer development and growth. In their model, maturity is treated as a continuous variable, while the number of mutation accumulation and telomere shortening are treated as discrete cell classes. The boundary conditions describe transition among different cell classes at division: cells lose telomeres and acquire mutations. The study showed that the more mutation classes and higher proliferation rate are sufficient to explain the faster growth of the cancer cell population.

To understand the effects of different damage segregation strategies on stem cell aging, we propose a novel model to integrate stem cell proliferation and differentiation with damage accumulation in the stem cell aging process, and feedback regulation from stem and TD cells. A system of hyperbolic PDEs is constructed to model two compartments in cell lineage: mitotic cells (stem cells) and post-mitotic cells (terminally differentiated, namely, TD cells). It is assumed that the cell cycle progression of stem cells is a continuous process while stem cell division is discrete. The boundary conditions of the PDEs model the stem cell renewal and differentiation at division when damage segregation takes place. Cell death is modeled as an outcome of damage accumulation. Stem cell proliferation and differentiation are regulated by feedbacks from the population of TD cells and stem cells. Aging effect is modeled through the inhibition from the damage accumulation on stem cell proliferation and self-renewal. Our analysis and numerical

simulations are carried out to compare the effects of different regulation and damage segregation strategies on population dynamics and stem cell fitness, which have not been discussed in the previous studies of stem cell regulation.

Our simulations suggest that equal distribution of the damaging substance between stem cells in symmetric renewal and less damage retention in stem cell in asymmetric division are favorable strategies, which reduce the death rate of stem cells and increase TD cell populations. Also, asymmetric damage segregation in stem cells leads to less concentrated damage distribution in stem cells population, which may be more robust to the stochastic change in damage. Compared to the feedbacks solely from TD cells, the feedback regulation from stem cells (autoregulation) can reduce oscillations and population overshoot in the process, and improve the fitness of stem cells by increasing the percentage of stem cells with less damage in the stem cell population.

This paper is structured as follows. The general description of our model is given in Section 2. In Section 3, a simple model without feedback regulations is presented to analyze the relation between population dynamics and various parameters. In Section 4 two more complex models with feedbacks from TD cells and stem cells are proposed to study different regulation mechanisms and the effect of segregation strategies.

2. Model Description

In our mathematical model, a simplified conceptual model, we consider two types of cells: mitotic cells, which include stem cells and multiple progenitor cells, and post-mitotic cells, which include all terminally differentiated cells. For simplicity, we call mitotic cells stem cells, and denote its population density by S ; and we call post-mitotic cells TD cells, and denote its population density by T .

To consider the evolution of cell populations, we define both $S(t, p, a)$ and $T(t, p, a)$ as functions of time t and two other continuous biological state variables: cell cycle progression p and damage level a . To describe cell maturation and aging processes, we make the following assumptions:

- When the amount of damage a accumulates and reaches a certain threshold a^* , stem cells die by an apoptosis-like process as a result of aging via damage accumulation; similar to stem cells, when the amount of damaged substance reaches a certain threshold a^c , TD cells are removed by an apoptosis-like process;
- Cell cycle progression p is an indicator variable, for which a stem cell divides when p increases to a threshold p^* , unless damage has already reached a threshold a^* , in which case the stem cell is removed before division. Although TD cells no longer divide, for simplicity, we keep cell cycle progression variable p in T , but assume the upper boundary for p is infinity.

By conservation law, a system of transport equations are derived to describe the evolution of S and T :

$$\frac{\partial S}{\partial t} + \frac{\partial}{\partial p}(V_p S) + \frac{\partial}{\partial a}(V_a S) = 0, \quad (2.1)$$

$$\frac{\partial T}{\partial t} + \frac{\partial}{\partial p}(U_p T) + \frac{\partial}{\partial a}(U_a T) = 0, \quad (2.2)$$

where V_p and U_p are rates of cell cycle progression for stem and TD cells, respectively; V_a and U_a are rates of damage accumulation for stem and TD cells, respectively. Note that various of feedbacks may regulate cell cycle progression and cell damage accumulations, and these functions will be specified in the following sections.

The boundary conditions at $p = 0$ describe the reproduction/division process accompanied by the segregation of damage substances. First, we assume that there are three types of cell divisions:

- (i) The daughter cells after division are two stem cells; this occurs with probability δ_1 .
- (ii) The daughter cells after division are two terminally differentiated (TD) cells; this occurs with probability δ_2 .
- (iii) The daughter cells after division are one stem cell and one TD cell; this occurs with probability δ_3 .

One of these three types of cell division occurs at the end of cell cycle $p = p^*$ with probability δ_1, δ_2 or δ_3 , where $\delta_1 + \delta_2 + \delta_3 = 1$ (See Fig. 2A). These three probabilities may be regulated by various feedbacks and will be specified in later sections.

Upon the completion of a cell cycle, the stem cell cycle progression p will be reset to zero and the damaged proteins are inherited from mother to daughters. The damage inheritance can be described by transition kernels $r_{i,S}(a, a')$ and $r_{i,T}(a, a')$. Based on the above assumptions, the boundary conditions for S and T cells at $p = 0$ are as follows:

$$V_p S(t, 0, a) = \int_0^{a^*} \delta_1 V_p r_{1,S}(a, a') S(t, p^*, a') da' + \int_0^{a^*} \delta_3 V_p r_{3,S}(a, a') S(t, p^*, a') da', \quad (2.3)$$

$$U_p T(t, 0, a) = \int_0^{a^*} \delta_2 V_p r_{2,T}(a, a') S(t, p^*, a') da' + \int_0^{a^*} \delta_3 V_p r_{3,T}(a, a') S(t, p^*, a') da'. \quad (2.4)$$

The first term of the right-hand side of (2.3) represents the stem cell production process through type (i) cell division; the second term of the right-hand side of (2.3) represents the stem cell process through type (iii) cell division; the first term of the right-hand side of (2.4) represents the TD cell production process through type (ii) cell division; the second term of the right-hand side of (2.4) represents the TD cell production process through type (iii) cell division. The transition kernel $r_{i,S}(a, a')$ describes how daughter stem cells with damage a come from the mother cells with damage a' after the i -th type of division; the transition kernel $r_{i,T}(a, a')$, describes how daughter TD cells with damage a come from the mother cells with damage a' in the i -th type of division. The transition kernels satisfy the following conservation conditions:

$$\begin{aligned} \int_0^{a^*} \int_0^{a^*} r_{1,S}(a, a') da da' &= 2, & \int_0^{a^*} \int_0^{a^c} r_{2,T}(a, a') da da' &= 2, \\ \int_0^{a^*} \int_0^{a^*} r_{3,S}(a, a') da da' &= 1, & \int_0^{a^*} \int_0^{a^c} r_{3,T}(a, a') da da' &= 1, \\ \int_0^{a^*} a r_{1,S}(a, a') da &= a', & \int_0^{a^c} a r_{2,T}(a, a') da &= a', \\ \int_0^{a^*} a r_{3,S}(a, a') da + \int_0^{a^c} a r_{3,T}(a, a') da &= a'. \end{aligned}$$

Since type (i) and type (ii) divisions induce two daughter stem cells and two daughter TD cells, respectively, the integrals of $r_{1,S}(a, a')$ and $r_{2,T}(a, a')$ equals to two; since type (iii) division induces one daughter stem cell and one daughter TD cell, the integrals of $r_{3,S}(a, a')$ and $r_{3,T}(a, a')$ both equals to one. The last three integrals are based on the assumption that the amount of damage is conserved during cell division. No-flux conditions on the boundary $a = 0$ are imposed to ensure conservation of population in the direction of a :

$$S(t, p, 0) = 0, \quad \text{for } t > 0, p \in [0, p^*], \quad (2.5)$$

$$T(t, p, 0) = 0, \quad \text{for } t > 0, p \in [0, \infty). \quad (2.6)$$

For TD cells, we assume that the population density of TD is zero when p is very large:

$$\lim_{p \rightarrow \infty} T(t, p, a) = 0. \quad (2.7)$$

The populations of stem and TD cells at time t are given by the integrals

$$P_S(t) = \int_0^{a^*} \int_0^{p^*} S(t, p, a) dp da \quad \text{and} \quad P_T(t) = \int_0^{a^c} \int_0^\infty T(t, p, a) dp da.$$

Based on this model, we will study the population dynamics under two scenarios:

- (i) For the simplest case where no feedbacks are involved in regulating the stem cell division and differentiation, the dynamics of stem cell population mainly depend on stem cell damage segregation rules. Different segregation rules may result in exploded, stabilized or extinct stem cell pool.
- (ii) Feedbacks from TD cells and stem cells are introduced and population evolutions are simulated to study the effect of different damage segregation rules.

Figure 2. Model descriptions of two-compartment stem cell system. Stem cells (mitotic cells) and TD cells (post-mitotic cells) are modeled as two compartments. A) Stem cells renew themselves and replenish TD cells. The cell cycle progression (p) and damage accumulation (a) are modeled as continuous processes. Stem cell division and damage segregation take place at the end of its cell cycle ($p = p^*$). Cells die when damage reaches a lethal threshold (a^* or a^c). B) In the simple model in Section 3, the proportions of three types of division are constants δ_i , and the damage segregation rules are fixed, i.e., $\alpha_i, \beta_i, \gamma_i$ are constants.

3. Model without feedback regulations

In this section, we start with the simple model where no feedbacks are included, and we additionally make the following assumptions to simplify the model (see Fig. 2B):

- Cell cycle progression velocities $V_p = v_p$ and $U_p = u_p$ are constants, so are the damage accumulation rates $V_a = v_a$ and $U_a = u_a$;
- The portions of three types of divisions δ_i are constants;
- When stem cells divide, damage is partitioned into portions α_1 and α_2 between stem cells, β_1 and β_2 between TD cells, or γ_1 and γ_2 between stem and TD cells, where $\alpha_1 + \alpha_2 = 1, \beta_1 + \beta_2 = 1$ and $\gamma_1 + \gamma_2 = 1$. Without loss of generality, we assume $\alpha_1 \leq \alpha_2$ and $\beta_1 \leq \beta_2$.

Under these assumptions, the transition kernels in the boundary conditions are Dirac delta functions:

$$r_{1,S}(a, a') = \delta(a/a' - \alpha_1) + \delta(a/a' - \alpha_2), \quad r_{2,T}(a, a') = \delta(a/a' - \beta_1) + \delta(a/a' - \beta_2),$$

$$r_{3,S}(a, a') = \delta(a/a' - \gamma_1), \quad r_{3,T}(a, a') = \delta(a/a' - \gamma_2),$$

and the boundary conditions in (2.3) and (2.4) become

$$S(t, 0, a) = \frac{\delta_1}{\alpha_1} S\left(t, p^*, \frac{a}{\alpha_1}\right) + \frac{\delta_1}{\alpha_2} S\left(t, p^*, \frac{a}{\alpha_2}\right) + \frac{\delta_3}{\gamma_1} S\left(t, p^*, \frac{a}{\gamma_1}\right) \quad (3.1)$$

$$T(t, 0, a) = \frac{v_p \delta_2}{u_p \beta_1} S\left(t, p^*, \frac{a}{\beta_1}\right) + \frac{v_p \delta_2}{u_p \beta_2} S\left(t, p^*, \frac{a}{\beta_2}\right) + \frac{v_p \delta_3}{u_p \gamma_2} S\left(t, p^*, \frac{a}{\gamma_2}\right). \quad (3.2)$$

Now, we study the role of the damage segregation rules in the long-term behavior of the stem cell population. Since the segregation rule is fixed, after sufficiently long time, the cellular damage at the end of cell cycle, temporarily assuming no death, converges to a **limit damage band**

$$\left[\frac{p^*}{v_p(1-\omega_1)}, \frac{p^*}{v_p(1-\omega_2)} \right],$$

where ω_1 and ω_2 are the minimum and the maximum of $\alpha_1, \alpha_2, \gamma_1$, respectively. The derivation of damage limit band can be found in Appendix (a).

The population dynamics turns out to depend on the proportions of three division types of stem cells and the position of the lethal threshold a^*/v_a with respect to the limit damage band $\left[\frac{p^*}{v_p(1-\omega_1)}, \frac{p^*}{v_p(1-\omega_2)} \right]$. Before proceeding, we define a key lumped parameter for studying the long-term population behavior,

$$f_r = \frac{2\delta_1 + \delta_3}{2} = \frac{1 + \delta_1 - \delta_2}{2}.$$

Here the parameter f_r is called the **self-renewal fraction**. Using the limit damage band and the self-renewal fraction, we can find some conditions for different long-term behaviors.

Proposition 3.1. *Stem cells become extinct, if the renewal fraction $f_r < 1/2$ or the limit damage band is completely above the lethal threshold, i.e., $\frac{a^*}{v_a} < \frac{p^*}{v_p(1-\omega_1)}$.*

Proposition 3.2. *Assume that the limit damage band lies completely below the lethal threshold, i.e., $\frac{a^*}{v_a} > \frac{p^*}{v_p(1-\omega_2)}$ and*

$$\int_0^{p^*} \int_0^{a^* - v_a(p^* - p)/v_p} S(0, p, a) da dp > 0.$$

Then stem cell population blows up if the renewal fraction $f_r > 1/2$, or is eventually conserved if $f_r = 1/2$.

The proofs of Propositions 3.1 and 3.2 can be found in Appendix (b). In the situations shown in Propositions 3.1 and 3.2, either damage accumulation does not affect cells or no cell can survive after sufficiently long time. Next we consider the more intriguing intermediate situations as follows, based on the following assumptions on the renewal fraction and the lethal threshold a^*/v_a lies within the limit damage band:

$$f_r > \frac{1}{2} \quad \text{and} \quad \frac{p^*}{v_p(1-\omega_1)} \leq \frac{a^*}{v_a} < \frac{p^*}{v_p(1-\omega_2)}. \quad (3.3)$$

Under the above assumptions in (3.3), there are many situations of damage segregation rules, and more importantly not all of the situations are biologically meaningful. Here, we will focus on one situation: the damage retention in stem cells through asymmetric division is smaller than the

damage segregation portions in symmetric stem cell renewal, i.e.,

$$\gamma_1 \leq \alpha_1 < \alpha_2. \quad (3.4)$$

This setting is based on the biological observation that asymmetric division is a favorable mechanism for stem cell lineage to remove the damage. For avoiding the case that the lethal threshold a^*/v_a lies below $\frac{p^*}{v_a(1-\alpha_1)}$, i.e., all asymmetrically renewing stem cells are destined to die, we further assume that

$$\frac{p^*}{v_p(1-\alpha_1)} < \frac{a^*}{v_a}. \quad (3.5)$$

By the assumptions (3.3)-(3.5), we obtain that

$$f_r > \frac{1}{2} \quad \text{and} \quad \frac{p^*}{v_p(1-\gamma_1)} \leq \frac{p^*}{v_p(1-\alpha_1)} < \frac{a^*}{v_a} < \frac{p^*}{v_p(1-\alpha_2)}. \quad (3.6)$$

Under the assumption (3.6), the stem cell population may approach to zero or blow up in the long-term behavior. The following proposition provides a condition to guarantee extinction of stem cells, as well as a condition for population blow-up, with the proof given in Appendix (b).

Proposition 3.3. *Assume that (3.6) holds.*

(a) *Define*

$$n = \min \left\{ m \in \mathbb{N} : \frac{a^*}{v_a} < \frac{p^*}{v_p(1-\alpha_2)} - \frac{p^*}{v_p} \left(\frac{1}{1-\alpha_2} - \frac{1}{1-\gamma_1} \right) \alpha_2^m \right\}. \quad (3.7)$$

If there exists a combination of δ_i such that $(2f_r)^n - \delta_1^n < 1$, then the stem cell population is eventually zero.

(b) *If $\delta_2 = 0$ and*

$$\int_0^{p^*} \int_0^{a^* - v_a(p^* - p)/v_p} S(0, p, a) da dp > 0,$$

then the stem cell population goes to infinity, or is bounded below to avoid stem cell extinction.

From Proposition 3.3, we can obtain several necessary conditions to maintain the stem cell population. Since the term $(2f_r)^n - \delta_1^n$ is increasing with n and goes to infinity when n tends to infinity, n has to be small enough to satisfy the condition $(2f_r)^n - \delta_1^n < 1$ in Proposition 3.3. From the definition of n , we observe that if

- the difference between $\frac{a^*}{v_a}$ and $\frac{p^*}{v_p(1-\alpha_2)}$ becomes small, or
- the difference between $\frac{1}{1-\alpha_2}$ and $\frac{1}{1-\gamma_1}$ becomes large, or
- the right-hand side $\frac{p^*}{v_p(1-\alpha_2)} - \frac{p^*}{v_p} \left(\frac{1}{1-\alpha_2} - \frac{1}{1-\gamma_1} \right) \alpha_2^m$ becomes large for each m ,

the value of n will increase and then $(2f_r)^n - \delta_1^n < 1$ cannot be satisfied in most of the combinations of δ_1, δ_2 and δ_3 . This result provides some conditions of the parameters to prevent stem cell extinction.

Let us discuss the above three possibilities. To reduce the difference between $\frac{a^*}{v_a}$ and $\frac{p^*}{v_p(1-\alpha_2)}$, the ratio of cell cycle progression speed v_p to damage accumulation speed v_a should be large enough and close to $\frac{p^*}{a^*(1-\alpha_2)}$; to increase the difference between $\frac{1}{1-\alpha_2}$ and $\frac{1}{1-\gamma_1}$, damage retention in asymmetric division should decrease, i.e., γ_1 should be small; to increase the right-hand side $\frac{p^*}{v_p(1-\alpha_2)} - \frac{p^*}{v_p} \left(\frac{1}{1-\alpha_2} - \frac{1}{1-\gamma_1} \right) \alpha_2^m$ for each m , damage distribution in self-renewal should become more symmetric, i.e., α_1 should increase to close to 0.5. In conclusion, when v_p/v_a increases, γ_1 decreases or α_1 increases, it may provide a better condition to maintain the stem cell population. These results are supported by the numerical simulations shown in Fig. 3.

In Fig. 3, we study the long-term population dynamics with different combinations of the division probabilities δ_1 and δ_3 . We uniformly generate 600 pairs of δ_1 and δ_3 with $0 < \delta_1 + \delta_3 \leq$

1 and $2\delta_1 + \delta_3 \geq 1$. We skip the region $2\delta_1 + \delta_3 < 1$ as it implies stem cell extinction whatever other parameters are (by Proposition 3.1). For each pair of δ_1 and δ_3 , the system is solved by the numerical method described in Appendix (e). The blue regions in Fig. 3 represent the cases of stem cell extinction and the yellow regions represents the non-extinction cases (the stem cell population may tend to a constant non-zero value, approach to infinity, or keep oscillating). When v_p/v_a increases (in Fig. 3A, $v_p/v_a = 1.67$; in Fig. 3B, $v_p/v_a = 2$), γ_1 decreases (in Fig. 3C, $\gamma_1 = 0.3$; in Fig. 3D, $\gamma_1 = 0.05$) or α_1 increases (in Fig. 3E, $\alpha_1 = 0.1$; in Fig. 3F, $\alpha_1 = 0.45$), the blue regions become smaller (the yellow regions become larger), which implies that there are more combinations of parameters (δ_1, δ_3) allowing stem cell survival.

For most of the combinations of parameters, the stem cell populations in the models without feedback regulation blow up to infinity or diminish to zero. Although the no-regulation assumption is not realistic, the simple model not only provides us a guidance on the selection of parameters used in the model with feedbacks, but also reveals that population dynamics are results of all factors: the cell cycle progression of stem cell, damage accumulation, fractions of divisions and damage segregation rules. In the next section, we will consider the combinations of parameters that guarantee exponential growth in the stem cell population and study feedback regulations that could lead to nonzero steady state population.

Figure 3. The long-term stem cell population dynamics with different combinations of the division probabilities δ_1 and δ_3 . The yellow regions represent the combinations of (δ_1, δ_3) whose corresponding population blow up or is eventually conserved, while the blue regions represent the situation where population go to zero. The parameter values are set to be $\alpha_1 = 0.3$, $\gamma_1 = 0.1$, $p^* = 1$, $a^* = 1$, $v_p = 0.2$ and $v_a = 0.1$, if not mentioned in the subfigures.

4. Model with feedback regulations

Biological evidence shows that some mammalian stem cells can switch between symmetric and asymmetric divisions in response to external and internal regulations [1,2]. For example, both epidermal [43] and neural [44] progenitors change from the primarily symmetric division that expands the stem-cell pool during embryonic development to primarily asymmetric in mid to the late gestation. It is also observed that nervous [45] and haematopoietic [46] stem cells in adults can divide symmetrically to replace lost cells through injury, although they divide asymmetrically under steady-state conditions. In particular, two types of feedbacks have been proposed [37,47,48] (also see Fig. 4A): long-range feedbacks responding to the population of TD cells, and short-range feedbacks acting in an autocrine fashion from stem cells. These two types of feedbacks regulate stem cell population through controlling two types of parameters:

- (i) the speed of stem cell division, V_p (or stem cell cycle progression),
- (ii) the probabilities of three types of cell division, δ_i (differentiation v.s. renewal).

Figure 4. Feedbacks in stem cell lineage. (A) Two types of feedback occur: the long-range feedback responding to the population of TD cells, and short-range feedback acting in an autocrine fashion in stem cells. (B) Two-compartment model with feedbacks from TD cells. (C) Two-compartment model with feedbacks from stem cells and TD cells.

However, the underlying mechanisms are not fully understood. In addition, the effect of damage segregation rules on cell population is completely unknown. In the following, we

use mathematical models with different feedbacks and damage segregation rules to study which types of feedback and damage segregation mechanisms are more plausible given the experimental observations.

For simplicity, we assume that damage segregation rules are fixed, i.e., $\alpha_i, \beta_i, \gamma_i$ are constants. Based on the results from the simple model in Section 3, to ensure that the stem cell population will blow up to infinity without any feedback regulation, additional assumptions are made:

- Self-renewal fraction $f_r > \frac{1}{2}$.
- Damage accumulation speed $V_a = v_a$ is slower than cell cycle progression of stem cells.

(a) Feedback only from TD cells

The long-range feedback, which acts through signals sent by differentiated cells and inhibits stem cell division and self-renewal, has been biologically observed in numerous tissues, including muscle [49], bone [50], skin [51], nervous system [52] and hematopoietic systems [53]. Despite this wealth of data, there is less understanding of the exact mechanisms of feedback regulation. A significant number of mathematical models have been developed to explore the possible mechanisms behind feedback regulations [26,28,32–34,36–40,54,55]. The dynamics of signaling molecule $s(t)$ can be described by a simple ODE as follows:

$$\frac{d}{dt}s(t) = \alpha - (\mu + \beta P^m)s(t), \quad (4.1)$$

where α is a constant synthesis rate, and the degradation is proportional to the level of s and affected by the cell population P of stem or TD cells. Since the dynamics of the signaling molecules take place on a faster time scale than the processes of cell proliferation and differentiation, we assume that the feedback signal can be approximated by a quasi-steady state solution [25,28,32–34,36–40,54,55]. By properly rescaling s , the quasi-steady state of s is given by a Hill function

$$s = \frac{1}{1 + (kP)^m}, \quad (4.2)$$

where k is a regulation constant to account for the sensitivity to the cell population and m is the Hill exponent. The function in (4.2) reflects the assumption that the signal intensity achieves its maximum in the absence of cells and decreases asymptotically to zero if the number of cells increases. Note that Hill functions are widely used to describe ligand-receptor interactions, which also makes them natural choices to model the actions of secreted feedback factors [36].

According to biological evidence, we may model the feedback from TD cells in the following ways (see Fig. 4B): when stem cell or TD cell population is small (in the early stage of development or with drastic loss due to injury), symmetric division predominates over asymmetric division; during the stable stage (mid and late gestation or tissue homeostasis), stem cells switch to asymmetric division. Such feedbacks may be added to the model by modifying division fractions δ_i as follows:

$$\delta_i(P_T) = \frac{\delta_i^0}{1 + (k_i^T P_T)^{m_T}}, \quad (4.3)$$

where $i \in \{1, 2\}$, δ_i^0 are basal division fractions, k_i^T are regulation constants, m_T is the Hill exponent and $P_T(t)$ is the TD cell population. The remaining division fraction δ_3 is defined as $1 - \delta_1 - \delta_2$.

Recent studies on cell cycles of neural stem cells have shown evidence that terminally differentiated cells may be a source of signaling molecules that inhibits cell cycle progression of stem cells [56]. Therefore, besides the above regulation on cell fate decisions, negative feedback from TD cells can also be involved to stem cell proliferation V_p (cell cycle progression speed), i.e., excessive number of TD cell may slow down the proliferation of stem cell, hence reduce the

Table 1. The choices of parameters, estimated based on biological evidence and previous modeling works, with details in Appendix i.

Parameters	Meaning	Value
p^*	Cell cycle threshold	1
a^*	Damage level threshold for stem cell	1
a^c	Damage level threshold for TD cell	1
v_p	Maximum cell cycle progression speed of stem cells	0.2
v_a	Constant damage accumulation speed of stem cells	0.06
u_p	Constant cell cycle progression speed of TD cells	0.02
u_a	Constant damage accumulation speed of TD cells	0.02
δ_1^0	Maximum fraction of symmetric renewal of stem cells	0.6
δ_2^0	Maximum fraction of symmetric differentiation of stem cells	0.3
δ_3^0	Maximum fraction of asymmetric division of stem cells	0.1
k_1^T	Regulation constant	10^{-8}
k_2^T	Regulation constant	0.5×10^{-8}
k_v^T	Regulation constant	0.5×10^{-8}
m_T	Hill exponent	2

population of both stem cells and TD cells. Thus, V_p can be modified as

$$V_p(P_T) = \frac{v_p}{1 + (k_v^T P_T)^{m_T}}, \quad (4.4)$$

where v_p is the initial cell cycle progression speed and k_v^T is a regulation constant. For other velocities, we set $V_a = v_a$, $U_a = u_a$ and $U_p = u_p$.

Among all parameters, we are most interested in the effect of segregation rules $\alpha_i, \beta_i, \gamma_i$ on the population dynamics. A set of reasonable estimations of the parameters other than $\alpha_i, \beta_i, \gamma_i$ are chosen based on appropriate biological and mathematical assumptions, and are shown in Table 1. The detailed reasoning of the selection of parameters is provided in Appendix i. In brief, we assume that initially, the stem cells are expanding, the cell cycle progression is fast, and the symmetric renewal predominates in three types of division. In the rest of this subsection, we will focus on the discussion of the effect of different combinations of segregation rules $\alpha_i, \beta_i, \gamma_i$.

In the simulations, we consider the following situations: α_1 varies from 0.1 to 0.5, indicating that the symmetry in damage segregation increases between the two stem daughter cells; β_1 varies from 0.1 to 0.5, indicating that the symmetry in damage segregation increases between TD cells; γ_1 varies from 0.1 to 0.9, indicating that the damage retention in stem cells increases in asymmetric division. Under different segregation rules, the following aspects are studied:

- the population size of TD cells and the population ratio of TD cells to stem cells at steady state.
- the death rate of stem cells and the fraction of three types of division at steady state.
- the dynamics of population evolution and the damage distribution in stem cell population at steady state.

Figure 5. Model with feedback only from TD cells: TD cell population and population ratio at steady state for different combinations of α_1 , β_1 and γ_1 . The parameters used in these simulations are shown in Table 1.

Population size of TD cells and population ratio of TD cells to stem cells

Fig. 5 shows that the damage distribution between TD cells in symmetric differentiation (β_1 and β_2) does not affect population size at steady state. When the parameter β_1 , which determines how damage is distributed between two TD cells in the symmetric differentiation of stem cells, varies from 0.1 to 0.5, neither stem cell population nor TD cell population has a big change. This result is biologically meaningful, since the replenishment of TD cells is efficient compared to the loss of TD cells due to damage accumulation. In the following part, we will focus on the case with $\beta_1 = 0.5$.

Fig. 5 also shows that as damage segregation α_i between stem cells becomes more symmetric, TD cell populations increase, while the ratios P_T/P_S are almost constant with a slight decrease. For fixed α_1 , as damage retention γ_1 in asymmetric division increases, both TD cell populations and the population ratios decrease. Interestingly, when $\alpha_1 = 0.5$, although the populations of TD cells are very close for the cases $\gamma_1 = 0.1, 0.3$ and 0.5 , the population ratios differ dramatically. Compared to $\gamma_1 = 0.1$, one needs a larger stem cell pool to generate a similar number of TD cells when damage retention is relative higher, i.e., $\gamma_1 = 0.5$. Moreover, it would be very difficult to find explicit formulas for the populations, since the population sizes at steady state are affected by all parameters as observed in the simulations. However, when steady state exists, we can give an upper-bound estimate of the populations of stem cells and TD cells, with details given in Appendix (d).

Death rate of stem cells and fraction of three types of division

Other than studying the population size, we also compare that death rate of stem cell and the fraction of three types of division at steady state. In our model, the death rates of stem cell (r_S) and TD cell (r_T) are measured in the following way:

$$r_S = \frac{\int_0^{p^*} v_a S(t, p, a^*) dp}{P_S} \quad \text{and} \quad r_T = \frac{\int_0^\infty u_a T(t, p, a^c) dp}{P_T}, \quad (4.5)$$

Figure 6. Results of the death rates of stem cell and the fractions of symmetric division in steady state for the model with feedbacks from TD cells under different damage segregation rules. A) Death rate of stem cell at steady state. The definition of death rate can be found in Eq. (4.5). B) Symmetric renewal fraction δ_1 at steady state. The parameters used in these simulations are shown in Table 1 and $\beta_1 = 0.5$.

where the numerators are population out-fluxes due to death and the denominators are the total populations. Fig. 6 shows that more equal distribution of damage in symmetric renewal and less damage retention in asymmetric division result in a lower death rate of stem cells and less symmetric division at steady state. According to our simulations, the segregation rules do not have much impact on the death rate of TD cells, which ranges from 2.130×10^{-4} to 3.345×10^{-4} . However, the death rate of stem cells changes dramatically as we vary the segregation rules. From Fig. 6A, we can see that the death rate of stem cells increases as α_1 decreases or γ_1 increases. The lowest death rate of stem cells is attained when damage is equally distributed between stem cells in symmetric renewal and damage retention is minimal in asymmetric division. To maintain the steady stem cell population and to replenish the short lived TD cells, a higher death rate should be associated with a fast turnover of stem cells. Indeed, when we examine the fractions of three types of division, we find that the higher death rate of stem cells is always associated with the higher fraction of symmetric renewal (Fig. 6B). These results suggest that to maintain stabilized populations, tissues may have different mechanisms that involve different damage segregation rules and division rules.

Dynamics of population evolution and damage distribution of stem cell population

The comparison of population dynamics among different combinations of α_1 and γ_1 (Fig. 7A) shows that higher damage retention results in oscillations in population evolution. Oscillations start to appear as we increase the damage retention γ_1 in stem cells during asymmetric division ($\gamma_1 = 0.7$ in Fig. 7A). Also, the oscillations become more severe when the damage distribution among stem cells in symmetric renewal becomes more symmetric ($\alpha_1 = 0.5$ and $\gamma_1 = 0.7$ in Fig. 7A). Moreover, population overshoots before steady state are observed in all cases with $\alpha_1 \neq 0.5$.

The results above suggest that more equal distribution of damage in symmetric renewal and less damage retention in asymmetric division are favorable segregation rules in four aspects: populations converge to steady states faster without oscillations or severe population overshoot; the death rate of stem cells is much smaller than that of TD cells; asymmetric division predominates in three types of division at steady state; not only the size of TD cell population is larger, the population ratio P_T/P_S is also larger.

However, the asymmetric damage segregation between stem cells in self-renewal may have benefits that cannot be shown from the analysis of total populations as integrated results. Our PDE modeling approach allows us to investigate more details of population density and damage distribution of stem cells. Fig. 7B gives the stem cell population densities corresponding to

Figure 7. Dynamics of population evolution and damage distribution of stem cell population at steady state for the model with feedbacks from TD cells under different damage segregation rules. A) Sample population dynamics under different combinations of α_1 and γ_1 . The red curves stand for TD cell population and the blue curves are for stem cell population. B) Population density and damage distribution for stem cells. In these three cases, we set $\gamma_1 = 0.3$. The other parameters used in these simulations are listed in Table 1 and $\beta_1 = 0.5$.

different segregation rules. As we studied in Section 3, damage segregation rules determine the limit damage band after a sufficiently long time. In particular, the less symmetric the segregation rules are, the wider the limit damage bands will be. Although asymmetric segregation rules may result in a greater percentage of stem cells inheriting more damage and accelerated death, it also leads to a higher percentage of healthier stem cells with less damage. This can be observed in the samples shown in Fig. 7B. When the symmetry of damage segregation in stem cell population increases, the damage distributions at the end of cell cycle for stem cells become more concentrated, as such symmetry increases. The concentrated damage distribution could be a disadvantage, since it is less resistant to external perturbations. In this sense, we think that some stem cells may sacrifice the lower death rate for more diversified damage distribution, in order to

protect the stem cell pool from a possible unfavorable situation that may lead to a sudden increase in damage.

(b) Feedback from both TD and stem cells

The maintenance of the stem cell pool is not only affected by signaling from mature cells, but also by the stem cell pool itself. Stem cells reside in the so-called stem cell niche, where both cellular and non-cellular components interact in order to control stem cell proliferation and differentiation [57–60]. The regulations from the stem cell population include two aspects: a negative feedback control of stem cell proliferation as a result of contact inhibition; and a self-renewal inhibition factor secreted by the stem cell niche, in which case the more stem cells there are, the less likely stem cells will divide symmetrically. Experiments also show that cells inheriting the majority of damage protein aggregates during asymmetric division have an increased cell cycle length and tendency to differentiate [14,16,17].

In addition to feedback regulations from differentiated cells, we are also interested in how the dynamics of populations would change if regulations from stem cells are included (See Fig. 4C). Many modeling works (see e.g. [37,40]) only consider the feedbacks from stem cell populations, due to the simplicity of their models. However, we include damage as a state variable in our model and could simulate feedbacks from both the stem cell population and the stem cell cellular damage level.

Thus, in our model, we assume that the excessive stem cell population and elevated stem cell cellular damage slow down stem cell proliferation and modify V_p in (4.4) as

$$V_p(P_T, P_S, a) = \frac{v_p}{1 + (k_v^T P_T)^{m_S} + (k_v^S P_S)^{m_S}} f(a), \quad (4.6)$$

where

$$f(a) = a_1 + \frac{b_1}{1 + e^{-k_1^a(a-a_1^0)}}$$

is a decreasing function of a with sigmoid shape, in which k_1^a , a_1^0 , a_1 and b_1 are constants.

We also assume that the excessive stem cell population inhibits stem cell symmetric division, and that the elevated stem cell cellular damage promotes stem cell differentiation via decreasing stem cell self-renewal. As a result, we modify δ_1 and δ_2 in (4.3) in the same way as above:

$$\delta_1(P_T, P_S, a) = \frac{\delta_1^0}{1 + (k_i^T P_T)^{m_S} + (k_i^S P_S)^{m_S}} g(a), \quad (4.7)$$

$$\delta_2(P_T, P_S, a) = \frac{\delta_2^0}{1 + (k_i^T P_T)^{m_S} + (k_i^S P_S)^{m_S}}, \quad (4.8)$$

where

$$g(a) = a_2 + \frac{b_2}{1 + e^{-k_2^a(a-a_2^0)}}$$

is another function of a sigmoid shape with constants k_2^a , a_2^0 , a_2 and b_2 .

As a continuation of Section (a), our analysis still focuses on the discussion of segregation rules. In addition to the parameters in Table 1, the values of more parameters are given in Table 2, for which the detailed reasoning can be found in Appendix ii.

To examine the change in population dynamics after introducing feedbacks from stem cells, the same combinations of segregation rules as in Section (a) are considered: α_1 varies from 0.1 to 0.5, β_1 varies from 0.1 to 0.5, and γ_1 varies from 0.1 to 0.9. Here, we study the effect of the feedbacks from stem cells on the following two aspects:

- the population size of TD cells and the population ratio of TD cells to stem cells at steady state,
- the dynamics of population evolution and the damage distribution of stem cell population at steady state.

Table 2. The choices of additional parameters, estimated based on biological evidence and previous modeling works, with details in Appendix ii.

Parameters	Meaning	Value
k_1^S	Regulation constant	10^{-7}
k_2^S	Regulation constant	0.25×10^{-7}
k_v^S	Regulation constant	0.5×10^{-7}
m_S	Hill exponent	2
a_1, a_2	Constant for sigmoid regulation	1.1
b_1, b_2	Constant for sigmoid regulation	-0.7
k_1^a, k_2^a	Stiffness parameter	20
a_1^0, a_2^0	Damage threshold for sigmoid regulation	0.75

Figure 8. TD cell population and population ratio in steady state for different combinations of α_1 , β_1 and γ_1 for the model with feedbacks from stem cells and TD cells. The parameters used in these simulations are shown in Tables 1 and 2 and $\beta_1 = 0.5$.

Population size of TD cells and the population ratio of TD cells to stem cells

Similar to the model in Section (a), damage distribution β_i between TD cells does not affect the results much. Since the regulation is stronger when we consider feedbacks from both stem cells and TD cells, the stabilized population of TD cells is smaller than that in Section (a) under the same parameters. It may not be meaningful to directly compare the size of populations in different models, since we do not have real experimental data. However, similar trends are observed in the model with the feedbacks from stem cells (see Fig. 8): for fixed α_1 , as damage retention γ_1 in stem cell increases, both the TD cell population and the population ratio P_S/P_T decrease; for fixed γ_1 , as damage segregation between stem cells becomes more symmetric, the TD cell population increases, but the population ratio is almost unchanged. It is worth mentioning that the population ratio increases a little bit for each damage segregation configuration, after adding feedbacks from stem cells. This means that the regulation is more efficient, since one needs a smaller stem cell pool to generate the same number of TD cells.

Figure 9. Dynamics of population evolution and damage distribution of stem cell population at steady state for the model with feedbacks from stem cells and TD cells under different segregation rules. A) Sample population dynamics under different combinations of α_1 and γ_1 . The red curves stand for TD cell population and the blue curves are for stem cell population. B) Population density and damage distribution for stem cells. In these three cases, we set $\gamma_1 = 0.3$. The other parameters used in these simulations are listed in Tables 1 and 2, and $\beta_1 = 0.5$.

Dynamics of population evolution and damage distribution of stem cell population

A notable consequence of adding feedbacks from stem cells is the reduction of the oscillations in population dynamics. Comparing Fig. 7A with Fig. 9A, after considering the feedbacks from stem cells, the oscillations in population dynamics disappear even when γ_1 is large, and that the population overshoot problem resolves when α_1 is small.

Since the feedbacks from stem cells described by (4.7) and (4.6) also depend on stem cell cellular damage level, we want to examine the effect of such regulations on damage distribution at the end of cell cycle by comparing the models with/without the feedbacks from stem cells (Fig. 7B and Fig. 9B). Similar to Section (a), the damage distributions at the end of cell cycle become more

concentrated, as the degree of symmetry in damage segregation increases. However, compared to the results in Section (a) (Fig. 7B), the model with the feedback from stem cells (see Fig. 9B) provides better fitness, since under same segregation rules, there are more cells with less damage at the end of cell cycle at steady state. This effect becomes most significant when the segregation rule is symmetric. This result suggests that slowing down the cell cycle progression of stem cells with a high level of damage and promoting such stem cells to differentiate can indeed improve the overall health of the stem cell population.

5. Conclusion

In this research, a novel model was developed to integrate stem cell proliferation and differentiation with damage accumulation in stem cell aging process. A system of two structured PDEs are used to model stem cells (including all multiple progenitors) and TD cells. In our model, cell cycle progression and damage accumulation are continuous while division is discrete, and the damage segregation takes place at each division. Aging effect is included through the inhibition from damage accumulation on stem cell proliferation and self-renewal. Regulations from TD cell and stem cell populations are incorporated through negative feedbacks on stem cell proliferation and symmetric division.

Our results showed that more equal distribution of damage between stem cells in symmetric renewal and less damage retention in stem cell in asymmetric division are still favorable segregation rules resulting in higher population size and greater population ratio; asymmetric damage segregation in stem cells leads to less concentrated damage distribution in stem cells population, which may be more robust to sudden increase in damage. **These two results provides an answer for why some types of stem cells can switch between symmetric and asymmetric divisions in response to external and internal regulations [1,2].**

Other than the feedbacks solely from TD cells, adding feedback regulations from stem cells can reduce oscillations and population overshoot in some situations with unfavorable damage segregation rules, say, in the situation where α_1 is small and γ_1 is large. Moreover, the stem cell feedback regulation can slow down the proliferation of stem cells with high level of damage to increases their tendency to differentiate. **Overall, the feedback regulation system can improve the fitness of stem cells by increasing the percentage of stem cells with low level of damage. This result provide a new aspect to understand the role of the self-regulations in stem cell lineage [38,47,52,61].**

In this study, our model provides a framework to study how the damage segregation rules at division affect the population dynamics. **Moreover, our model is an excellent tool to understand the data which may result from observing the dynamics of regeneration with damage distribution in stem cell population after a knock-out process. In the future work, we can extend our study to the system with more than two stages, including progenitor cells [40,61] to get a better understanding for the system of stem cell lineage.**

Data Accessibility. This article has no additional data. The code for the numerical method can be found in the electronic supplementary material.

Authors' Contributions. All authors were involved in the design of the model and the analysis of the numerical data; YW and WCL carried out the numerical tests, and drafted the manuscript; CSC critically revised the manuscript. All authors gave final approval for publication and agree to be held accountable for the work performed therein.

Competing Interests. The authors declare no competing interests.

Funding. CSC is supported by NSF DMS 1813071 and DMS 1253481. WCL is supported by the General Research Fund of the Research Grants Council of the Hong Kong Special Administrative Region (Project No. CityU 11303117) and CityU Strategic Research Grant (Project No. CityU 11300518).

Acknowledgements. WCL thanks for the support of Mathematical Biosciences Institute during his visit at Ohio State University.

Appendix

(a) Derivation of the limit damage band

Suppose a stem cell currently has damage level a_0 in the beginning of a cell cycle. We want to record the damage in the following cycles. At the end of the first cycle, the damage accumulates to $a_0 + p^*/v_p$. After division, a stem daughter cell inherits damage $(a_0 + p^*/v_p)\ell_1$ with $\ell_1 \in \{\alpha_1, \alpha_2, \gamma_1\}$, whose damage gets to $(a_0 + p^*/v_p)\ell_1 + p^*/v_p = a_0\ell_1 + (p^*/v_p)(\ell_1 + 1)$ at the end of the second cycle. After division, a stem daughter cell inherits damage $(a_0\ell_1 + (p^*/v_p)(\ell_1 + 1))\ell_2$ with $\ell_2 \in \{\alpha_1, \alpha_2, \gamma_1\}$, and the damage accumulates to $(a_0\ell_1 + (p^*/v_p)(\ell_1 + 1))\ell_2 + tp^*/v_p = a_0\ell_1\ell_2 + (p^*/v_p)(\ell_1\ell_2 + \ell_2 + 1)$ at the end of the third cycle. By induction, we can easily show that the damage in a stem daughter cell (if not dead yet) at the end of the m -th cycle is

$$a_0 \prod_{i=0}^{m-1} \ell_i + (p^*/v_p) \sum_{i=0}^{m-1} \prod_{k=i}^{m-1} \ell_k \quad \text{with } \ell_0 = 1 \text{ and } \ell_i \in \{\alpha_1, \alpha_2, \gamma_1\} \text{ for } i \geq 1.$$

Let ω_1 and ω_2 be the minimum and the maximum of $\alpha_1, \alpha_2, \gamma_1$, respectively. In the above, considering a fixed segregation portion $\ell_i = \omega_j$ for all $i \geq 1$, we get the damage level

$$a_0 \omega_j^{m-1} + (p^*/v_p) \frac{1 - \omega_j^m}{1 - \omega_j}, \quad j = 1, 2.$$

The limits as $m \rightarrow \infty$ yields the limit damage band

$$\left[\frac{p^*}{v_p(1 - \omega_1)}, \frac{p^*}{v_p(1 - \omega_2)} \right].$$

(b) Proofs of the propositions

(i) Analytic solution

Before we prove the propositions, we first obtain the analytic solution which can provide us a tool used in the proofs. To simplify the analysis, we make the following change of variables which will be used in the proofs:

$$\tilde{p} = \frac{p}{v_p}, \quad \tilde{a} = \frac{a}{v_a},$$

and the domains of new variables are $0 \leq \tilde{p} \leq t_p = \frac{p^*}{v_p}$ and $0 \leq \tilde{a} \leq t_a = \frac{a^*}{v_a}$. Given constant cell cycle progression and damage accumulation for stem cells, the biological meaning of t_p and t_a can be interpreted as

- t_p is the duration of one cell cycle.
- t_a is the duration needed to accumulate damage to the lethal threshold from zero damage.

With the new variables, (2.1) and (2.2) become

$$\frac{\partial \tilde{S}}{\partial t} + \frac{\partial \tilde{S}}{\partial \tilde{p}} + \frac{\partial \tilde{S}}{\partial \tilde{a}} = 0, \quad (\text{A.1})$$

$$\frac{\partial \tilde{T}}{\partial t} + \frac{u_p}{v_p} \frac{\partial \tilde{T}}{\partial \tilde{p}} + \frac{u_a}{v_a} \frac{\partial \tilde{T}}{\partial \tilde{a}} = 0, \quad (\text{A.2})$$

where $\tilde{S}(t, \tilde{p}, \tilde{a}) = S(t, p, a)$ and $\tilde{T}(t, \tilde{p}, \tilde{a}) = T(t, p, a)$. For simplicity, we will drop all the tildes on the notations. The dynamics of stem cells is independent of TD cells, while the dynamics of TD cells is determined by stem cell through boundary condition (3.2). Once the behavior of stem cell is determined, the behavior of TD cell can be easily derived. So we mainly focus on the dynamics of stem cells.

After a change of variable, (A.1) together with the initial condition and boundary condition (3.1) can be solved by the method of characteristics. Due to the complexity of boundary conditions, no closed form of $S(t, p, a)$ can be obtained, and the solution is expressed in a recursive form.

For the first cell cycle, i.e., $0 < t < t_p$, we have three cases:

(i) If $t \leq p$ and $t \leq a$, the solution is determined by the initial condition

$$S(t, p, a) = S(0, p - t, a - t); \quad (\text{A.3})$$

(ii) If $t > a$ and $a < p$, the solution is determined by the boundary condition on $a = 0$

$$S(t, p, a) = S(t - a, p - a, 0); \quad (\text{A.4})$$

(iii) If $t > p$ and $p < a$, the solution is determined by boundary condition on $p = 0$

$$S(t, p, a) = S(t - p, 0, a - p). \quad (\text{A.5})$$

According to these three cases, $S(t, p, a)$ can be solved as

$$S(t, p, a) = \begin{cases} S(0, p - t, a - t) & \text{if } t \leq p \text{ and } t \leq a, \\ 0 & \text{if } t > a \text{ and } a < p, \\ H(t, p, a, \alpha_1, t_a) \frac{\delta_1}{\alpha_1} S\left(0, t_p - (t - p), \frac{a - p}{\alpha_1} - (t - p)\right) \\ \quad + H(t, p, a, \alpha_2, t_a) \frac{\delta_1}{\alpha_2} S\left(0, t_p - (t - p), \frac{a - p}{\alpha_2} - (t - p)\right) \\ \quad + H(t, p, a, \gamma_1, t_a) \frac{\delta_3}{\gamma_1} S\left(0, t_p - (t - p), \frac{a - p}{\gamma_1} - (t - p)\right) & \text{if } t > p \text{ and } p < a, \end{cases} \quad (\text{A.6})$$

where the function H is given by

$$H(t, p, a, \omega, t_a) = \begin{cases} 1 & \text{if } \frac{a-p}{\omega} < t_a \text{ and } \frac{a-p}{\omega} - (t-p) > 0, \\ 0 & \text{otherwise.} \end{cases} \quad (\text{A.7})$$

For the time beyond the first cell cycle, i.e., $t > t_p$, by tracing back the process of damage accumulation by one cycle, $S(t, p, a)$ is solved as

$$\begin{aligned} S(t, p, a) = & H(t_p, p, a, \alpha_1, t_a) \frac{\delta_1}{\alpha_1} S\left(t - t_p, p, \frac{a - p}{\alpha_1} - (t_p - p)\right) \\ & + H(t_p, p, a, \alpha_2, t_a) \frac{\delta_1}{\alpha_2} S\left(t - t_p, p, \frac{a - p}{\alpha_2} - (t_p - p)\right) \\ & + H(t_p, p, a, \gamma_1, t_a) \frac{\delta_3}{\gamma_1} S\left(t - t_p, p, \frac{a - p}{\gamma_1} - (t_p - p)\right). \end{aligned} \quad (\text{A.8})$$

Applying (A.8) iteratively and eventually (A.6), we can obtain the density of the stem cell population at any time.

(ii) Proof of Proposition 3.1

First, we suppose $f_r < 1/2$. Consider any fixed time $t \gg t_p$. By (A.8), we have

$$\begin{aligned} 0 \leq S(t + t_p, p, a) &\leq \frac{\delta_1}{\alpha_1} S\left(t, p, \frac{a-p}{\alpha_1} - (t_p - p)\right) \\ &\quad + \frac{\delta_1}{\alpha_2} S\left(t, p, \frac{a-p}{\alpha_2} - (t_p - p)\right) \\ &\quad + \frac{\delta_3}{\gamma_1} S\left(t, p, \frac{a-p}{\gamma_1} - (t_p - p)\right). \end{aligned}$$

Then by integration and substitutions

$$\begin{aligned} 0 \leq P_S(t + t_p) &= \int_0^{t_p} \int_0^{t_a} S(t + t_p, p, a) da dp \\ &\leq (\delta_1 + \delta_1 + \delta_3) \int_0^{t_p} \int_0^{t_a} S(t, p, a) da dp \\ &= (2f_r)P_S(t). \end{aligned}$$

By the continuity of $P_S(t)$, it attains a maximum P_{\max} in a time interval $[t_0, t_0 + t_p]$ for a fixed t_0 . Repeated application of the above inequality, in view of the arbitrariness of t , implies that for any $n \geq 1$, and for any t in $[t_0 + nt_p, t_0 + (n + 1)t_p]$, we have

$$P_S(t) \leq (2f_r)^n P_{\max},$$

which clearly establishes that $\lim_{t \rightarrow \infty} P_S(t) = 0$.

For the second part, when the limit damage band exceeds the lethal threshold, i.e., $t_a < \frac{t_p}{1-\omega_1}$, we let $\epsilon = \frac{t_p}{1-\omega_1} - t_a > 0$. If we assume that $\frac{a-p}{\omega} < t_a$, where $\omega \in \{\alpha_1, \alpha_2, \gamma_1\}$, we have

$$\begin{aligned} a &< \omega t_a + p, \\ a &< \omega \left(\frac{t_p}{1-\omega} - \epsilon \right) + p, \\ a &> \frac{a-p}{\omega} - (t_p - p) + C, \end{aligned}$$

where $C = (1 - \omega_2)\epsilon > 0$ which is independent to a and p . Suppose that $t \ll t_p$, by (A.8), we have

$$\begin{aligned} S(t, p, a) &= H(t_p, p, a, \alpha_1, t_a) \frac{\delta_1}{\alpha_1} S\left(t - t_p, p, g(a, \alpha_1)\right) \\ &\quad + H(t_p, p, a, \alpha_2, t_a) \frac{\delta_1}{\alpha_2} S\left(t - t_p, p, g(a, \alpha_2)\right) \\ &\quad + H(t_p, p, a, \gamma_1, t_a) \frac{\delta_3}{\gamma_1} S\left(t - t_p, p, g(a, \gamma_1)\right), \end{aligned}$$

where $g(a, \omega) = \frac{a-p}{\omega} - (t_p - p)$. For each term in the right hand side, if $\frac{a-p}{\omega} \geq t_a$, H is zero; if $\frac{a-p}{\omega} < t_a$, we have $g(a, \omega) < a - C$. When $g(a, \omega)$ is less than or equal to zero, the term $S\left(t - t_p, p, g(a, \omega)\right) = 0$; when $g(a, \omega)$ is larger than zero, we can repeat the process and express the solution at t in terms of the solutions at $t - nt_p$ where n is a positive integer. Because $g(a, \omega) < a - C$, the damage level will be reduced by a constant value in each process. Hence, with finite value n , all the terms $S\left(t - nt_p, p, g\right)$ will become zero as g is less than zero.

Intuitively, the stem cell population always diminishes to zero, since after sufficiently long time the offsprings of stem cells with initial zero damage, even in the slowest way of accumulating damage, i.e., $\ell_i = \omega_1$ for every i , would gain damage to reach t_a and die, and other offsprings in the evolution gain more damage and would die sooner. This result is true, no matter what the proportions of three division types are.

(iii) Proof of Proposition 3.2

By the assumption that $t_a > \frac{t_p}{1-\omega_2}$, no cell will die due to damage accumulation and f_r completely determines the evolution of stem cell population. If $f_r > 1/2$, the portion of generated stem cells through divisions is greater than 1, which means that the stem cell population is increasing to infinity. Indeed, for any t , let $\omega \in \{\alpha_1, \alpha_2, \gamma_1\}$, we define

$$I(\omega) := \frac{1}{\omega} \int_0^{t_p} \int_0^{\omega_2 t_a + p} H(t_p, p, a, \omega, t_a) S\left(t - t_p, p, \frac{a-p}{\omega} - (t_p - p)\right) da dp.$$

By the definition of H and change of variables with $\omega \leq \omega_1$ and $t_a > t_p/(1-\omega_2)$,

$$\begin{aligned} I(\omega) &\geq \frac{1}{\omega} \int_0^{t_p} \int_{\omega t_p - \omega p + p}^{\omega t_a + p} S\left(t - t_p, p, \frac{a-p}{\omega} - (t_p - p)\right) da dp \\ &= \int_0^{t_p} \int_0^{t_a - t_p + p} S(t - t_p, p, a) da dp \\ &\geq \int_0^{t_p} \int_0^{\omega_2 t_a + p} S(t - t_p, p, a) da dp. \end{aligned} \tag{A.9}$$

By (A.8) and (A.9),

$$\begin{aligned} P_S(t) &\geq \int_0^{t_p} \int_0^{\omega_2 t_a + p} S(t, p, a) da dp \\ &= \delta_1 I(\alpha_1) + \delta_2 I(\alpha_2) + \delta_3 I(\gamma_1) \\ &\geq 2f_r \int_0^{t_p} \int_0^{\omega_2 t_a + p} S(t - t_p, p, a) da dp. \end{aligned}$$

Repeating this procedure, if $t = nt_p + \tilde{t}$ and $\tilde{t} \in [0, t_p)$, then we have

$$P_S(t) \geq (2f_r)^n \int_0^{t_p} \int_0^{t_a - t_p + p} S(\tilde{t}, p, a) da dp.$$

By (A.6), we can have

$$\int_0^{t_p} \int_0^{t_a - t_p + p} S(\tilde{t}, p, a) da dp \geq \int_0^{t_p} \int_0^{t_a - t_p + p} S(0, p, a) da dp.$$

With the assumptions $f_r > 1/2$ and

$$\int_0^{t_p} \int_0^{t_a - t_p + p} S(0, p, a) da dp > 0.$$

Now we proved that the population will blow up when $n \rightarrow \infty$.

Intuitively, after sufficient long time, there is no death by damage accumulation since $t_a \geq \frac{t_p}{1-\omega_2}$. Hence in the situation that $f_r = 1/2$, we can use similar method to show that one of the offsprings after stem cell division remains a stem cell and the stem cell population is conserved.

(iv) Proof of Proposition 3.3

We assume that (3.6) holds:

$$f_r > \frac{1}{2} \quad \text{and} \quad \frac{t_p}{1-\gamma_1} \leq \frac{t_p}{1-\alpha_1} < t_a < \frac{t_p}{1-\alpha_2}.$$

We first prove (a). For integers $m \geq 1$, define

$$n(k) = \min \left\{ m \in \mathbb{N} : t_a < \frac{1-\gamma_1^k}{1-\gamma_1} t_p \alpha_2^m + \frac{1-\alpha_2^m}{1-\alpha_2} t_p \right\}.$$

Let $P_S(kt_p)$ denote the stem cell population at the end of the k -th cycle, when the damage levels in all cells are larger than $a_k = \frac{1-\gamma_1^k}{1-\gamma_1} t_p$, from the discussion in (a). The damage level in stem daughter cells (if not dead yet) after $n(k)$ more cycles would be greater than

$$a_k \prod_{i=0}^{n-1} \ell_i + t_p \sum_{i=0}^{n-1} \prod_{k=i}^{n-1} \ell_k \quad \text{with } \ell_0 = 1 \text{ and } \ell_i \in \{\alpha_1, \alpha_2, \gamma_1\} \text{ for } i \geq 1, \tag{A.10}$$

where $n = n(k)$. If we set $\ell_i = \alpha_2$ for all $0 \leq i \leq n(k)$, the definition of $n(k)$ can imply that the damage level in (A.10) is larger than t_a at the end of the $(k + n(k))$ th cycle, and the daughter cells in this situation will die. Thus, for the possible situations for the cell to survive at the end of the $(k + n(k))$ -th cycle, there should be at least one $\ell_i = \gamma_1$ or α_1 for $1 \leq i \leq n(k)$. Clearly the proportion of such stem daughter cells in the population is

$$(2f_r)^{n(k)} - \delta_1^{n(k)}.$$

Suppose there exists a combination of m and δ_i such that $(2f_r)^{n(k)} - \delta_1^{n(k)} < 1$. By the similar method used in ii, we have

$$P_S((k + n(k))t_p) \leq P_S(kt_p)((2f_r)^{n(k)} - \delta_1^{n(k)}).$$

It is easy to show that the sequence $\{(2f_r)^n - (\delta_1)^n\}$ is increasing in n , under the assumption $f_r > \frac{1}{2}$; the sequence $\{n(k)\}$ is non-increasing in k , i.e., $n(k + 1) \leq n(k)$. Then applying the above inequality with k replaced by $k + n(k)$, one gets

$$\begin{aligned} &P_S((k + n(k) + n(k + n(k)))t_p) \\ &\leq P_S((k + n(k))t_p)((2f_r)^{n(k+n(k))} - \delta_1^{n(k+n(k))}) \\ &\leq P_S((k + n(k))t_p)((2f_r)^{n(k)} - \delta_1^{n(k)}) \\ &\leq P_S(kt_p)((2f_r)^{n(k)} - \delta_1^{n(k)})^2 \end{aligned}$$

Repeating this argument, we have a sequence of population approach to zero as $(2f_r)^{n(k)} - \delta_1^{n(k)} < 1$. Since $\{(2f_r)^{n(k)} - (\delta_1)^{n(k)}\}$ is non-increasing in k , we can just consider the condition $\{(2f_r)^{n(k)} - (\delta_1)^{n(k)}\} < 1$ when k is sufficiently large. Then, we can simplify the condition $\{(2f_r)^n - (\delta_1)^n\} < 1$ with

$$n = \min \left\{ m \in \mathbb{N} : t_a < \frac{1}{1-\gamma_1} t_p \alpha_2^m + \frac{1-\alpha_2^m}{1-\alpha_2} t_p \right\}.$$

Therefore, part (a) is proven. Next we work with (b). Similar to the method used in the proof of iii with the assumption

$$\frac{p^*}{v_p(1-\gamma_1)} \leq \frac{p^*}{v_p(1-\alpha_1)} < t_a,$$

we obtain

$$\begin{aligned} P_S(t) &\geq \int_0^{t_p} \int_0^{\omega_2 t_a + p} S(t, p, a) \, da \, dp \\ &\geq \delta_1 I(\alpha_1) + \delta_3 I(\gamma_1) \\ &\geq (\delta_1 + \delta_3) \int_0^{t_p} \int_0^{\omega_2 t_a + p} S(t - t_p, p, a) \, da \, dp. \end{aligned}$$

When $\delta_2 = 0$, $\delta_1 + \delta_3 = 1$ and

$$P_S(t) \geq \int_0^{t_p} \int_0^{\omega_2 t_a + p} S(t - t_p, p, a) \, da \, dp.$$

By repeating this procedure, we have

$$P_S(t) \geq \int_0^{t_p} \int_0^{t_a - t_p + p} S(0, p, a) da dp > 0$$

which implies that the stem cell population blows up to infinity, or is bounded below to avoid stem cell extinction.

(c) Parameter estimation

(i) Parameters in the model with feedback from TD cells

Although there is no concrete biological data in stem cell division types or specific regulation mechanisms to our knowledge, we discuss the selection of parameters in Table 1 referring to both the biological observations [43–46] and the previous mathematical models [25,26,28,32–34,36–40,54,55]. For the simulations we considered here, we assume that damage segregates symmetrically between stem cells and between TD cells, i.e., $\alpha_1 = 0.5$, $\beta_1 = 0.5$, while the damage retention in asymmetric division is low, i.e., $\gamma_1 = 0.25$.

Cell cycle progression speed and damage accumulation speeds v_p , v_a and u_a

It is generally accepted that the population of mitotic cells is much smaller than that of terminally differentiated cells, although it is very difficult to identify and count stem cells and progenitors [57–60]. Due to lack of experimental data, we refer to some existing modeling works for a reasonable range for the ratio of mitotic cells to TD cells. In the simulations of [36,55], populations of post-mitotic cells are about 4–10 folds of that of mitotic cells.

On the other hand, mitotic cells are assumed to have a smaller death rate than post-mitotic cells [7]. However, there is no precise description of death rates due to two reasons: the death rate of stem cells is not observable, and the death rate of TD cells is tissue specific. In various modeling works, the assumed death rate of TD cells ranges from 10^{-1} to 10^{-4} of the total population [25, 39]. Analysis and simulations in [25,39,40] suggest that a large death rate of TD cells result in instability of population. In our model, the death rates are measured in the following way:

$$r_S = \frac{\int_0^p v_a S(t, p, a^*) dp}{P_S} \quad (\text{A.11})$$

$$r_T = \frac{\int_0^\infty u_a T(t, p, a^c) dp}{P_T} \quad (\text{A.12})$$

where the numerators are population out-fluxes due to death and the denominators are the total populations. By properly choosing v_p , v_a , u_a , we wish to achieve that P_T/P_S is no less than 5, r_T is about 10^{-4} , and r_S is reasonably small, compared to r_T . Here we take $v_p = 0.2$, $v_a = 0.06$ and $u_a = 0.02$.

Maximum fractions of three types of cell division δ_1^0 , δ_2^0 and δ_3^0

It was observed that stem cells mainly undergo asymmetric renewal and differentiation when stem cell pool is expanding or when TD cells suffer from great loss, and enter into an inactively dividing stage and then mainly undergo asymmetric division at steady state [43–46]. Since the feedbacks we consider are negative regulations, δ_1^0 provides the maximum fraction of symmetric renewal. By the property of the Hill function, $\delta_1(P_T)$ can switch between large and small values in response to small and large populations of TD cells. But we observed that as we increase the initial δ_1^0 , population overshoot and oscillations appear before convergence (Fig. 10A). Moreover, the population ratio P_T/P_S decreases (from 5.664 to 3.622) and death rates of stem cell r_S increases (from $r_S = 1.191 \times 10^{-7}$ to 1.365×10^{-4}) as δ_1^0 increases from 0.6 to 0.8. Hence, we take $\delta_1^0 = 0.6$, $\delta_2^0 = 0.3$ and $\delta_3^0 = 0.1$.

Regulation constants k_1^T , k_2^T and k_v^T

Figure 10. Simulations for the model with feedback from TD cells for different settings of parameters.

A) Comparison of different initial division fractions. B) Comparison of different regulation parameters. C) Comparison of Hill exponents. The other parameters which are not listed in the figures are $k_1^T = 10^{-8}$, $k_2^T = 0.5 \times 10^{-8}$, $k_v^T = 0.5 \times 10^{-8}$, $\alpha_1 = 0.5$, $\beta_1 = 0.5$, $\gamma_1 = 0.25$, $v_p = 0.2$, $v_a = 0.06$, $u_a = 0.02$, $m_T = 2$, $\delta_1^0 = 0.6$, $\delta_2^0 = 0.3$, $\delta_3^0 = 0.1$.

Many works suggest that regulation parameters k_i^T and k_v^T are closely related to the TD cell population at steady state [26,28,32,39]. These modeling works mainly focus on the blood system, which is one of the most well studied systems and serves as a paradigm for understanding stem cells. The regulation parameter k in those models ranges from 10^{-7} to 10^{-10} , which corresponds to red blood cell population of size $10^7 \sim 10^{10}$. However, some other works that do not specify the type of stem cells may choose some other ranges. For example, [36] chose 10^{-3} for regulation constant and the resulting population magnitude is about 10^3 . In our simulations, we choose 10^{-8} as the scale for the magnitudes of k_i^T and k_v^T .

On the other hand, by choosing appropriate regulation parameters k_1^T , k_2^T and k_v^T in Eq. (4.3) and Eq. (4.4), asymmetrical division will predominate in three types division of stem cells at steady state. In our simulations shown in Fig. 10B, we found that equal strengths of feedback on stem cell symmetric renewal δ_1 and cell cycle progression V_p would result in oscillations. Stronger regulation on V_p relative to δ_1 is needed to obtain stabilized population evolution (Fig. 10B). We also found that the relative strengths of regulation on symmetric renewal δ_1 and symmetric differentiation δ_2 play a role in determining the death rate of stem cells ($r_S = 2.408 \times 10^{-4}$ when $(k_1^T, k_2^T, k_v^T) = (10^{-8}, 10^{-8}, 0.5 \times 10^{-8})$; $r_S = 1.191 \times 10^{-7}$ when $(k_1^T, k_2^T, k_v^T) = (10^{-8}, 0.5 \times 10^{-8}, 0.5 \times 10^{-8})$). To obtain a reasonably small death rate of stem cells, stronger regulation on δ_2 is also needed. Hence, we take $k_1^T = 10^{-8}$, $k_2^T = 0.5 \times 10^{-8}$ and $k_v^T = 0.5 \times 10^{-8}$.

Hill exponent for feedbacks m_T

The modeling work [40] suggests that a low Hill exponent in the feedback is needed to avoid oscillations, where the authors observed unstable steady states for Hill exponent $m \geq 3$ through direct simulations in their ODE models. We also found similar phenomena in our model, where oscillations appear in the simulations with $m_T > 4$. Thus we need $m_T \in \{1, 2, 3\}$. To make the best estimation of m_T , we also need to consider all biological observations. Our simulations show that when $m_T = 1$ the death rate of stem cell is only about one third of that of TD cells ($r_S = 9.586 \times 10^{-5}$ when $m_T = 1$; $r_S = 1.191 \times 10^{-7}$ when $m_T = 2$), while when $m_T = 3$ there is population overshoot before convergence to steady state although the death rate of stem cell decreases to 10^{-15} (Fig. 10C). Hence we choose $m_T = 2$ in our model.

After a large number of trials, we arrive at the best choice of the parameters in Table 1 with consideration of all biological and mathematical reasons. Under these parameters, the resulting death rates are $r_S = 1.191 \times 10^{-7}$ and $r_T = 2.357 \times 10^{-4}$. The populations at steady state are $P_S = 2.496 \times 10^7$, $P_T = 1.414 \times 10^8$ with the ratio $P_T/P_S = 5.664$. But we need to point out that all our assumptions are pieced together through different biological and mathematical research, more precise estimations of parameters need real experimental data.

(ii) Parameters in the model with feedbacks from TD cells and stem cells

Inspired by the analysis in Section i, we selected the most reasonable parameters in Table 2, based on both biological and mathematical reasons as follows.

Regulation constants k_1^S , k_2^S and k_v^S , and Hill exponent for feedbacks m_S

Similar to Section i, due to the consideration of reasonable population ratio and death rate of stem cells, we still assume that the feedbacks from stem cell population on cell cycle progression V_p and symmetric differentiation δ_2 are stronger than that on symmetric renewal δ_1 . That is, we intentionally choose smaller k_v^S , k_2^S , compared to k_1^S . And the Hill exponent m_S is chosen to be 2.

To determine the magnitude of k_1^S , we simulate the population evolution with k_1^S ranging from 10^{-6} to 10^{-8} . We found that the relative magnitude of k_1^T and k_1^S has little influence on the population ratio. To effectively model the regulation from the stem cell population, 10^{-7} turns out to be the best choice, according to our simulations under the choice of parameters in Table 1. Hence, we take $k_1^S = 10^{-7}$, $k_2^S = 0.25 \times 10^{-7}$ and $k_v^S = 0.5 \times 10^{-7}$.

Constants for sigmoid regulation

According to experiments, mitotic cells inheriting more damage protein aggregates have an increased cell cycle length and tendency to differentiate [14,16]. To model these observations, we assume that the multipliers $f(a)$ and $g(a)$ decrease in a and have sigmoid shape. Parameters a_i and b_i in the multipliers determine their maximum and minimum strengths, a_i^0 locates the threshold in damage of the transition and k_i^a controls the stiffness of the transition. Due to lack of experimental data or previous modeling works to refer to, we assume $f(a)$ and $g(a)$ take the same form and range between [0.4, 1.1] with a transition at $a_1^0 = 0.75$. Hence, we take $a_1 = a_2 = 1.1$, $b_1 = b_2 = -0.7$, $k_1^a = k_2^a = 20$ and $a_1^0 = 0.75$.

With a large number of simulations, we present the best choice of the parameters in Table 2 and conclude that this set of parameters is our best estimations. Under these parameters, the resulting death rates are $r_S = 2.560 \times 10^{-7}$, $r_T = 2.314 \times 10^{-4}$ for stem cells and TD cells, respectively. The populations at steady state are $P_S = 9.593 \times 10^6$, $P_T = 6.709 \times 10^7$ with the ratio $P_T/P_S = 6.994$.

(d) Estimation of populations in the models with feedbacks from TD cells

First we recall the assumptions: $a^* \leq a^c$, $V_a = v_a$, $U_p = u_p$ and $U_a = u_a$ are constants, and δ_i and V_p have feedbacks from TD cells described by (4.3) and (4.4). The populations of stem and TD

cells at time t are give by

$$P_S(t) = \int_0^{a^*} \int_0^{p^*} S(t, p, a) dp da, \quad (\text{A.13})$$

$$P_T(t) = \int_0^{a^c} \int_0^\infty T(t, p, a) dp da. \quad (\text{A.14})$$

Under the above assumptions, we integrate (2.1) and (2.2) in p and a to get

$$\frac{dP_S}{dt} = -v_a \int_0^{p^*} S(t, p, a^*) - S(t, p, 0) dp - V_p(P_T) \int_0^{a^*} S(t, p^*, a) - S(t, 0, a) da, \quad (\text{A.15})$$

$$\frac{dP_T}{dt} = -u_a \int_0^\infty T(t, p, a^c) - T(t, p, 0) dp + u_p \int_0^{a^c} T(t, 0, a) da. \quad (\text{A.16})$$

By boundary conditions (2.5), (2.6), (3.1), and (3.2), we rewrite the above equations as

$$\begin{aligned} \frac{dP_S}{dt} &= -v_a \int_0^{p^*} S(t, p, a^*) dp - V_p(P_T) \left[\int_0^{a^*} S(t, p^*, a) da - \frac{\delta_1(P_T)}{\alpha_1} \int_0^{a^*} S\left(t, p^*, \frac{a}{\alpha_1}\right) da \right. \\ &\quad \left. - \frac{\delta_1(P_T)}{\alpha_2} \int_0^{a^*} S\left(t, p^*, \frac{a}{\alpha_2}\right) da - \frac{\delta_3(P_T)}{\gamma_1} \int_0^{a^*} S\left(t, p^*, \frac{a}{\gamma_1}\right) da \right], \\ \frac{dP_T}{dt} &= -u_a \int_0^\infty T(t, p, a^c) dp + V_p(P_T) \left[\frac{\delta_2(P_T)}{\beta_1} \int_0^{a^c} S\left(t, p^*, \frac{a}{\beta_1}\right) da \right. \\ &\quad \left. + \frac{\delta_2(P_T)}{\beta_2} \int_0^{a^c} S\left(t, p^*, \frac{a}{\beta_2}\right) da + \frac{\delta_3(P_T)}{\gamma_2} \int_0^{a^c} S\left(t, p^*, \frac{a}{\gamma_2}\right) da \right]. \end{aligned}$$

Making a change of variable in a and noticing that $a^* \leq a^c$, we obtain a system of ODEs, called system I:

$$\frac{dP_S}{dt} = -v_a \int_0^{p^*} S(t, p, a^*) dp + V_p(P_T)(\delta_1(P_T) - \delta_2(P_T)) \int_0^{a^*} S(t, p^*, a) da, \quad (\text{A.17})$$

$$\frac{dP_T}{dt} = -u_a \int_0^\infty T(t, p, a^c) dp + V_p(P_T)(1 + \delta_2(P_T) - \delta_1(P_T)) \int_0^{a^*} S(t, p^*, a) da. \quad (\text{A.18})$$

Consider a related system II given by the following equation and (A.18):

$$\frac{dP_S}{dt} = V_p(P_T)(\delta_1(P_T) - \delta_2(P_T)) \int_0^{a^*} S(t, p^*, a) da. \quad (\text{A.19})$$

Then under the same initial condition, the solution to system I is not larger than that to system II, by comparison theorems for ODEs.

Now we assume that both systems I and II have steady state solutions, denoted by and (\bar{P}_S, \bar{P}_T) and $(\tilde{P}_S, \tilde{P}_T)$, respectively. Then \tilde{P}_T can be obtained by solving

$$\delta_1(P_T) - \delta_2(P_T) = 0.$$

Assuming appropriate values of parameters to guarantee solvability of the last equation, we have the upper-bound estimate

$$\bar{P}_T \leq \tilde{P}_T = \left(\frac{\delta_1^0 - \delta_2^0}{\delta_2^0(k_1^T)^{m_T} - \delta_1^0(k_2^T)^{m_T}} \right)^{\frac{1}{m_T}}. \quad (\text{A.20})$$

Next we continue to estimate \bar{P}_S . First we observe that in the steady state both $S(t, p, a)$ and $T(t, p, a)$ are constant along characteristic curves and we refer the reader to Fig. 11 for an illustration of the situation. Since in system II there is no outflux of stem cells on the boundary

Figure 11. Demonstration of upper bound estimation of the population.

$a = a^*$, we have

$$\tilde{P}_S = p^* \int_0^{a^*} S(t, p^*, a) da.$$

On the other hand, by the intermediate value theorem for integrals, we have, for some $h \in [h_1, h_2]$ that

$$\tilde{P}_T = h \int_0^\infty T(t, p, a^c) dp,$$

where

$$h_1 = a^c - \frac{\eta_2}{1 - \omega_2} \Delta a \quad \text{and} \quad h_2 = a^c - \frac{\eta_1}{1 - \omega_1} \Delta a$$

with $\eta_1 = \min\{\beta_1, \beta_2, \gamma_2\}$, $\eta_2 = \max\{\beta_1, \beta_2, \gamma_2\}$ and $\Delta a = \frac{p^*}{V_p(\tilde{P}_T)} v_a$. Using that \tilde{P}_T is a steady state population, we have by (A.17)

$$u_a \tilde{P}_T = h V_p(\tilde{P}_T) \int_0^{a^*} S(t, p^*, a) da = h V_p(\tilde{P}_T) \tilde{P}_S / p^*$$

and thus the estimate

$$\bar{P}_S \leq \tilde{P}_S = \frac{u_a p^*}{h V_p(\tilde{P}_T)} \tilde{P}_T \leq \frac{u_a p^*}{V_p(\tilde{P}_T) h_1} \tilde{P}_T, \quad (\text{A.21})$$

where \tilde{P}_T is given in (A.20).

Since in Section (a) we intentionally choose the parameters such that the death rate of stem cells is small, i.e., the outflux $v_a \int_0^{p^*} S(t, p, a^*) dp$ is small, the upper bound in (A.20) is very close to the real TD cell population in the steady state. However, the estimation for stem cells may not be sharp, due to the variation of the segregation rules. As an example, with the parameters in Table 1, we have that the estimated populations are $(\tilde{P}_S, \tilde{P}_T) = (\frac{6}{13} \sqrt{2} \times 10^7, \sqrt{2} \times 10^8)$ and the steady state populations in the simulation are $(\bar{P}_S, \bar{P}_T) = (2.496 \times 10^7, 1.414 \times 10^8)$.

(e) Numerical Scheme

Consider functions V_p, V_a, U_p, U_a dependent on p, a only, Eq. (2.1) and Eq. (2.2) can be rewritten as the following system, with $W = [S, T]^t$:

$$\frac{\partial}{\partial t}W + \frac{\partial}{\partial p}F(W) + \frac{\partial}{\partial a}G(W) = 0, \quad (\text{A.22})$$

where

$$F(W) = [f_1(W), f_2(W)]^t = [V_p S, U_p T]^t$$

and

$$G(W) = [g_1(W), g_2(W)]^t = [V_a S, U_a T]^t.$$

In the following simulation, we use third-order WENO scheme and third-order TVD Runge-Kutta time integrator.

As U_p, U_a, V_p, V_a are all positive, the numerical approximation $W_{i,j}$ to the exact solution $W(p_i, a_j, t)$ satisfies the following ODE system:

$$\frac{dW_{i,j}(t)}{dt} = -\frac{\hat{F}_{i+1/2,j} - \hat{F}_{i-1/2,j}}{\Delta p} - \frac{\hat{G}_{i,j+1/2} - \hat{G}_{i,j-1/2}}{\Delta a}, \quad (\text{A.23})$$

where $\hat{F}_{i+1/2,j}$ is called numerical flux, the design of which is the key ingredient to a successful scheme. For the third-order WENO scheme, the numerical flux $\hat{F}_{i+1/2,j}$ is defined as follows:

$$\hat{F}_{i+1/2,j} = \omega_1 \hat{F}_{i+1/2,j}^{(1)} + \omega_2 \hat{F}_{i+1/2,j}^{(2)}, \quad (\text{A.24})$$

where $\hat{F}_{i+1/2,j}^{(m)}$ for $m = 1, 2$, are the two second-order accurate fluxes on two different stencils given by

$$\hat{F}_{i+1/2,j}^{(1)} = -\frac{1}{2}F_{i-1,j} + \frac{3}{2}F_{i,j} \quad \hat{F}_{i+1/2,j}^{(2)} = \frac{1}{2}F_{i,j} + \frac{1}{2}F_{i+1,j}. \quad (\text{A.25})$$

The nonlinear weights ω_m are given by

$$\omega_m = \frac{\alpha_m}{\sum_{k=1}^2 \alpha_k}, \quad m = 1, 2, \quad (\text{A.26})$$

where

$$\alpha_k = \frac{\gamma_k}{(\varepsilon + \beta_k)^2} \quad k = 1, 2, \quad (\text{A.27})$$

and

$$\beta_1 = (F_{i,j} - F_{i-1,j})^2 \quad \beta_2 = (F_{i+1,j} - F_{i,j})^2 \quad (\text{A.28})$$

and

$$\gamma_1 = \frac{1}{3}, \quad \gamma_2 = \frac{2}{3}. \quad (\text{A.29})$$

The parameter ε ensures that the denominator never gets to 0, and is fixed at $\varepsilon = 10^{-6}$ in the computation in this work. Similar construction can be applied to the direction of a .

The time concretization is implemented by a third-order TVD Runge-Kutta method:

$$W^{(1)} = W^n + \Delta t L(W^n, t^n), \quad (\text{A.30})$$

$$W^{(2)} = \frac{3}{4}W^n + \frac{1}{4}W^{(1)} + \frac{1}{4}\Delta t L(W^{(1)}, t^n + \Delta t),$$

$$W^{n+1} = \frac{1}{3}W^n + \frac{2}{3}W^{(2)} + \frac{2}{3}\Delta t L(W^{(2)}, t^n + \frac{1}{2}\Delta t),$$

where L denotes the RHS of Eq.(71).

A CFL condition is needed for stability:

$$\alpha \frac{\Delta t}{\min\{\Delta p, \Delta a\}} < 1, \quad (\text{A.31})$$

where $\alpha = \max\{U_p, U_a, V_p, V_a\}$. In this paper, we take $\Delta p = \Delta a = 0.01$ and $\Delta t = 0.005$.

The code for the numerical method can be found in electronic supplementary material.

References

- Morrison SJ, Kimble J. 2006 Asymmetric and symmetric stem-cell divisions in development and cancer. *Nature* **441**, 1068–1074.
- Knoblich JA. 2008 Mechanisms of asymmetric stem cell division. *Cell* **132**, 583–597.
- Jilkin A. 2019 Mathematical Models of Stem Cell Differentiation and Dedifferentiation. *Current Stem Cell Reports* **5**, 66–72.
- Rando TA. 2006 Stem cells, ageing and the quest for immortality. *Nature* **441**, 1080–1086.
- Liu L, Rando TA. 2011 Manifestations and mechanisms of stem cell aging. *The Journal of cell biology* **193**, 257–266.
- Adams PD, Jasper H, Rudolph KL. 2015 Aging-induced stem cell mutations as drivers for disease and cancer. *Cell stem cell* **16**, 601–612.
- Schultz MB, Sinclair DA. 2016 When stem cells grow old: phenotypes and mechanisms of stem cell aging. *Development* **143**, 3–14.
- Terskikh VV, Vasil'ev AV, Voroteliak EA. 2009 Stem cell self-renewal: the role of asymmetric division. *Biology bulletin* **36**, 425–429.
- Aguilaniu H, Gustafsson L, Rigoulet M, Nyström T. 2003 Asymmetric inheritance of oxidatively damaged proteins during cytokinesis. *Science* **299**, 1751–1753.
- Shcheprova Z, Baldi S, Frei SB, Gonnet G, Barral Y. 2008 A mechanism for asymmetric segregation of age during yeast budding. *Nature* **454**, 728–734.
- Higuchi-Sanabria R, Charalel JK, Viana MP, Garcia EJ, Sing CN, Koenigsberg A, Swayne TC, Vevea JD, Boldogh IR, Rafelski SM et al.. 2016 Mitochondrial anchorage and fusion contribute to mitochondrial inheritance and quality control in the budding yeast *Saccharomyces cerevisiae*. *Molecular biology of the cell* **27**, 776–787.
- Fuentealba LC, Eivers E, Geissert D, Taelman V, De Robertis E. 2008 Asymmetric mitosis: Unequal segregation of proteins destined for degradation. *Proceedings of the National Academy of Sciences* **105**, 7732–7737.
- Bufalino MR, DeVeale B, van der Kooy D. 2013 The asymmetric segregation of damaged proteins is stem cell–type dependent. *J Cell Biol* pp. jcb–201207052.
- Bufalino MR, van der Kooy D. 2014 The aggregation and inheritance of damaged proteins determines cell fate during mitosis. *Cell Cycle* **13**, 1201–1207.
- Katajisto P, Döhla J, Chaffer CL, Pentinmikko N, Marjanovic N, Iqbal S, Zoncu R, Chen W, Weinberg RA, Sabatini DM. 2015 Asymmetric apportioning of aged mitochondria between daughter cells is required for stemness. *Science* **348**, 340–343.
- Moore DL, Pilz GA, Araúz-Bravo MJ, Barral Y, Jessberger S. 2015 A mechanism for the segregation of age in mammalian neural stem cells. *Science* **349**, 1334–1338.
- Espada L, Ermolaeva MA. 2016 DNA damage as a critical factor of stem cell aging and organ homeostasis. *Current Stem Cell Reports* **2**, 290–298.
- Moore DL, Jessberger S. 2017 Creating age asymmetry: consequences of inheriting damaged goods in mammalian cells. *Trends in Cell Biology* **27**, 82–92.
- Edelstein-Keshet L, Israel A, Lansdorp P. 2001 Modelling perspectives on aging: can mathematics help us stay young?. *Journal of theoretical biology* **213**, 509–525.
- Mooney KM, Morgan AE, Mc Auley MT. 2016 Aging and computational systems biology. *Wiley Interdisciplinary Reviews: Systems Biology and Medicine* **8**, 123–139.
- Loeffler M, Roeder I. 2004 Conceptual models to understand tissue stem cell organization. *Current Opinion in Hematology* **11**, 81–87.
- Roeder I, Loeffler M. 2002 A novel dynamic model of hematopoietic stem cell organization based on the concept of within-tissue plasticity. *Experimental Hematology* **30**, 853–61.
- Portugal R, Land M, Svaiter BF. 2008 A computational model for telomere-dependent cell-replicative aging. *BioSystems* **91**, 262–267.
- McHale PT, Lander AD. 2014 The protective role of symmetric stem cell division on the accumulation of heritable damage. *PLoS Comput Biol* **10**, e1003802.
- Crauste F, Pujo-Menjouet L, Génieys S, Molina C, Gandrillon O. 2008 Adding self-renewal in committed erythroid progenitors improves the biological relevance of a mathematical model of erythropoiesis. *Journal of Theoretical Biology* **250**, 322–338.

26. Doumic M, Marciniak-Czochra A, Perthame B, Zubelli JP. 2011 A structured population model of cell differentiation. *SIAM Journal on Applied Mathematics* **71**, 1918–1940.
27. Gwiazda P, Jamróz G, Marciniak-Czochra A. 2012 Models of discrete and continuous cell differentiation in the framework of transport equation. *SIAM Journal on Mathematical Analysis* **44**, 1103–1133.
28. Getto P, Marciniak-Czochra A, Nakata Y et al.. 2013 Global dynamics of two-compartment models for cell production systems with regulatory mechanisms. *Mathematical biosciences* **245**, 258–268.
29. Klose M, Florian M, Gerbaulet A, Geiger H, Glauche I. 2019 Hematopoietic stem cell dynamics are regulated by progenitor demand: lessons from a quantitative modeling approach. *Stem Cells* **37**, 948–957.
30. Komarova NL, van den Driessche P. 2018 Stability of control networks in autonomous homeostatic regulation of stem cell lineages. *Bulletin of Mathematical Biology* **80**, 1345–1365.
31. Renardy M, Jilkine A, Shahriyari L, Chou CS. 2018 Control of cell fraction and population recovery during tissue regeneration in stem cell lineages. *Journal of Theoretical Biology* **445**, 33–50.
32. Marciniak-Czochra A, Stiehl T, Ho AD, Jäger W, Wagner W. 2009 Modeling of asymmetric cell division in hematopoietic stem cells—regulation of self-renewal is essential for efficient repopulation. *Stem cells and development* **18**, 377–386.
33. Stiehl T, Marciniak-Czochra A. 2011 Characterization of stem cells using mathematical models of multistage cell lineages. *Mathematical and Computer Modelling* **53**, 1505–1517.
34. Nakata Y, Getto P, Marciniak-Czochra A, Alarcón T. 2012 Stability analysis of multi-compartment models for cell production systems. *Journal of Biological Dynamics* **6**, 2–18.
35. Chou CS, Lo WC, Gokoffski KK, Zhang YT, Wan FY, Lander AD, Calof AL, Nie Q. 2010 Spatial dynamics of multistage cell lineages in tissue stratification. *Biophysical Journal* **99**, 3145–3154.
36. Rodriguez-Brenes IA, Wodarz D, Komarova NL. 2013 Stem cell control, oscillations, and tissue regeneration in spatial and non-spatial models. *Frontiers in oncology* **3**, 82.
37. Komarova NL. 2013 Principles of regulation of self-renewing cell lineages. *PloS one* **8**, e72847.
38. Buzi G, Lander AD, Khammash M. 2015 Cell lineage branching as a strategy for proliferative control. *BMC biology* **13**, 13.
39. Bélair J, Mackey MC, Mahaffy JM. 1995 Age-structured and two-delay models for erythropoiesis. *Mathematical biosciences* **128**, 317–346.
40. Lo WC, Chou CS, Gokoffski KK, Wan FYM, Lander AD, Calof AL, Nie Q. 2009 Feedback regulation in multistage cell lineages. *Mathematical biosciences and engineering: MBE* **6**, 59.
41. Roeder I, Herberg M, Horn M. 2009 An "age" structured model of hematopoietic stem cell organization with application to chronic myeloid leukemia. *Bulletin of Mathematical Biology* **71**, 602–26.
42. Kapitanov G. 2012 A mathematical model of cancer stem cell lineage population dynamics with mutation accumulation and telomere length hierarchies. *Mathematical Modelling of Natural Phenomena* **7**, 136–165.
43. Lechler T, Fuchs E. 2005 Asymmetric cell divisions promote stratification and differentiation of mammalian skin. *Nature* **437**, 275–280.
44. Noctor SC, Martínez-Cerdeño V, Ivic L, Kriegstein AR. 2004 Cortical neurons arise in symmetric and asymmetric division zones and migrate through specific phases. *Nature neuroscience* **7**, 136–144.
45. Bodine DM, Seidel NE, Orlic D. 1996 Bone marrow collected 14 days after in vivo administration of granulocyte colony-stimulating factor and stem cell factor to mice has 10-fold more repopulating ability than untreated bone marrow. *Blood* **88**, 89–97.
46. Zhang R, Zhang Z, Zhang C, Zhang L, Robin A, Wang Y, Lu M, Chopp M. 2004 Stroke transiently increases subventricular zone cell division from asymmetric to symmetric and increases neuronal differentiation in the adult rat. *Journal of Neuroscience* **24**, 5810–5815.
47. Watt FM, Hogan BL. 2000 Out of Eden: stem cells and their niches. *Science* **287**, 1427–1430.
48. Yang J, Axelrod DE, Komarova NL. 2017 Determining the control networks regulating stem cell lineages in colonic crypts. *Journal of Theoretical Biology* **429**, 190–203.

49. McPherron AC, Lawler AM, Lee SJ. 1997 Regulation of skeletal muscle mass in mice by a new TGF-beta superfamily member. *nature* **387**, 83.
50. Daluiski A, Engstrand T, Bahamonde ME, Gamar LW, Agius E, Stevenson SL, Cox K, Rosen V, Lyons KM. 2001 Bone morphogenetic protein-3 is a negative regulator of bone density. *Nature genetics* **27**, 84–88.
51. Yamasaki K, Toriu N, Hanakawa Y, Shirakata Y, Sayama K, Takayanagi A, Ohtsubo M, Gamou S, Shimizu N, Fujii M et al.. 2003 Keratinocyte growth inhibition by high-dose epidermal growth factor is mediated by transforming growth factor β autoinduction: a negative feedback mechanism for keratinocyte growth. *Journal of investigative dermatology* **120**, 1030–1037.
52. Wu HH, Ivkovic S, Murray RC, Jaramillo S, Lyons KM, Johnson JE, Calof AL. 2003 Autoregulation of neurogenesis by GDF11. *Neuron* **37**, 197–207.
53. Tzeng YS, Li H, Kang YL, Chen WC, Cheng WC, Lai DM. 2011 Loss of Cxcl12/Sdf-1 in adult mice decreases the quiescent state of hematopoietic stem/progenitor cells and alters the pattern of hematopoietic regeneration after myelosuppression. *Blood* **117**, 429–439.
54. Lei J, Levin SA, Nie Q. 2014 Mathematical model of adult stem cell regeneration with cross-talk between genetic and epigenetic regulation. *Proceedings of the National Academy of Sciences* **111**, E880–E887.
55. Yang J, Plikus MV, Komarova NL. 2015 The role of symmetric stem cell divisions in tissue homeostasis. *PLoS Comput Biol* **11**, e1004629.
56. Salomoni P, Calegari F. 2010 Cell cycle control of mammalian neural stem cells: putting a speed limit on G1. *Trends in Cell Biology* **20**, 233–43.
57. Scadden DT. 2006 The stem-cell niche as an entity of action. *Nature* **441**, 1075–1079.
58. Morrison SJ, Spradling AC. 2008 Stem cells and niches: mechanisms that promote stem cell maintenance throughout life. *Cell* **132**, 598–611.
59. Walker M, Patel K, Stappenbeck T. 2009 The stem cell niche. *The Journal of pathology* **217**, 169–180.
60. Discher DE, Mooney DJ, Zandstra PW. 2009 Growth factors, matrices, and forces combine and control stem cells. *Science* **324**, 1673–1677.
61. Lander AD, Gokoffski KK, Wan FYM, Nie Q, Calof AL. 2009 Cell Lineages and the Logic of Proliferative Control. *PLOS Biology* **7**, 1–1.